# An Expanded Benchmark that Rediscovers and Affirms the Edge of Uncertainty Sampling for Active Learning in Tabular Datasets

**Po-Yi Lu**                                                                 *d09944015@csie.ntu.edu.tw*
*National Taiwan University, Taipei, Taiwan*

**Yi-Jie Cheng**                                                            *eva.cheng1214@gmail.com*
*National Taiwan University, Taipei, Taiwan*

**Chun-Liang Li**                                                        *chunlial@cs.washington.edu*
*University of Washington, WA, USA*

**Hsuan-Tien Lin**                                                             *htlin@csie.ntu.edu.tw*
*National Taiwan University, Taipei, Taiwan*

**Reviewed on OpenReview:** *https: // openreview. net/ forum? id= 855yo1Ubt2*

## Abstract

Active Learning (AL) addresses the crucial challenge of enabling machines to efficiently gather labeled examples through strategic queries. Among the many AL strategies, Uncertainty Sampling (US) stands out as one of the most widely adopted. US queries the example(s) that the current model finds uncertain, proving to be both straightforward and effective. Despite claims in the literature suggesting superior alternatives to US, community-wide acceptance remains elusive. In fact, existing benchmarks for tabular datasets present conflicting conclusions on the continued competitiveness of US. In this study, we review the literature on AL strategies in the last decade and build the most comprehensive open-source AL benchmark to date to understand the relative merits of different AL strategies. The benchmark surpasses existing ones by encompassing a broader coverage of strategies, models, and data. Through our investigation of the conflicting conclusions in existing tabular AL benchmarks by evaluation under broad AL experimental settings, we uncover fresh insights into the often-overlooked issue of using machine learning models–**model compatibility** in the context of US. Specifically, we notice that adopting the different models for the querying unlabeled examples and learning tasks would degrade US's effectiveness. Notably, our findings affirm that US maintains a competitive edge over other strategies when paired with compatible models. These findings have practical implications and provide a concrete recipe for AL practitioners, empowering them to make informed decisions when working with tabular classifications with limited labeled data. The code for this project is available on `https://github.com/ariapoy/active-learning-benchmark`.

## 1 Introduction

Supervised learning models can achieve competitive results with sufficient high-quality labeled data. However, acquiring such data can be costly in specific domains. This situation calls for Active Learning (AL), a learning paradigm that strategically selects the most valuable unlabeled examples for labeling. AL has the capability of achieving better performance with lower labeling costs, which has been widely studied and applied in various domains, such as computer vision (Li & Guo, 2013; Demir et al., 2015; Beluch et al., 2018),

Table 1: Comparison between Yang & Loog (2018); Zhan et al. (2021) and our benchmark. (D) means aspects of datasets; (M) means aspects of base models; (Q) means aspects of query strategies; (A) means aspects of analysis; (O) means aspects of an open source tool. Our benchmark fetches up lacking query strategies in Yang & Loog (2018) and lacking analysis in Zhan et al. (2021) to provide a comprehensive comparison.

|     |                              | Yang & Loog (2018) | Zhan et al. (2021) | Ours |
|-----|------------------------------|--------------------|--------------------|------|
| (D) | More than 100K examples      | ✓                  |                    | ✓    |
| (D) | More than 400 features       | ✓                  |                    | ✓    |
| (M) | LR                           | ✓                  |                    | ✓    |
| (M) | RBFSVM                       |                    | ✓                  | ✓    |
| (M) | RF                           |                    |                    | ✓    |
| (Q) | Model uncertainty            | ✓                  | ✓                  | ✓    |
| (Q) | Bayesian uncertainty         |                    |                    | ✓    |
| (Q) | Data diversity               |                    | ✓                  | ✓    |
| (Q) | Hybrid criteria              | ✓                  | ✓                  | ✓    |
| (Q) | Redesigned learning framework|                    | ✓                  | ✓    |
| (A) | AUBC                         | ✓                  | ✓                  | ✓    |
| (A) | Average ranking              | ✓                  |                    | ✓    |
| (A) | Comparison with Uniform      | ✓                  |                    | ✓    |
| (O) | Released datasets            |                    | ✓                  | ✓    |
| (O) | Unified AL protocol          |                    |                    | ✓    |
| (O) | Analysis tools               |                    |                    | ✓    |

natural language processing (Liu et al., 2021; Schröder et al., 2021; Kishaan et al., 2020), and biology and medical fields (Hao et al., 2020; Nath et al., 2020; Logan et al., 2022).

Among the many AL strategies, Uncertainty Sampling (US) stands out as a straightforward and efficient query strategy by selecting the most uncertain examples for labeling based on the model's prediction confidence. US has demonstrated success across multiple applications (Kishaan et al., 2020; Narayanan et al., 2020; Nath et al., 2020); while US is widely used, several AL studies have developed more sophisticated query strategies to address specific limitations in particular scenarios (Donmez et al., 2007; Huang et al., 2010; Li et al., 2015).

Two large-scale benchmarks for pool-based AL have been developed to evaluate existing strategies for classification on tabular datasets (Yang & Loog, 2018; Zhan et al., 2021). However, they present conflicting conclusions regarding the preferred query strategies. While Yang & Loog (2018) suggested that the straightforward US strategy excels across the majority of datasets, Zhan et al. (2021) argued that Learning Active Learning (LAL) (Konyushkova et al., 2017) outperforms US.

Given the lack of consistent comparisons across diverse contexts and the contradictory conclusions drawn from the previous two extensive benchmarks, there is a critical need for a benchmark that accurately represents the current state of AL techniques in this field. Therefore, this work aims to build the most comprehensive AL benchmark compared to previous benchmarks, focusing on datasets, base models, query strategies, and analysis aspects, as highlighted in Table 1. Our benchmark is the most comprehensive open-source framework to date, crafted by integrating a transparent and unified interface. This unified interface cooperates with existing GitHub repositories, such as libact (Yang et al., 2017), Google AL playground (Yilei "Dolee" Yang, 2017), ALiPy (Tang et al., 2019), ModAL (Danka & Horvath), scikit-activeml (Kottke et al., 2021), and sets a new standard for future research.

Subsequently, we assess the performance of query strategies specifically for classifications on tabular data, which is widely used in various real-world applications due to its structured nature and the availability of

diverse datasets. Our benchmarking results show that US is SOTA on 18 of the 29 binary-class datasets and 5 of the 7 multi-class datasets.

Furthermore, through our investigation under different AL experimental settings, we uncover the reason for the substandard performance of US in Zhan et al. (2021) is **model compatibility**. The incompatibility between a model used within US querying the unlabeled examples (query-oriented model) and a model being evaluated for the tasks (task-oriented model) degrades the performance because the queired examples might not be the most uncertain to the current task-oriented model. Through careful study, we affirm that US maintains a competitive edge over other strategies when used with compatible settings on Logistic Regression (LR), Radial Basis Function kernel Support Vector Machine (RBFSVM), Random Forest (RF), and Gradient Boosting Decision Tree (GBDT). In summary, we recommend adopting US with compatible settings as the first choice for practitioners, providing a clear baseline for AL in real-world usage from the community.

In this work, we make the following contributions:

- To our knowledge, our benchmark is the most comprehensive, surpassing existing benchmarks in terms of datasets, models, query strategies, and analyses.

- We re-benchmark existing strategies for tabular datasets, demonstrating the US's competitiveness on most datasets, and, importantly, uncover profound insights into the often-overlooked issue of **model compatibility** in the context of US.

- We offer a reproducible and open-source benchmarking framework, which includes preparing datasets, an active learning process, and analysis tools to facilitate future research in the community.

## 2 Preliminary

In this section, we extend the Settles (2012)'s literature to the current state of pool-based AL research, addressing the gap created by the lack of an open-source benchmark and highlighting significant developments in query strategies over the last decade. We also introduce the experimental protocol of our benchmark, which facilitates a deeper understanding of the critical components involved in pool-based AL, helping readers to comprehensively evaluate the efficacy of different query strategies in this domain.

### 2.1 Literature survey of pool-based active learning

Settles (2012) formalized the pool-based active learning protocol as follows:

**Initial setup**   The process begins with a small labeled pool $D_l = \{(x_1, y_1), \ldots, (x_N, y_N)\}$, where $x_n \in \mathbb{R}^d$ is $d$-dimension features, $y_n \in \mathcal{Y}$ is label, and $|D_l| = N$ is the number of labeled examples, and a large unlabeled pool $D_u = \{x_{N+1}, \ldots, x_{N+M}\}$, where $|D_u| = M$ is the number of unlabeled examples; and an oracle $O$ that provides ground truth labels.

**Execution setup**   The active learning algorithm operates over $T$ rounds within a total query budget, where each round involves querying the label of one unlabeled example from $D_u$ until the budget is exhausted.

**Query steps in each round**

1. **Query**: Employ the query strategy $\mathcal{Q}$ to select an example $x_j$ from $D_u$.

2. **Label**: Acquire the label $y_j$ for $x_j$ from an oracle $O(x_j) = y_j$.

3. **Update pools**: Move the new labeled example from $D_u$ to $D_l$, i.e., $D_l \leftarrow D_l \cup \{(x_j, y_j)\}$, $D_u \leftarrow D_u \setminus \{(x_j)\}$.

4. **Update the model**: Retrain the model using the updated labeled pool $D_l$.

**Prediction on the test set**   Finally, we train the model $\mathcal{G}$ on the latest labeled pool $D_l$ and make predictions on new examples from the unseen testing set $D_{\text{te}}$.

The critical element in pool-based active learning is the query strategy $\mathcal{Q}$. A naïve uniform sampling (Uniform) method randomly selects unlabeled examples for labeling. Uniform does not utilize active learning strategies and serves primarily as a baseline. The overarching goal of active learning is to develop a query strategy that outperforms the Uniform baseline, and there are already numerous query strategies available today. Based on Settles (2012), we classify existing query strategies into six categories: **model uncertainty**, **expected model changing**, **representation exploiting**, **hybrid criteria**, **Bayesian methods**, and **redesigned Learning Framework**. In the next section, we first introduce different query strategies with an illustrative example, and then we further introduce each type of query strategy and their variants in detail.

### 2.1.1   An illustrative example of types of different methods

In this section, we illustrate the characteristics of six distinct types of query strategies. We denote negative examples with red points, positive examples with blue points, and unlabeled examples with gray points. Figure 1 demonstrates the properties of these methods, highlighting the queried example with a black square box. Given the model's decision boundary, marked with a green dashed line,

- **Model uncertainty** strategy selects the example closest to the decision boundary, reflecting high marginal uncertainty.

- **Expected model changing** chooses an example that, if labeled negative as displayed in Figure 1, would significantly change the current model to the new model displayed in sketched green line.

- **Representative Exploiting** selects the centre of the densest cluster, which does not rely on the current model.

- **Hybrid criteria** balance uncertainty with density by querying uncertain examples in denser regions compared to pure Uncertainty Sampling.

- **Bayesian methods** identify examples within uncertain regions with high posterior variance, as illustrated by the colored areas in Figure 1.

- **Redesigned Learning Framework** selects the most rewarding query strategy and then queries a new example by it.

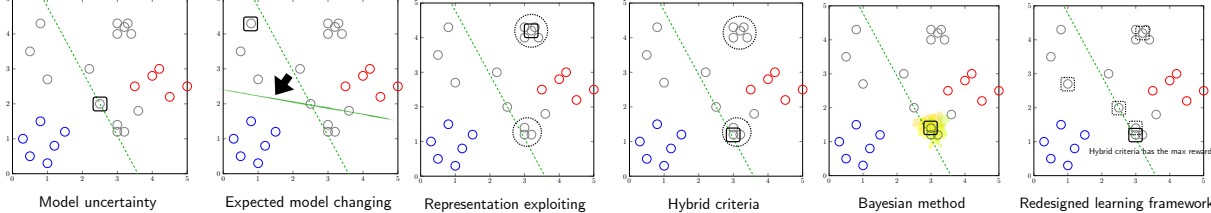

Figure 1: Illustration examples of **model uncertainty**, **expected model changing**, **representation exploiting**, **hybrid criteria**, **Bayesian method**, and **redesigned learning framework**.

Given the high-level idea of different types of query strategies, we begin with **model uncertainty** to guide readers through the relationships and historical development of these query strategies.

**Model uncertainty**   Uncertainty Sampling (US) is a prevalent query strategy in pool-based active learning, where it selects examples for labeling based on the degree of uncertainty regarding the model's prediction. US assumes that examples about which the model is most uncertain are likely to yield the highest information gain upon being labeled. Various measures can be employed to quantify uncertainty, including the margin

score and entropy of the predictions of an examples in the unlabeled pool returned by the current model. In binary classification scenarios, using margin and entropy scores are equivalent in terms of defining model uncertainty (see Appendix B.5). Previous works have found that US is a strong baseline for most pool-based active learning problems (Cawley, 2011; Yang & Loog, 2018; Karamcheti et al., 2021; Schröder et al., 2021; Bahri et al., 2022).

In contrast to US, which relies on a single model to quantify uncertainty, Query By Committee (QBC) (Seung et al., 1992) quantifies uncertainty through multiple models to address the sampling bias in US (Settles, 2012). QBC operates on the principle of disagreement among a committee of models, each representing a different model derived from the training set. Specifically, QBC selects the unlabeled example where there is the maximal disagreement among the committee members. Disagreement is measured by voting entropy, defined as the entropy of the distribution of the committee's votes. A higher voting entropy of an example indicates more significant disagreement and, consequently, a higher value for querying.

**Expected model changing**  Previous query strategies aim to query the most informative example for the current model. In this category, we strive to query the most informative example to reduce the model's error in the future. For instance, Expected Error Reduction (EER) queries the highest expected error of the future output over an unlabeled pool, where the error would be estimated by the Monte-Carlo approach (Roy & McCallum, 2001). Similarly, Variance Reduction (VR) estimates the variance of the model's output based on its Fisher information, which estimates the inverse of the lower bound on the variance of the model's parameters (Cover, 1999; Schein & Ungar, 2007). Another method proposed the difference between the error reduction and the cost of obtaining the label to query the most informative example over an unlabeled pool effectively (Kapoor et al., 2007).

**Representation exploiting**  US and QBC might perform poorly due to outliers or sampling bias that results in querying a non-representative example during the query process (Dasgupta & Hsu, 2008; Yang et al., 2015; Shui et al., 2020). Although EER and VR take the input distribution into account via estimating expected future error over all unlabeled examples, these methods are computationally expensive, making them unsuitable for large datasets (Settles, 2012). In this category, we depart from strategies that rely on model predictions and instead focus on the structure/representation of data, an approach we refer to as *model-free*. Hierarchical Sampling (Hier) is a model-free representation sampling method that exploits hierarchical clustering to explore the data structure of the unlabeled pool (Dasgupta & Hsu, 2008). Hier randomly selects an example from the subtree of the hierarchical clustering tree to obtain its label. Then, the tree structure is iteratively updated by making the labels in the cluster more pure and focusing on the remaining impure clusters.

The query strategies mentioned in Settles (2012) are long-standing. However, the survey should be updated with the latest approaches. Graph Density (Graph) is also a model-free representation sampling method that exploits cluster structure by applying graph-based clustering techniques to the unlabeled pool without depending on any model. Similar to Graph, Core-Set uses K-Means clustering on the embedding space extracted from the data transformation (such as deep convolutional neural networks) and then queries unlabeled examples closest to the centers of clusters. Sener & Savarese (2018) show that Core-Set works well on image classification tasks. Besides Graph and Core-Set, we could categorize recent query strategies into three categories: **hybrid criteria**, **Bayesian method**, and **redesigned learning framework**.

**Hybrid criteria**  Several works study the combination of uncertainty and diversity information to improve previous query strategies. For example, Density-Weight Uncertainty Sampling (DWUS) assumes that informative examples should have both high uncertainty and be representative of the data distribution (Nguyen & Smeulders, 2004), so DWUS designs a weighted uncertainty score by averaging an example's similarity to the remaining examples in the training set. Hinted Support Vector Machine (HintSVM) focused on selecting an example of an updated decision boundary that passes through unqueried regions instead of reducing its margin only (Li et al., 2015). QUerying Informative and Representative Examples (QUIRE) formulated the informativeness and representativeness with kernel matrices (Huang et al., 2010), which characterizes the similarity between labeled examples and unlabeled examples, to select an example with large self-similarity and large similarity to most remaining examples in the unlabeled pool. Representative Marginal Cluster

Mean Sampling (MCM) queries examples within the model's margin closest to the K-Means centers in the embedding space (Xu et al., 2003), which inherits the benefits from Core-Set and US. Recently, Batch Mode Discriminative and Representative (BMDR) and Self-Paced Active Learning (SPAL) have been designed to query a batch of examples with elaborated empirical risk minimization (Wang & Ye, 2015; Tang & Huang, 2019). BMDR queries the example that expects to minimize the empirical risk on the labeled and unlabeled pools using a self-learning approach and distribution difference between the labeled pool and training set. Following the objective function of BMDR, SPAL modifies the constraint of the objective function (1) to improve BMDR's performance. Please refer to Appendix B.5 for the detailed information.

**Bayesian method**  Although QBC aims to query the most disagreeable example, the voter entropy might ignore each model's confidence regarding its predictions, potentially reducing efficiency. To address this issue, Bayesian Active Learning by Disagreement (BALD) queries the most uncertain example across the ensemble models but confident in the single model (Houlsby et al., 2011). This approach can be interpreted as the conditional mutual information between the model's prediction and its parameters. BALD aims to query the example with high conditional mutual information, where the model's prediction is uncertain, but the model's parameters are certain.

**Redesigned learning framework**  As the number of query strategies increases, some are designed to automatically select the optimal strategy from multiple heuristic query strategies. For example, Active Learning By Learning (ALBL) treats the learning problem as a multi-armed bandit problem (Hsu & Lin, 2015). It thus selects the optimal strategy from a set of query strategies and queries the example based on this strategy that maximizes the estimated reward at each round. Learning Active Learning (LAL) formulates the query process as a regression problem to learn the strategy from various types of toy data (Konyushkova et al., 2017). LAL queries the example from the learned regression function, which predicts the potential error reduction.

### 2.1.2  Deep active learning

Besides previous query strategies for conventional machine learning models, such as Logistic Regression (LR), and Radial Basis Function kernel Support Vector Machine (RBFSVM), Beck et al. (2021) and Zhan et al. (2022) compared additional query strategies designed for deep learning models used in computer vision classification tasks. Their results show that US outperforms data diversity-based sampling strategies (Core-Set, Variational Adversarial Active Learning) (Sinha et al., 2019). Moreover, hybrid criteria query strategies, such as Batch Active learning by Diverse Gradient Embeddings (BADGE) (Ash et al., 2019), Learning Loss for Active Learning (LPL) (Yoo & Kweon, 2019), and Wasserstein Adversarial Active Learning (WAAL) (Shui et al., 2020), achieve competitive results better than US. Given that many recent works have taken the burden to study deep active learning on computer vision tasks (Zhan et al., 2022), transformer models (Rauch et al., 2023), and cross-domain scenarios (Werner et al., 2024), we consider "the use of deep active learning in tabular data" still requires a lot of effort to study deeply. Therefore, we focus on providing the practical guide and benchmark of active learning methods for *tabular datasets* in this work.

### 2.2  Experimental protocol for the benchmark

Section 2.1 depicts an abstract process of pool-based active learning. To concretize the experimental protocol for the benchmark, we illustrate the framework in Figure 2. In this framework, we define the training set as the union of the labeled pool and unlabeled pool, denoted as $D_{\text{tr}} = D_{\text{l}} \cup D_{\text{u}}$. First, we split the dataset into disjoint training and testing sets, i.e., $D_{\text{tr}} \cap D_{\text{te}} = \emptyset$, to simulate a real-world learning scenario. After splitting the dataset, we sample from the labeled pool $D_{\text{l}}$ within $D_{\text{tr}}$ and leave the remaining examples as the unlabeled pool $D_{\text{u}}$ to set up the initial environment. Furthermore, we isolate a query-oriented model $\mathcal{H}$ from the task-oriented model $\mathcal{G}$ in Section 2.1. The query-oriented model is used for selecting the most informative example during the query step while the task-oriented model is used for prediction on the test set, as depicted in Figure 2.

To distinguish the relationship between the query-oriented model and the task-oriented model, we define **model compatibility** as the setting where the examples obtained by the query-oriented model might be

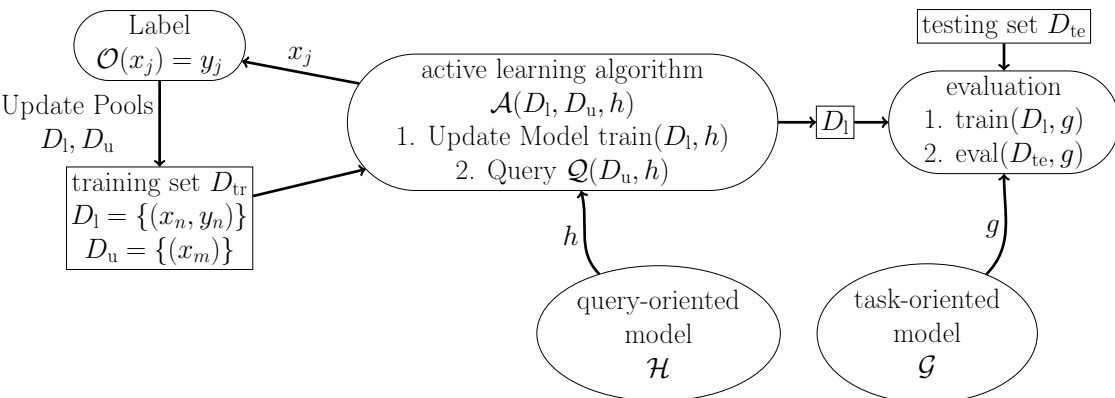

Figure 2: The Framework of Active Learning Experiments. Rectangles represent datasets including labeled pool, unlabeled pool, and test set. Rounded rectangles represent processes including an active learning algorithm, labeling, and evaluation. Circles represent models. In this work, we differentiate the relationship between two models: task-oriented and query-oriented.

different from using the task-oriented model. After introducing the **model compatibility**, we can further discuss the query strategies that depend on the query-oriented models to query new examples. For example, the query-oriented model in HintSVM and QUIRE are restricted to the SVM model due to their theoretical design contrasting *model-free* strategies, which do not rely on the query-oriented model (See Section 2.1).

We notice that the setting of the **model compatibility** is an often-overlooked issue in previous benchmarks when they compared query strategies under the same task-oriented model (Yang & Loog, 2018; Zhan et al., 2021). Although some works discuss the influence of using different models for query informative examples in deep active learning (Yoo & Kweon, 2019; Sinha et al., 2019), it remains unclear in the context of Uncertainty Sampling. In this work, we denote the compatible query-oriented and task-oriented models for Uncertainty Sampling as US-Compatible (US-C) and non-compatible models as US-Non-Compatible (US-NC). Section 5.1 studies the impact of model compatibility on US to clarify the conflicting conclusion in previous benchmarks.

The benchmark aims to provide a standardized framework for evaluating and comparing different query strategies in a fair manner. Following (Guyon et al., 2010; 2011; Desreumaux & Lemaire, 2020; Zhan et al., 2021), we utilize the Area Under the Budget Curve (AUBC) as a summary metric to quantify the results of learning curves. A learning curve tracks the performance of model $\mathcal{G}$ at each round of the active learning process, typically using evaluation metrics such as accuracy. AUBC provides a concise way to compare the overall performance of different learning curves of query strategies. Figure 3 demonstrates that US, BALD, and LAL achieve higher accuracy more quickly than Uniform, corresponding to the mean AUBC of US (85.78%) and BALD (85.72%), which are better than LAL (85.52%), Uniform (84.77%), and Core-Set (84.47%) in detail. Furthermore, we report the accuracy of the task-oriented model under different labeled data sizes and data utilization rates of the query strategy for more detail.

## 3 Experimental settings

We employ most of the settings outlined in the prior benchmark (Zhan et al., 2021). For each dataset $D$, we reserve 40% as the unseen test set $D_{\text{te}}$ for performance evaluation. Then, for the remaining 60%, our default protocol is uniformly sampling few examples as the initial labeled pool $D_{\text{l}}$ and leave the others as the unlabeled pool $D_{\text{u}}$.

$$D = D_{\text{tr}} \cup D_{\text{te}}, \quad |D_{\text{te}}| = 0.4|D|,$$
$$D_{\text{tr}} = D_{\text{l}} \cup D_{\text{u}}, \quad |D_{\text{l}}| = k \times |\mathcal{Y}|,$$

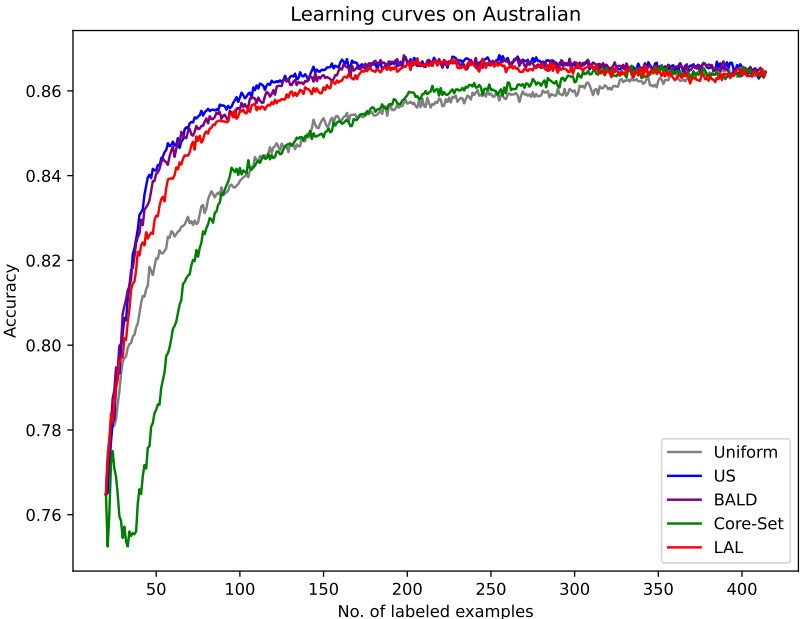

Figure 3: The learning curves (test accuracy vs. number of labeled examples) of query strategies on *Australian* dataset (Chang & Lin, 2011).

where $k$ is the positive number to control the size of the initial labeled pool. In the following, we clarify the differences and expansions in our benchmark compared to the previous benchmarks (Yang & Loog, 2018; Zhan et al., 2021).

**Remove improper use of query strategies** The previous benchmark simultaneously evaluated query strategies that either support or do not support multi-class or batch size greater than one, may have affected the validity of claims when comparing results across different aspects (Zhan et al., 2021). Therefore, we restrict our evaluation only on valid use of query strategies to ensure consistency and fairness.

**Include comprehensive datasets with a unified format.** We select 26 binary datasets from (Zhan et al., 2021) and (Yang & Loog, 2018). Besides, we ensure consistency in the source datasets and the composition of the initial labeled and unlabeled pools. For instance, we scaled raw data features to $[-1, 1]$ for all datasets.[1] We added 4 datasets from the other tabular data benchmark (Grinsztajn et al., 2022) to expand the coverage of the binary classifications. These datasets were selected based on their large-scale, class-imbalanced, and high-dimensional properties to better reflect real-world scenarios. We also extend our benchmark by adding 8 multi-class classification across diverse fields from UCI datasets (Dua & Graff, 2017). Moreover, we extend 2 domain-specific datasets, including utilization of Vision Transformer (ViT) (Dosovitskiy et al., 2021) as a feature extractor for CIFAR10 (Krizhevsky et al., 2009), and BERT (Devlin et al., 2019) for IMDB (Maas et al., 2011), to connect our findings with modern machine learning research. Please refer to Table 17 for the properties of 40 datasets.

**Include broad types of query strategies.** Zhan et al. (2021) extended the query strategies from Yang & Loog (2018) to 17 query strategies. However, the redundancy of query strategies, such as US and Informative Cluster Diverse (InfoDiv) (See Appendix B.5 for more detail.), may lead to repetitive and limited insights into the benchmark. Therefore, we only keep the most representative 12 query strategies: US, QBC, Hier, Graph, Core-Set, HintSVM, QUIRE, DWUS, MCM, BMDR, ALBL, and LAL. We further expand the

---

[1]We retained the original scaling for some of the LIBSVM datasets, such as *Heart*, *Ionosphere*, and *Sonar*, which were already scaled to the range of $[-1, 1]$.

Table 2: Settings of query-oriented models $\mathcal{H}$ for specific query strategies $\mathcal{Q}$.

| $\mathcal{Q}$ | $\mathcal{H}$ | Reason of choice |
|---|---|---|
| HintSVM | RBFSVM | the implementation in libact |
| QUIRE | RBFSVM | the implementation in libact |
| QBC | LR($C$ = 0.1), RBFSVM, RF, Linear Discriminant Analysis | the inheritance of Zhan et al. (2021) |
| ALBL | Combination of multiple $\mathcal{Q}$ with same $\mathcal{H}$: US, HintSVM | the default settings in libact |
| LAL | RF | the implementation in ALiPy |

benchmark to explore a broader range of query strategies by including BALD, a popular query strategy in deep learning (Gal et al., 2017).

**Adopt a tree-based model.** Previous benchmarks studied Logistic Regression and RBFSVM.[2] In this work, we further studied tree-based models such as XGBoost (Chen & Guestrin, 2016) and Random Forest (Breiman, 2001) (See Appendix D), as recommended by the earlier benchmark for tabular datasets (Grinsztajn et al., 2022). To clarify the relationship between query-oriented and task-oriented models, we report some query strategies that do not use tree-based model as the query-oriented model in Table 2.

We disclose the construction of the initial labeled pool, data preprocessing steps, and the choice of models, which can significantly impact the experimental results (Ji et al., 2023), that saves participants time examining the settings and critical considerations for designing active learning experiments. Notably, Ji et al. (2023) recommended using consistent "initial sets" across multiple runs to minimize the impact of randomness and ensure fair comparisons. In our study, randomness arises from two main sources: the train-test split used to derive datasets $D_{\text{tr}}, D_{\text{te}}$ and the construction of initial sets for $D_{\text{l}}, D_{\text{u}}$ (See Section 3.). While randomness from the train-test split is unavoidable without predefined training and test sets, we adhere Ji et al. (2023)'s suggestions by keeping the initial labeled sets fixed to mitigate randomness. A detailed investigation into the effects of randomness due to the train-test split is presented in Appendix C.3. In summary, our findings indicate that Uncertainty Sampling exhibits consistent performance across most datasets, confirming its stability within our benchmark.

**Handle errors and exceptions of experiments.** We report the issues encountered and solutions when we conduct experiments. Because current modules cannot support *cold-start* problems[3], we run the experiments repeatedly and skip any seed that lacks labels in the training or test set at the initial setup. For execution, we set a maximum running time of 72 hours for executing a query strategy on a dataset to ensure completion within a reasonable time (Denote 'TLE' in Table 6).

This section outlines the necessary information to conduct experiments for the benchmark. In our implementation and report, we strive to ensure the reproducibility of all results under these specific settings and processes corresponding to Figure 2. Furthermore, we compare our settings and results with the existing benchmark (Zhan et al., 2021) in Appendix B and Appendix C, covering any additional modifications or improvements needed.

## 4 Benchmarking results

This section presents the benchmarking results for XGBoost in Table 3. We repeated experiments 100 times for small datasets with a size less than 2000 ($K_{\text{S}} = 100$) and 10 times for large datasets ($K_{\text{L}} = 10$). We set a total query budget of 3000 to reduce running time for large datasets. Next, we verify the superiority of Uncertainty Sampling over other query strategies. Furthermore, we investigate whether existing query

---

[2] We also reproduce the previous benchmarks with different base models in Appendix C

[3] The *cold-start* problem is that some classes are under-represented in the initial labeled pool (Yuan et al., 2020; Brangbour et al., 2020).

Table 3: Benchmarking results of XGBoost. The numbers are mean AUBC (↑ is better). We report the baseline method (Uniform), the best query strategy with its mean AUBC (BEST_QS, BEST), and the worst query strategy with its mean AUBC (WORST_QS, WORST).

|  | Uniform | BEST_QS | BEST | WORST_QS | WORST |
|---|---|---|---|---|---|
| Appendicitis | 81.51% | US | 82.85% | DWUS | 80.74% |
| Sonar | 75.10% | US | 76.06% | Core-Set | 74.11% |
| Parkinsons | 84.24% | US | 86.63% | HintSVM | 83.12% |
| Ex8b | 84.21% | LAL | 85.16% | DWUS | 83.05% |
| Heart | 78.37% | BALD | 79.35% | HintSVM | 77.83% |
| Haberman | 67.69% | US | 69.17% | HintSVM | 66.82% |
| Ionosphere | 87.96% | US | 89.95% | DWUS | 81.85% |
| Clean1 | 76.54% | US | 78.99% | Graph | 76.30% |
| Breast | 95.57% | LAL | 96.31% | DWUS | 91.85% |
| Wdbc | 94.07% | LAL | 95.24% | HintSVM | 93.96% |
| Australian | 84.77% | US | 85.78% | HintSVM | 83.89% |
| Diabetes | 72.62% | US | 73.62% | HintSVM | 71.49% |
| Mammographic | 79.46% | BALD | 80.78% | DWUS | 78.80% |
| Ex8a | 92.06% | Core-Set | 94.07% | HintSVM | 84.75% |
| Tic | 90.11% | US | 90.65% | DWUS | 89.11% |
| German | 72.68% | US | 74.03% | DWUS | 71.78% |
| Splice | 91.89% | US | 93.76% | DWUS | 89.51% |
| Gcloudb | 87.85% | LAL | 88.68% | QUIRE | 85.99% |
| Gcloudub | 92.98% | US | 94.30% | DWUS | 86.12% |
| Checkerboard | 98.72% | LAL | 99.49% | DWUS | 86.83% |
| Spambase | 93.16% | US | 94.51% | HintSVM | 91.09% |
| Banana | 87.70% | LAL | 88.45% | HintSVM | 79.70% |
| Phoneme | 85.78% | US | 87.77% | DWUS | 82.42% |
| Ringnorm | 93.76% | US | 95.46% | Core-Set | 64.58% |
| Twonorm | 95.43% | US | 96.39% | HintSVM | 83.38% |
| Phishing | 94.20% | US | 96.24% | DWUS | 91.68% |
| Covertype | 74.11% | US | 76.64% | DWUS | 61.34% |
| Bioresponse | 72.92% | BALD | 74.50% | Core-Set | 72.04% |
| Pol | 96.03% | BALD | 97.62% | HintSVM | 90.52% |

strategies bring more benefits than Uniform for each dataset. After comparing the performance of query strategies in binary classification datasets, we further initiate the experiments for multi-class classification and domain-specific problems to enrich the scope of this work. In addition, we reproduced the benchmarking results from (Zhan et al., 2021) with RBFSVM in Appendix C and constructed the new benchmark for RF in Appendix D.

## 4.1 Verify superiority

Referring to Table 3, we observe that US attains the highest mean AUBC among all query strategies on 18 datasets, indicating its superior performance compared to other query strategies on average. The remaining dominant query strategies are LAL and BALD, which achieve the highest AUBC on 6 and 4 datasets, respectively.

Besides AUBC, we also observe learning curves from different perspectives. Specifically, we check the model's accuracy with varying ratios of labeled examples on each dataset. Table 4 shows the model's accuracy with 20% labeled examples on each dataset, and US outperforms other query strategies on more than half (15) datasets. Please refer to Appendix A for more comparisons under other ratios. Beyond using a fixed query

Table 4: Accuracy (↑ is better) of the model with 20% labeled examples: We report the model's accuracy with 20% labeled examples on each dataset. The scores with **bold** indicate the best performance, and with *italics* indicate the second-place performance on a dataset. 'TLE' means a query strategy exceeds the time limit.

| | Uniform | US | QBC | BALD | Hier | Graph | Core-Set | HintSVM | QUIRE | DWUS | MCM | BMDR | ALBL | LAL |
|---|---|---|---|---|---|---|---|---|---|---|---|---|---|---|
| Sonar | 67.82% | 67.29% | 66.95% | 66.73% | 66.45% | 67.83% | 66.25% | 65.88% | 67.99% | *68.15%* | 66.82% | **68.56%** | 67.25% | 67.32% |
| Parkinsons | 79.26% | **80.71%** | 79.36% | *80.51%* | 79.40% | 78.90% | 78.68% | 78.41% | 78.60% | 79.69% | 80.10% | 78.81% | 79.91% | 80.27% |
| Ex8b | 80.55% | 80.69% | 79.98% | 79.37% | 80.24% | 79.13% | 81.11% | 80.32% | **81.26%** | 78.76% | 80.45% | *81.26%* | 80.74% | 80.60% |
| Heart | 75.73% | *77.19%* | 75.06% | **77.36%** | 75.59% | 76.31% | 76.69% | 74.82% | 76.36% | 75.09% | 75.41% | 75.91% | 76.36% | 76.39% |
| Haberman | 69.24% | *70.61%* | 69.20% | 70.15% | 68.54% | 68.96% | 67.68% | 68.43% | 67.69% | 68.51% | **71.04%** | 69.15% | 68.67% | 70.50% |
| Ionosphere | 83.59% | *86.96%* | 83.84% | 86.78% | 84.18% | 82.28% | 83.24% | 81.84% | 80.82% | 74.91% | 84.02% | 80.28% | 86.73% | **87.21%** |
| Clean1 | 68.34% | **69.86%** | 68.03% | 69.10% | 68.57% | 68.21% | 66.74% | 67.59% | 68.42% | 68.34% | 67.36% | 66.42% | *69.35%* | 69.31% |
| Breast | 95.17% | **96.73%** | 95.17% | *96.72%* | 95.54% | 95.29% | 94.54% | 94.77% | 94.93% | 90.07% | 96.57% | 94.77% | 96.16% | 96.66% |
| Wdbc | 92.91% | 95.34% | 92.50% | *95.36%* | 92.91% | 92.99% | 93.04% | 91.94% | 92.66% | 92.54% | 95.26% | 92.68% | 94.75% | **95.55%** |
| Australian | 83.63% | **85.55%** | 83.77% | *85.33%* | 83.87% | 83.90% | 82.97% | 81.95% | 82.75% | 82.51% | 84.86% | 83.58% | 83.28% | 85.13% |
| Diabetes | 71.84% | **73.96%** | 72.34% | *73.35%* | 72.07% | 72.38% | 71.73% | 70.14% | 71.81% | 70.54% | 72.85% | 72.16% | 71.78% | 72.80% |
| Mammographic | 79.66% | **82.51%** | 79.66% | *82.30%* | 79.48% | 79.17% | 79.52% | 78.70% | 80.41% | 79.39% | 82.09% | 80.03% | 80.33% | 81.67% |
| Ex8a | 88.22% | 88.55% | 87.81% | 88.75% | 87.94% | 89.41% | *91.69%* | 78.04% | 78.12% | 78.06% | 88.41% | 89.22% | 85.21% | **91.88%** |
| Tic | 89.25% | **90.43%** | 89.23% | *90.38%* | 89.30% | 89.72% | 89.33% | 88.12% | 89.83% | 85.76% | 89.58% | TLE% | 89.32% | 89.15% |
| German | 71.23% | **72.90%** | 71.37% | *72.73%* | 71.08% | 71.48% | 70.46% | 71.61% | 71.06% | 68.76% | 71.97% | TLE% | 71.55% | 72.11% |
| Splice | 89.07% | **92.95%** | 88.88% | *92.65%* | 89.00% | 89.25% | 84.99% | 85.74% | 88.99% | 84.69% | 91.33% | 89.04% | 88.65% | 90.25% |
| Gcloudb | 87.82% | **89.55%** | 87.98% | 89.24% | 87.95% | 88.19% | 88.27% | 84.62% | 84.61% | 85.48% | *89.49%* | 87.96% | 88.35% | 89.36% |
| Gcloudub | 91.25% | **94.11%** | 91.45% | 92.40% | 91.92% | 92.28% | 88.58% | 83.11% | 85.69% | 80.24% | 91.69% | 89.76% | 88.91% | *93.61%* |
| Checkerboard | 98.76% | 96.89% | 98.80% | *99.46%* | 99.35% | 98.95% | 98.59% | 91.40% | 88.72% | 79.82% | 99.46% | 99.09% | 98.26% | **99.80%** |
| Spambase | 92.47% | *94.84%* | 92.69% | **94.91%** | 92.54% | 92.64% | 92.06% | 88.95% | TLE% | 92.54% | 94.61% | TLE% | 92.66% | 94.68% |
| BaTLEa | 87.46% | 87.93% | 87.46% | 87.65% | 87.53% | 87.50% | 88.02% | 71.37% | 76.11% | 74.94% | *88.49%* | TLE% | 86.99% | **88.94%** |
| Phoneme | 83.97% | *87.13%* | 83.73% | **87.24%** | 84.66% | 84.13% | 84.70% | 80.83% | TLE% | 78.60% | 86.36% | TLE% | 83.73% | 86.52% |
| Ringnorm | 92.79% | *95.75%* | 93.33% | **95.77%** | 92.35% | 92.40% | 51.86% | 57.27% | TLE% | 55.71% | 95.33% | TLE% | 92.45% | 92.55% |
| Twonorm | 94.99% | **96.61%** | 95.04% | 96.53% | 95.07% | 95.24% | 95.73% | 79.31% | TLE% | 94.26% | *96.60%* | TLE% | 95.52% | 95.90% |
| Phishing | 93.41% | **96.19%** | 93.01% | *96.04%* | 92.83% | 93.35% | 93.18% | 91.75% | TLE% | 88.11% | 95.70% | TLE% | 94.37% | 95.94% |
| Covertype | 72.30% | **75.26%** | 72.05% | *75.26%* | TLE% | 64.54% | TLE% | 62.90% | TLE% | 59.97% | 71.54% | TLE% | 69.30% | 73.92% |
| Bioresponse | 69.96% | **73.14%** | 71.05% | *72.87%* | 70.48% | 71.11% | 66.59% | 66.70% | TLE% | 69.96% | 71.34% | TLE% | 70.33% | 72.07% |
| Pol | 95.29% | **98.19%** | 95.52% | *98.10%* | 95.40% | 86.38% | 93.68% | 86.75% | TLE% | 95.29% | 97.58% | TLE% | 95.18% | 97.75% |

budget in Table 4, we also check the metric of the number of queried examples required to reach 99% of the model's performance trained on the full budget. Table 5 demonstrates that Uncertainty Sampling stably achieves first place or second place on 21 datasets, which shows a consistent conclusion with Table 3 and Table 4. More results are revealed in Appendix A.

Finally, we verify the ranking performance of query strategies across multiple datasets. Specifically, we assess the average and standard deviation of the rankings by seeds of the query strategy on each dataset. Then, we apply the Friedman test with a 5% significance level to test for statistical significance. The p-values of the Friedman test are less than 5% for all datasets, indicating that the performance differences between query strategies are statistically significant. Table 6 demonstrates that US ranks first on 18 datasets, and LAL, BALD, and MCM often achieve second and third ranks.

These results show that the straightforward and efficient US outperforms others on most datasets. These outcomes also correspond to previous work claiming US is the strong baseline with LR (Yang & Loog, 2018) and RBFSVM, which we re-benchmarked in Appendix C. We recommend that practitioners initiate their pool-based active learning projects with US.

## 4.2 Verify usefulness

We investigate the *usefulness* of query strategies in Section 4.2. The analysis of *usefulness* can uncover which query strategy brings more benefits than Uniform, offering practitioners a reality check on the effectiveness of a query strategy. Specifically, we investigate the improvement of the optimal stopping point of query strategies over Uniform. The optimal stopping point is the point where the model achieves the target accuracy with the least number of labeled examples. We refer to the *data utilization rate* (Culver et al., 2006), which is the number of labeled examples to achieve the target accuracy divided by the number of labeled examples required by Uniform. In this benchmark, we set the target accuracy as the accuracy with the total query budget minus 0.01. Table 7 shows the data utilization rate of the optimal stopping point of query strategies over Uniform. We observe that US, BALD, MCM, and LAL achieve a higher data utilization rate than Uniform on most datasets.

Table 5: The minimum number of queried examples required to reach 99% accuracy (↓ is better) of the model: The **bold** indicates the first place and *italics* indicates the second place. 'TLE' means a query strategy exceeds the time limit.

| | Uniform | US | QBC | BALD | Hier | Graph | Core-Set | HintSVM | QUIRE | DWUS | MCM | BMDR | ALBL | LAL |
|---|---|---|---|---|---|---|---|---|---|---|---|---|---|---|
| Appendicitis | 28.44 | 25.58 | 29.52 | 25.06 | 28.59 | 27.50 | 26.03 | 25.63 | 26.92 | 31.49 | **24.71** | 28.68 | 25.43 | *24.74* |
| Sonar | 73.45 | 65.70 | 68.14 | *65.21* | 72.94 | 75.91 | 77.40 | 73.36 | 75.83 | 75.70 | 70.33 | 71.05 | 67.15 | **63.78** |
| Parkinsons | 66.44 | **44.56** | 66.81 | 47.26 | 58.10 | 75.95 | 65.68 | 81.31 | 70.40 | 67.83 | 49.78 | 60.99 | 59.36 | *45.41* |
| Ex8b | 51.02 | **42.87** | 54.42 | 46.34 | 45.86 | 55.63 | *43.27* | 58.98 | 52.16 | 66.05 | 43.28 | 49.59 | 43.56 | 45.09 |
| Heart | 47.75 | *41.11* | 48.64 | **40.14** | 51.24 | 45.84 | 48.88 | 52.44 | 46.48 | 51.28 | 44.87 | 45.48 | 45.19 | 43.35 |
| Haberman | 25.53 | **24.59** | 28.73 | 25.35 | 27.02 | 27.17 | 29.75 | 30.51 | 29.40 | 28.74 | 26.36 | 26.81 | 28.67 | *25.12* |
| Ionosphere | 97.04 | **53.38** | 98.79 | *55.72* | 91.89 | 95.68 | 88.46 | 104.41 | 96.79 | 179.88 | 66.03 | 120.41 | 70.33 | 57.49 |
| Clean1 | 207.79 | **152.39** | 207.93 | *155.50* | 201.93 | 211.52 | 191.98 | 198.64 | 196.44 | 207.79 | 166.59 | 199.24 | 180.07 | 166.30 |
| Breast | 70.81 | 38.78 | 65.98 | *38.02* | 56.25 | 90.22 | 61.85 | 64.05 | 60.09 | 288.68 | 40.14 | 81.19 | 39.19 | **30.39** |
| Wdbc | 102.53 | 48.87 | 111.69 | *48.00* | 89.32 | 106.51 | 88.53 | 110.64 | 92.47 | 110.06 | 51.92 | 100.71 | 49.67 | **44.84** |
| Australian | 88.71 | **55.39** | 79.43 | *59.18* | 86.36 | 92.02 | 101.76 | 119.76 | 96.14 | 104.74 | 66.84 | 96.08 | 77.46 | 64.00 |
| Diabetes | 56.67 | *49.28* | 57.50 | 54.89 | 53.78 | 58.07 | 59.12 | 104.25 | 62.42 | 88.29 | 57.51 | 56.83 | 56.63 | **44.80** |
| Mammographic | 30.29 | 28.73 | 39.66 | **27.09** | 42.09 | 39.69 | 37.89 | 35.29 | 31.48 | 50.08 | 30.52 | 31.67 | 28.13 | *27.10* |
| Ex8a | 263.54 | 191.79 | 250.31 | 193.59 | 241.43 | 261.52 | **154.35** | 379.03 | 363.91 | 312.23 | 188.39 | 204.05 | 367.55 | *168.98* |
| Tic | 144.75 | *90.08* | 148.10 | **89.12** | 137.06 | 151.89 | 194.89 | 172.11 | 132.64 | 185.20 | 119.46 | TLE | 166.50 | 132.92 |
| German | 178.81 | **124.03** | 163.55 | 138.11 | 184.64 | 177.15 | 177.90 | 164.53 | 169.58 | 256.71 | 144.83 | TLE | 146.80 | *129.60* |
| Splice | 310.57 | **134.00** | 305.19 | *140.50* | 307.94 | 285.83 | 264.18 | 321.10 | 290.00 | 444.19 | 158.67 | 302.19 | 274.96 | 214.72 |
| Gcloudb | 56.49 | *36.83* | 63.12 | 38.09 | 48.29 | 71.45 | 42.10 | 116.48 | 158.24 | 73.58 | 37.54 | 45.44 | 45.08 | **34.30** |
| Gcloudub | 239.57 | *124.60* | 228.82 | 140.76 | 194.34 | 252.24 | 309.11 | 510.70 | 355.73 | 512.64 | 150.74 | 270.05 | 206.95 | **119.19** |
| Checkerboard | 140.40 | 233.72 | 146.03 | 91.92 | 100.84 | 121.44 | 159.23 | 528.25 | 529.48 | 619.52 | *72.11* | 107.50 | 151.65 | **37.22** |
| Spambase | 941.80 | **221.00** | 865.50 | *247.30* | 780.30 | 1090.40 | 991.00 | 1670.10 | TLE | 965.40 | 322.60 | TLE | 734.70 | 324.20 |
| BaTLEa | 540.70 | 496.30 | 450.10 | 537.10 | 509.10 | 579.50 | *319.20* | 2081.00 | 1593.80 | 2618.60 | 489.60 | TLE | 729.50 | **221.90** |
| Phoneme | 1742.80 | **611.30** | 1665.60 | *622.70* | 1248.10 | 1750.50 | 1332.40 | 1792.80 | TLE | 2690.90 | 750.70 | TLE | 1501.40 | 738.90 |
| Ringnorm | 1317.10 | **386.90** | 1150.40 | *410.00* | 1400.60 | 1332.20 | 2564.60 | 1959.10 | TLE | 2066.90 | 546.80 | TLE | 1189.30 | 922.80 |
| Twonorm | 525.50 | **179.40** | 622.30 | *189.10* | 574.90 | 531.10 | 398.20 | 2638.30 | TLE | 921.80 | 222.00 | TLE | 382.50 | 375.20 |
| Phishing | 1080.50 | 282.60 | 1366.60 | *273.50* | 1244.70 | 1396.00 | 1151.10 | **20.00** | TLE | 2500.00 | 374.50 | TLE | 578.00 | 322.90 |
| Covertype | 1981.60 | 796.50 | 2115.80 | 842.30 | TLE | **20.00** | TLE | TLE | TLE | *20.00* | TLE | TLE | TLE | 1008.00 |
| Bioresponse | 1182.30 | **771.00** | 1262.50 | 793.50 | 1112.20 | 1160.80 | 1068.80 | TLE | TLE | 1182.30 | 908.50 | TLE | TLE | *777.40* |
| Pol | 1001.00 | *288.90* | 938.70 | **272.40** | 930.90 | 955.60 | 1501.00 | 300.00 | TLE | 1001.00 | 404.30 | TLE | 1236.60 | 312.20 |

To further investigate the usefulness of US, we check the improved accuracy ($\tau$) of US, BALD, Core-Set, and LAL over Uniform on effective dataset (*Covertype*) and ineffective dataset (*Checkerboard*) on average with different scales of the total budget. Figure 4 shows that the performance of US and BALD gains significant benefits on large scale dataset. However, US suffers from the sampling bias on *Checkerboard* with a small budget, while BALD is more stable. We notice that a query strategy with good performance brings more benefits at the early stage of the learning process.

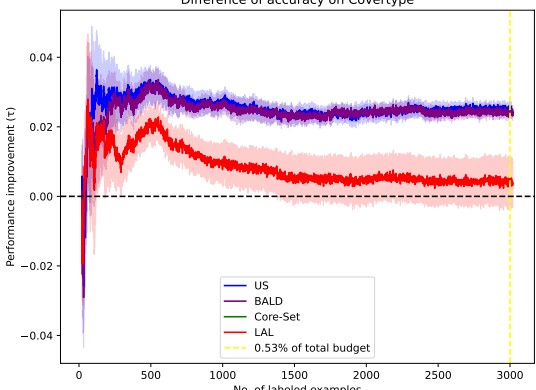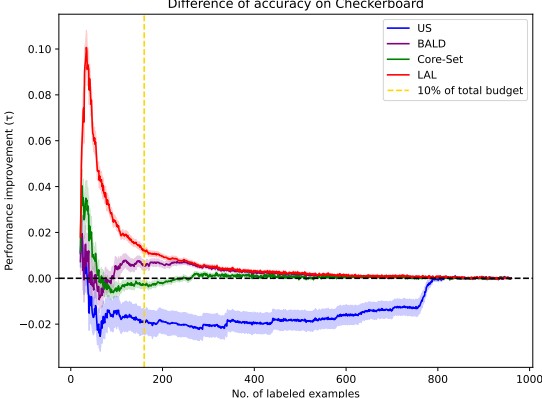

Figure 4: Mean difference of accuracy (improvement) of a query strategy from Uniform on *Covertype* (left) and *Checkerboard* (right). Note that there are no results of Core-Set on *Covertype* due to the time limit (TLE).

Table 6: Average Ranking of Query Strategies (↓ is better): We report query strategies with the best average ranking. The scores with [1], [2], or [3] mean the 1st, 2nd and 3rd performance on a dataset. 'TLE' means a query strategy exceeds the time limit.

| | US | QBC | BALD | Hier | Graph | Core-Set | HintSVM | QUIRE | DWUS | MCM | BMDR | ALBL | LAL |
|---|---|---|---|---|---|---|---|---|---|---|---|---|---|
| Appendicitis | **4.83**[1] | 7.50 | 5.51[3] | 7.63 | 8.30 | 7.15 | 7.34 | 7.82 | 9.12 | *5.20*[2] | 7.85 | 6.92 | 5.83 |
| Sonar | **4.79**[1] | 6.38 | *4.97*[2] | 6.83 | 7.52 | 8.35 | 8.04 | 7.79 | 6.71 | 6.19 | TLE | 5.39 | 5.04[3] |
| Parkinsons | **3.39**[1] | 8.41 | *3.79*[2] | 7.03 | 9.45 | 8.37 | 10.92 | 8.47 | 8.74 | 4.59 | 7.26 | 6.34 | 4.24[3] |
| Ex8b | *5.31*[2] | 7.95 | 6.34 | 7.60 | 8.28 | 5.83 | 9.09 | 7.42 | 10.07 | **5.08**[1] | 7.13 | 5.50 | 5.40[3] |
| Heart | *4.89*[2] | 7.72 | **4.76**[1] | 7.66 | 8.17 | 7.18 | 9.65 | 8.34 | 6.43 | 6.33 | 7.10 | 7.04 | 5.73[3] |
| Haberman | **4.38**[1] | 7.69 | *4.56*[2] | 7.04 | 8.10 | 8.97 | 9.76 | 9.55 | 6.07 | 4.74[3] | 7.07 | 7.95 | 5.12 |
| Ionosphere | **2.67**[1] | 7.19 | *2.89*[2] | 7.12 | 8.22 | 8.28 | 9.38 | 10.10 | 12.74 | 4.51 | 10.08 | 4.30 | 3.52[3] |
| Clean1 | **2.61**[1] | 8.13 | *2.96*[2] | 8.33 | 8.76 | 7.87 | 8.47 | 7.39 | 8.59 | 5.30 | TLE | 5.53 | 4.06[3] |
| Breast | 3.98[3] | 8.72 | *3.72*[2] | 7.14 | 10.02 | 7.43 | 8.88 | 7.50 | 12.97 | 4.24 | 8.64 | 4.88 | **2.88**[1] |
| Wdbc | 3.67[3] | 9.47 | *3.44*[2] | 8.15 | 9.35 | 8.41 | 9.57 | 9.02 | 9.54 | 3.95 | 9.59 | 3.71 | **3.13**[1] |
| Australian | **2.80**[1] | 7.75 | *3.20*[2] | 7.70 | 7.47 | 8.97 | 10.38 | 8.36 | 8.65 | 4.66 | 8.30 | 8.32 | 4.44[3] |
| Diabetes | **3.64**[1] | 7.15 | *4.35*[2] | 6.88 | 7.09 | 7.17 | 10.44 | 8.44 | 10.01 | 4.98[3] | 7.20 | 8.14 | 5.51 |
| Mammographic | 3.46[3] | 8.69 | **3.30**[1] | 7.67 | 8.07 | 9.09 | 8.24 | 9.06 | 9.54 | *3.41*[2] | 7.51 | 9.07 | 3.89 |
| Ex8a | 4.37[3] | 6.10 | 4.42 | 5.90 | 6.16 | **1.70**[1] | 11.68 | 10.75 | 10.24 | 4.43 | TLE | 8.94 | *3.31*[2] |
| Tic | **2.65**[1] | 5.66 | *2.99*[2] | 6.47 | 6.84 | 8.49 | 9.32 | 8.31 | 9.24 | 4.59[3] | TLE | 7.28 | 6.16 |
| German | **3.10**[1] | 7.29 | *3.52*[2] | 7.81 | 7.23 | 7.41 | 6.87 | 8.24 | 10.92 | 4.14[3] | TLE | 6.02 | 5.45 |
| Splice | **1.52**[1] | 7.30 | *1.86*[2] | 7.18 | 8.61 | 9.29 | 9.82 | 6.40 | 11.62 | 3.37[3] | TLE | 6.62 | 4.41 |
| Gcloudb | 4.24[3] | 7.25 | 4.69 | 7.51 | 8.19 | 5.93 | 10.71 | 11.04 | 11.67 | *4.02*[2] | 7.07 | 5.56 | 3.12[1] |
| Gcloudub | **2.52**[1] | 6.41 | 3.68[3] | 4.91 | 7.13 | 8.75 | 12.44 | 10.70 | 12.45 | 4.51 | 7.60 | 7.05 | *2.85*[2] |
| Checkerboard | 6.37 | 7.15 | 4.94 | 5.03 | 7.44 | 6.54 | 11.36 | 11.48 | 12.81 | *3.72*[2] | 4.58[3] | 8.34 | **1.24**[1] |
| Spambase | **1.50**[1] | 7.80 | *1.70*[2] | 6.80 | 8.10 | 8.30 | 11.00 | TLE | 7.80 | 3.30[3] | TLE | 6.20 | 3.50 |
| Banana | 5.20 | 5.70 | 5.70 | 3.60[3] | 8.20 | *3.00*[2] | 10.60 | TLE | 10.40 | 5.00 | TLE | 7.20 | **1.40**[1] |
| Phoneme | **1.60**[1] | 8.10 | *2.00*[2] | 5.20 | 8.40 | 6.40 | 10.00 | TLE | 10.90 | 3.30 | TLE | 7.00 | 3.10[3] |
| Ringnorm | **1.40**[1] | 5.10 | *1.60*[2] | 6.30 | 8.00 | 10.50 | 9.00 | TLE | 10.50 | 3.00[3] | TLE | 6.30 | 4.30 |
| Twonorm | **1.30**[1] | 7.90 | *1.70*[2] | 8.50 | 7.40 | 6.00 | 11.00 | TLE | 10.00 | 3.00[3] | TLE | 5.20 | 4.00 |
| Phishing | **1.40**[1] | 7.90 | *1.60*[2] | 7.20 | 8.70 | 6.20 | 10.10 | TLE | 10.90 | 3.60 | TLE | 5.00 | 3.40[3] |
| Covertype | **1.40**[1] | 3.80 | *1.90*[2] | TLE | 5.00 | TLE | TLE | TLE | 6.00 | TLE | TLE | TLE | 2.90[3] |
| Bioresponse | *1.90*[2] | 6.40 | **1.60**[1] | 6.40 | 6.20 | 8.60 | TLE | TLE | 6.60 | 3.70 | TLE | TLE | 3.60[3] |
| Pol | *1.80*[2] | 5.80 | **1.50**[1] | 6.20 | 9.80 | 9.20 | 11.00 | TLE | 6.70 | 4.00 | TLE | 7.30 | 2.70[3] |

Tables 4–7 demonstrate that the *Checkerboard* and *Banana* datasets pose challenges for Uncertainty Sampling. Both are synthetic two-dimensional datasets introduced in previous works (Alcala-Fdez et al., 2010; Konyushkova et al., 2017). To study the cause of the Uncertainty Sampling's failure, we visualize scatter plots at selected rounds throughout the active learning process. Specifically, we examine the size of labeled pool at 20, 200, and 800 for *Checkerboard*, while 20, 400, and 800 for *Banana*. These rounds correspond to the initial labeled pool, the round at which Uncertainty Sampling performs worse than Uniform, and the round at which Uncertainty Sampling achieves comparable performance to Uniform, as observed in the learning curves for each dataset presented in Figure 5.

Figures 6 (*Checkerboard*) and 7 (*Banana*) illustrate existing **unexplored regions** either at the initial or during intermediate rounds. Our analysis reveals that when datasets have multiple overlapping positive and negative regions, Uncertainty Sampling tends to query examples from these overlapping regions rather than exploring less-covered regions, particularly due to the uneven distribution of the initial labeled pool. In this paragraph, we analyze the possible reasons for the failure of US. Other related works investigating the causes of US failure is discussed in Section 5.4.

## 4.3 Expanded benchmarks: multi-class classifications and domain-specific data

### 4.3.1 Multi-class classification datasets

We extend the evaluation to include 7[4] multi-class classification problems to demonstrate the broad scope of the benchmark and improve the validity of the US competitive edge. The multi-class datasets cover several fields of applications such as biology, physics, climate, business, healthcare, and social science, which reveals

---

[4]We exclude *RT-IoT2022* for this experiment because the highly imbalanced ratio of the dataset results in lacking classes for the training set when initializing the labeled pool by the default protocol (See Section 3).

Table 7: Data utilization rate (↓ is better): We report the data utilization rate of query strategies. The **bold** indicates the first place and *italics* indicates the second place. The scores with pink color indicate that the query strategy does not provide more benefits than Uniform. 'TLE' means a query strategy exceeds the time limit.

| | US | QBC | BALD | Hier | Graph | Core-Set | HintSVM | QUIRE | DWUS | MCM | BMDR | ALBL | LAL |
|---|---|---|---|---|---|---|---|---|---|---|---|---|---|
| Appendicitis | 77.96% | 93.97% | **71.10%** | 88.47% | 84.87% | 77.22% | 73.31% | 81.74% | 101.34% | 73.73% | 88.66% | 72.73% | *71.81%* |
| Sonar | 94.66% | 98.91% | **92.74%** | 103.50% | 109.41% | 115.15% | 107.62% | 110.06% | 106.54% | 103.24% | 103.20% | 94.73% | *93.96%* |
| Parkinsons | **65.14%** | 100.21% | 70.28% | 86.74% | 116.07% | 99.84% | 123.64% | 103.34% | 101.74% | 74.59% | 92.52% | 90.79% | *67.29%* |
| Ex8b | **90.08%** | 109.88% | 93.86% | 95.59% | 107.28% | 91.14% | 126.40% | 104.15% | 129.39% | *90.31%* | 103.05% | 90.49% | 93.70% |
| Heart | *86.90%* | 107.95% | **84.11%** | 107.17% | 97.55% | 103.64% | 113.07% | 103.62% | 109.48% | 97.06% | 95.77% | 91.87% | 89.85% |
| Haberman | **82.19%** | 95.22% | *82.67%* | 111.45% | 105.52% | 129.80% | 161.28% | 141.81% | 107.57% | 92.66% | 104.88% | 118.27% | 100.55% |
| Ionosphere | **61.01%** | 111.80% | *63.87%* | 105.88% | 108.81% | 96.76% | 119.75% | 111.23% | 209.02% | 76.15% | 139.82% | 78.44% | 65.46% |
| Clean1 | **75.75%** | 103.42% | *76.96%* | 99.94% | 104.20% | 94.65% | 97.48% | 95.94% | 100.00% | 82.71% | 99.69% | 88.50% | 82.04% |
| Breast | 69.35% | 109.64% | *66.14%* | 89.34% | 147.47% | 95.73% | 100.94% | 95.69% | 494.88% | 71.37% | 133.41% | 66.72% | **49.26%** |
| Wdbc | 64.80% | 133.42% | *62.73%* | 108.66% | 131.75% | 109.41% | 138.49% | 116.93% | 117.16% | 68.42% | 119.39% | 64.54% | **59.28%** |
| Australian | **77.16%** | 115.07% | *79.27%* | 107.96% | 128.48% | 144.82% | 159.60% | 127.39% | 130.52% | 92.17% | 128.82% | 111.12% | 85.45% |
| Diabetes | 93.03% | 102.01% | 107.05% | *89.91%* | 114.95% | 108.80% | 181.30% | 112.29% | 171.75% | 94.00% | 105.48% | 101.62% | **76.47%** |
| Mammographic | 92.07% | 120.26% | **78.18%** | 124.13% | 128.84% | 112.22% | 237.91% | 103.86% | 202.17% | 88.01% | 119.19% | 82.66% | *79.15%* |
| Ex8a | 80.82% | 101.76% | 82.20% | 97.18% | 110.12% | **62.62%** | 164.20% | 158.36% | 134.14% | 79.50% | 84.62% | 153.98% | *69.71%* |
| Tic | *80.79%* | 128.22% | **79.62%** | 118.52% | 137.88% | 151.43% | 147.20% | 111.52% | 164.54% | 105.34% | TLE | 137.67% | 118.13% |
| German | *99.67%* | 119.68% | 103.79% | 125.82% | 122.31% | 139.69% | 114.99% | 124.65% | 190.73% | 113.27% | TLE | 113.41% | **95.44%** |
| Splice | **50.12%** | 109.42% | *52.06%* | 110.84% | 101.56% | 96.64% | 116.59% | 97.02% | 165.10% | 58.39% | 105.95% | 99.16% | 77.40% |
| Gcloudb | 73.44% | 124.36% | *71.91%* | 103.12% | 145.65% | 78.12% | 222.56% | 289.43% | 152.25% | 72.98% | 85.25% | 72.23% | **64.86%** |
| Gcloudub | *67.80%* | 118.36% | 79.22% | 102.15% | 131.47% | 168.09% | 310.10% | 198.38% | 303.37% | 87.16% | 156.37% | 117.76% | **66.35%** |
| Checkerboard | 231.82% | 140.07% | 93.83% | 110.34% | 115.64% | 154.47% | 529.48% | 543.18% | 660.77% | *72.37%* | 105.94% | 153.28% | **39.50%** |
| Spambase | **25.23%** | 97.59% | *28.02%* | 87.65% | 121.97% | 106.69% | 196.13% | TLE | 101.91% | 37.69% | TLE | 82.08% | 37.18% |
| Banana | 111.31% | 95.73% | 121.28% | 106.59% | 117.47% | *65.76%* | 448.34% | 393.05% | 574.50% | 103.93% | TLE | 132.50% | **43.66%** |
| Phoneme | **35.59%** | 97.05% | *36.87%* | 73.34% | 103.54% | 80.84% | 107.40% | TLE | 165.20% | 43.68% | TLE | 90.82% | 44.67% |
| Ringnorm | **31.06%** | 91.33% | *32.57%* | 111.24% | 102.61% | 208.70% | 158.46% | TLE | 195.48% | 44.30% | TLE | 94.07% | 73.23% |
| Twonorm | **34.59%** | 115.91% | *36.78%* | 112.05% | 103.37% | 75.10% | 529.37% | TLE | 173.40% | 43.62% | TLE | 73.05% | 74.54% |
| Phishing | **27.08%** | 131.84% | *28.23%* | 117.78% | 137.85% | 109.86% | 57.30% | TLE | 244.87% | 37.25% | TLE | 55.39% | 31.95% |
| Covertype | *41.33%* | 109.86% | **40.35%** | TLE | 116.98% | TLE | TLE | TLE | 117.86% | TLE | TLE | TLE | 46.23% |
| Bioresponse | *64.95%* | 105.42% | **64.58%** | 89.29% | 96.76% | 91.78% | TLE | TLE | 100.00% | 73.95% | TLE | TLE | 66.72% |
| Pol | *29.38%* | 96.56% | **27.82%** | 96.41% | 98.11% | 152.58% | 34.08% | TLE | 100.00% | 41.06% | TLE | 126.70% | 31.84% |

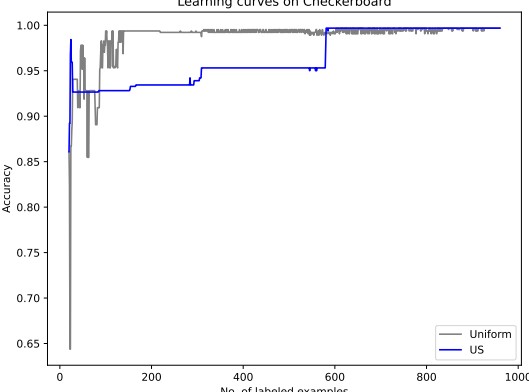
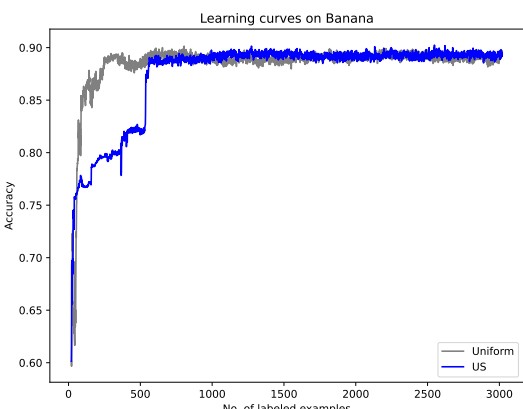

Figure 5: The learning curves (test accuracy vs. number of labeled examples) of query strategies on *Checkerboard* (left) and *Banana* (right).

the value of this benchmark for real-world applications (See Table 17 for details). As an initiating demo, we only adopt the query strategies that are valid for the multi-class classifications and representative of each category of query strategy described in Section 2.1. Therefore, we choose US, BALD, MCM, and Core-Set. In particular, the different uncertainty measures in US behave differently for multi-class classifications (Settles, 2012), so we compare the least confidence (US-LC), the smallest margin (US-SM), and the maximum entropy (US-ME) to verify their performance.

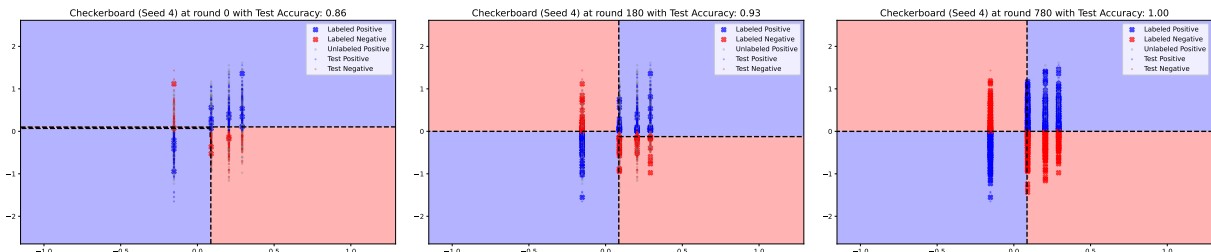

Figure 6: The scatter plots of *Checkerboard* at 20 (left), 200 (middle), and 800 (right) labeled examples. We denote red points as negative examples, blue points as positive examples, and gray points as unlabeled examples. We mark an example with a **cross** in labeled pool $D_l$ and others with **dot**s.

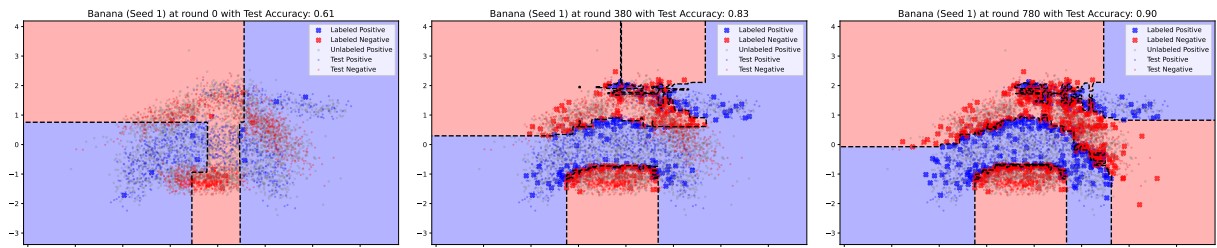

Figure 7: The scatter plots of *Banana* at 20 (left), 200 (middle), and 800 (right) labeled examples. The format is the same as the Figure 6.

Table 8 demonstrates the superiority of uncertainty sampling with margins (US-SM) performing well in multi-class classifications. Although US-SM does not achieve first place in *Iris* and *Wine*, the difference between all query strategies, including Uniform, is insignificant in these datasets. In this experiment, we verify that US-SM with compatible XGBoost models would significantly improve the model's performance for the multi-class classifications.

### 4.3.2 Domain-specific datasets

Our benchmark also includes domain-specific datasets. One is *CIFAR-10* (Krizhevsky et al., 2009), which belongs to the computer vision (CV) domain; the other is *IMDB* (Maas et al., 2011), belonging to the natural language process (NLP) domain. We incorporate deep learning models, such as the ViT feature extractor for *CIFAR-10* and BERT tokenizer for *IMDB*, to transform the images and texts to embedding spaces[5]. Then, we treat them as tabular datasets and follow the same active learning process described in Section 3.

Table 9 shows that Uncertainty Sampling with different measures is still competitive to Uniform. In particular, US-ME stands out from other uncertainty measures, and all query strategies achieve similar performance in this experiment. The results indicate that by utilizing the feature extractor to convert image or text data to a tabular format, domain-specific tabular data essentially differs from the regular tabular structure.

In this section, we initialize the preliminary investigation on active learning for domain-specific scenarios. Specifically, we utilize feature extractors for *CIFAR-10* and *IMDB* to convert domain-specific data sets to tabular data sets and verify the feasibility of US for this protocol. However, there is still a lack of comparisons for other approaches. For example, the feature extractors for *CIFAR-10* and *IMDB* we chose are pre-trained on large-scale datasets and might be considered the 'external knowledge' for the benchmark. Such 'external knowledge' would be obtained by semi-supervised learning or self-supervised learning on both labeled and unlabeled pools (Zhang et al., 2023) or from foundation models (Gupte et al., 2024). We leave the

---

[5]We used the pre-trained ViT (`https://huggingface.co/google/vit-base-patch16-224`) and BERT tokenizer (`https://huggingface.co/google-bert/bert-base-uncased`) from Hugging Face repository

Table 8: 'Mean ± standard deviation' AUBC of XGBoost for multi-class classifications (↑ is better). The scores with [1], [2], or [3] denote the 1st, 2nd and 3rd performance on a dataset. 'TLE' denotes a query strategy that exceeds the time limit. The scores with underline denote its mean AUBC is greater than Uniform's 'mean + standard deviation' AUBC.

| | Uniform | US-SM | US-LC | US-ME | BALD | Core-Set | MCM |
|---|---|---|---|---|---|---|---|
| Iris | 92.95%±1.88% | 94.05%±0.85% | 94.06%±0.84%[3] | 94.01%±0.91% | *94.07%±0.85%[2]* | **94.45%±0.51%[1]** | 92.87%±0.54% |
| Wine | 92.40%±1.25% | 92.95%±0.50%[3] | *92.98%±0.47%[2]* | 92.94%±0.52% | **93.14%±0.52%[1]** | 92.85%±0.96% | 91.94%±0.64% |
| Abalone | 88.06%±0.27% | **90.14%±0.10%[1]** | 90.04%±0.10%[3] | 89.92%±0.13% | *90.06%±0.11%[2]* | 88.51%±0.26% | 89.57%±0.17% |
| Academic Success | 91.17%±0.21% | **92.30%±0.08%[1]** | 92.28%±0.09%[3] | 92.24%±0.09% | *92.29%±0.08%[2]* | 91.20%±0.25% | 91.73%±0.19% |
| Satellite | 74.02%±0.32% | **74.80%±0.17%[1]** | 74.44%±0.26%[3] | 74.33%±0.31% | *74.49%±0.23%[2]* | 74.00%±0.28% | 74.38%±0.23% |
| Dry Bean | 22.39%±0.47% | **23.08%±0.00%[1]** | *22.95%±0.09%[2]* | 22.45%±0.05% | 22.50%±0.44% | 22.34%±0.05% | 22.73%±0.08%[3] |
| Diabetes 130 | 53.92%±0.43% | **55.26%±0.44%[1]** | *54.39%±0.87%[2]* | 53.77%±0.74% | TLE | 54.13%±0.15%[3] | 54.04%±0.58% |

Table 9: 'Mean ± standard deviation' AUBC of XGBoost for multi-class classifications (↑ is better). The scores with [1], [2], or [3] denote the 1st, 2nd and 3rd performance on a dataset. 'TLE' denotes a query strategy that exceeds the time limit. The scores with underline denote its mean AUBC is greater than Uniform's 'mean + standard deviation' AUBC.

| | Uniform | US-SM | US-LC | US-ME | BALD | Core-Set | MCM |
|---|---|---|---|---|---|---|---|
| CIFAR-10 | 97.59%±0.11% | 98.49%±0.02%[3] | 98.48%±0.04% | *98.50%±0.02%[2]* | 98.47%±0.03% | **98.51%±0.03%[1]** | 97.02%±0.68% |
| IMDB | 93.75%±0.09% | 93.88%±0.06% | 93.88%±0.06% | *93.91%±0.06%[2]* | 93.90%±0.07%[3] | **93.98%±0.01%[1]** | 93.89%±0.06% |

investigation on the protocol choice for domain-specific datasets and its impact on active learning methods in future work.

## 5 Analysis of uncertainty sampling

In this section, we first study the impact of **model compatibility** on Uncertainty Sampling, which clarifies the conflicting conclusions between our benchmark and the previous work of Zhan et al. (2021) (Section 5.1). Then, we extend the benchmark by evaluating the usefulness of Uncertainty Sampling on three real-world datasets, which are large-scale or high-dimension used in the recent tabular benchmark (Grinsztajn et al., 2022) (Section 5.2). Lastly, we study the sensitivity of active learning protocols for Uncertainty Sampling, including imbalanced datasets with the one-shot protocol, hyper-parameters/ model complexity, and query batch sizes (Section 5.3). We also present the limitations of Uncertainty Sampling to remind AL practitioners of the ineffectiveness of some tabular classification scenarios (Section 5.4).

### 5.1 Impact of non-compatible models for uncertainty sampling

In contrast to the broader performance comparisons in earlier sections, Section 5.1 focuses on the **model compatibility** with US. Our investigation demonstrates that the incompatibility between query-oriented and task-oriented models significantly influences the performance of US. An example of model incompatibility is that the previous benchmark adopted US with LR($C = 1$) as the query-oriented model and RBFSVM as the task-oriented model (Zhan et al., 2021).[6] Through careful analysis, we found that when non-compatible models are used (denoted as US-NC), the performance of US (denoted as US-C) notably drops, as shown in Table 19. This drop is primarily due to the misalignment of the decision boundaries between the query-oriented and task-oriented models, which can lead the query-oriented model to select samples that are not the most uncertain for the task-oriented model, as illustrated in Figure 8. In summary, our benchmarking highlights that by utilizing compatible models, US-C consistently performs better than US-NC on average.

We compare different combinations of query-oriented and task-oriented models based on LR, RBFSVM, and RF. Figure 9 and Appendix C.4 emphasize that compatible model pairs perform better than non-compatible

---

[6]See Zhan et al. (2021)'s implementation for more details `https://github.com/SineZHAN/ComparativeSurveyIJCAI2021PoolBasedAL/blob/master/Algorithm/baseline-google-binary.py#L242`.

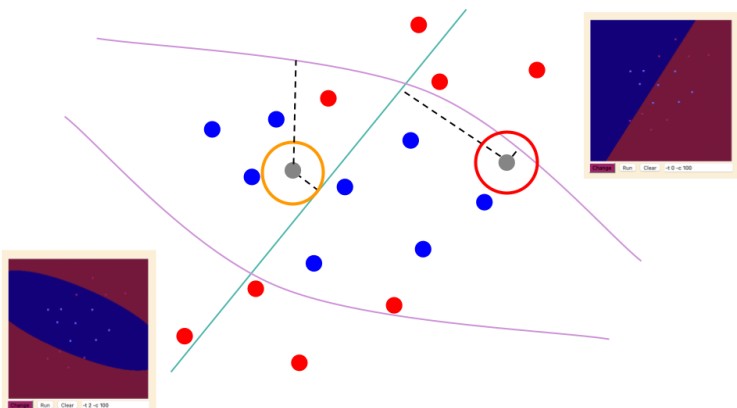

Figure 8: Given RBFSVM as the task-oriented model, we study the non-compatible query-oriented model with LR($C = 0.1$). The red and blue points represent labeled examples. The gray points represent unlabeled examples. The cyan and magenta lines indicate the decision boundaries of query models LR($C = 0.1$) and RBFSVM trained on current labeled examples. If we adopt US, the non-compatible setting queries a sample (orange circle) that is most uncertain to LR($C = 0.1$) rather than the most uncertain sample to RBFSVM (red circle).

model pairs for US, evident across 22 datasets, where the optimal AUBC score occurs with compatible models, i.e., the highest AUBC score is found along the diagonal. Although some results demonstrate that non-compatible models are slightly better than compatible models, such as *Splice* and *Banana* in Figure 18, these instances were exceptions rather than the norm in our benchmark.

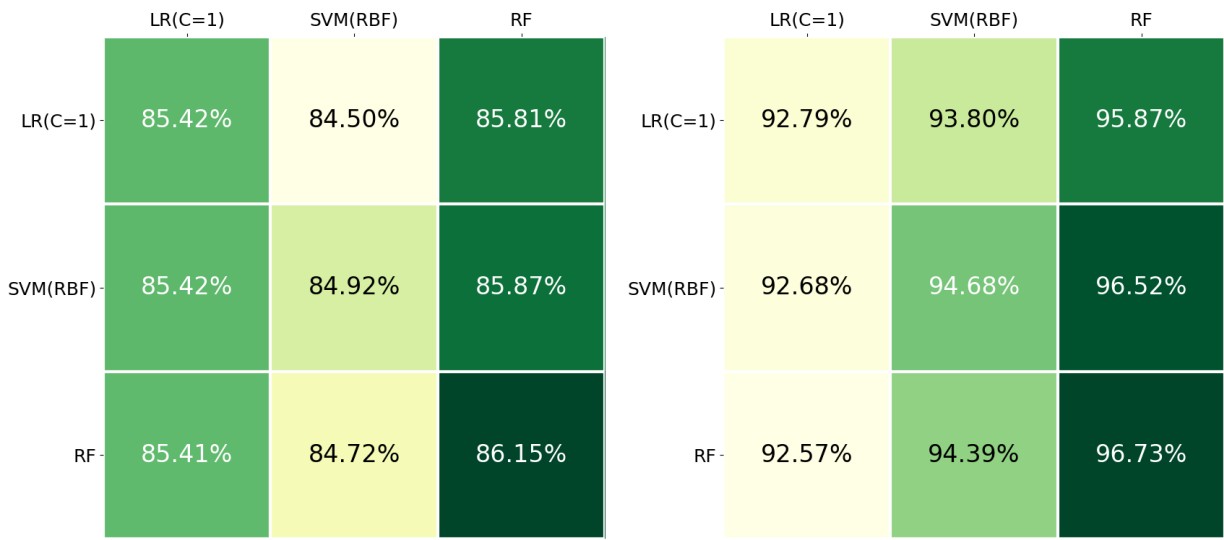

Figure 9: Mean AUBC of a query-oriented model (rows) and a task-oriented model (columns) on *Australian* (left) and *Phishing* (right)

In summary, we advocate for the default use of compatible model parings in US for practical applications. This setting simplifies the model selection process and can potentially yield better performance across various datasets.

## 5.2 Extending the usefulness of uncertainty sampling

We extend the existing benchmark to real-world datasets used in another tabular data benchmark (Grinsztajn et al., 2022) to demonstrate the usefulness of US within our current benchmark and its potential applicability and benefits across a more comprehensive array of real-world datasets. Real-world datasets include a larger number of examples, such as *Pol* and *Covertype*, and higher dimensions, such as *Bioresponse*. By extending our evaluation to these datasets, we aim to illustrate that the consistent usefulness of US is not limited to the existing benchmark.

In Figure 10, similar to Section 4.2, US could bring more benefits than Uniform at the early stage. These results affirm that US has potential as an applicable approach across large-scale and high-dimension scenarios, which encourages the exploration of US in broader applications.

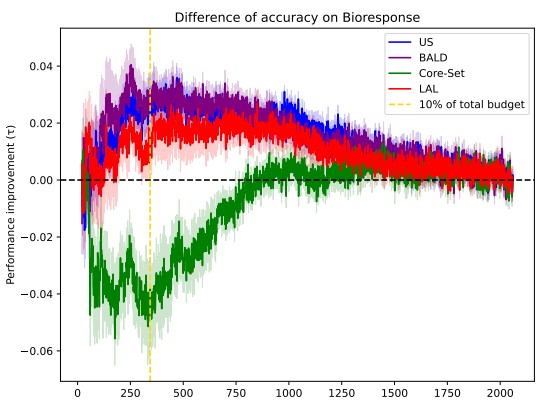 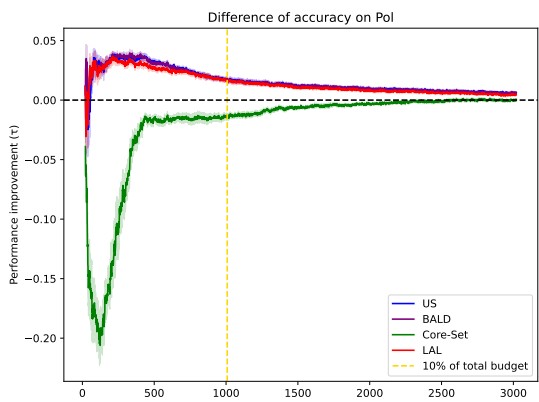

Figure 10: Mean difference of AUBC (improvement) of a query strategy from Uniform on *Bioresponse* (left) and *Pol* (right).

## 5.3 Sensitive of active learning protocols for uncertainty sampling

### 5.3.1 Outcomes of the imbalanced datasets with the one-shot protocol

We collect imbalanced datasets in our benchmark (See Table 17 for details). To evaluate the results under imbalanced data, we report the weighted F1 score[7] of datasets. We especially check the datasets with an imbalance ratio $r \geq 4$ with the one-shot protocol, where at least one label is used for each class. This scenario is practical, that the new project is initialized with only a few data for each class with an unknown label distribution of the test set.

Table 10 demonstrates that no query strategy is significantly superior on most datasets. In particular, all query strategies cannot have significant improvement over Uniform in *Appendicitis*, *Myocardial*, *Abalone*, and *Diabetes 130*. We argue that existing active learning algorithms lack consideration of imbalanced data and expect practitioners to investigate Uncertainty Sampling on imbalanced tabular datasets in future work[8].

### 5.3.2 Outcomes of different hyper-parameters for the query-oriented model

Previous sections illustrate that Uncertainty Sampling with compatible XGBoost task-oriented and query-oriented models is a strong baseline for our benchmark. In this section, we verify whether the change in the model complexity of the query-oriented model would influence the Uncertainty Sampling. The model

---

[7]The weighted F1 score is usually used for imbalanced datasets to handle unequal class distribution issues.

[8]Recent work investigates active learning for imbalanced datasets in the CV domain and suggests incorporating a balancing step into the labeling process to mitigate imbalance within the labeled pool (Aggarwal et al., 2020).

Table 10: 'Mean ± standard deviation' weighted F1 score of XGBoost for imbalanced classifications with the one-shot protocol at 20% total budget (↑ is better). The scores with **bold** denote the best performance on a dataset. The scores with underline denote its mean AUBC is greater than Uniform's 'mean + standard deviation' AUBC.

| Data | Uniform | US-SM | Core-Set | BALD | MCM |
|---|---|---|---|---|---|
| Appendicitis | 80.07%±8.439% | 80.15%±8.804% | **84.20%±7.734%** | *80.87%±10.225%* | 80.38%±8.579% |
| Tic | 88.99%±1.303% | **91.08%±1.147%** | 88.18%±1.691% | *91.02%±1.170%* | 90.13%±1.480% |
| Myocardial | 95.14%±0.423% | 95.50%±0.144% | 95.21%±0.199% | *95.53%±0.133%* | **95.56%±0.096%** |
| Abalone | *20.98%±1.001%* | 20.97%±1.039% | 19.57%±1.017% | **20.99%±1.293%** | 20.69%±1.073% |
| Dry Bean | 90.83%±0.443% | *92.27%±0.203%* | 90.70%±0.652% | **92.29%±0.205%** | 91.33%±0.520% |
| Diabetes 130 | *49.85%±0.853%* | **50.52%±0.872%** | 49.41%±0.199% | TLE | 49.31%±1.339% |
| RT-IoT2022 | 98.52%±0.250% | *99.71%±0.028%* | 83.80%±10.681% | **99.72%±0.053%** | 98.93%±1.054% |

complexity of the XGBoost model is controlled by the hyper-parameters, e.g., we could reduce the proportional number of leaves in the trees by adjusting `min_child_weight` in XGBoost. Therefore, we further launched thought experiments that compare the default and best hyper-parameters of the query-oriented model on small datasets over the same structure with different hyper-parameters. Concretely, we follow the hyper-parameters tuning process in previous work (Grinsztajn et al., 2022) to get the best hyper-parameters of the XGBoost models for each dataset. Then, we create the XGBoost models with these hyper-parameters and repeat the same active learning process described in Section 3.

Figure 11 demonstrates that the query-oriented model with the best hyper-parameters obtains slightly better mean AUBC than the default hyper-parameters on most datasets. We conjecture that XGBoost, which has high model complexity, performs stably on our benchmark; hence, the tuning hyper-parameters do not affect these results too much. In summary, the compatible models with the default hyper-parameters of XGBoost would achieve good performance for most tabular datasets.

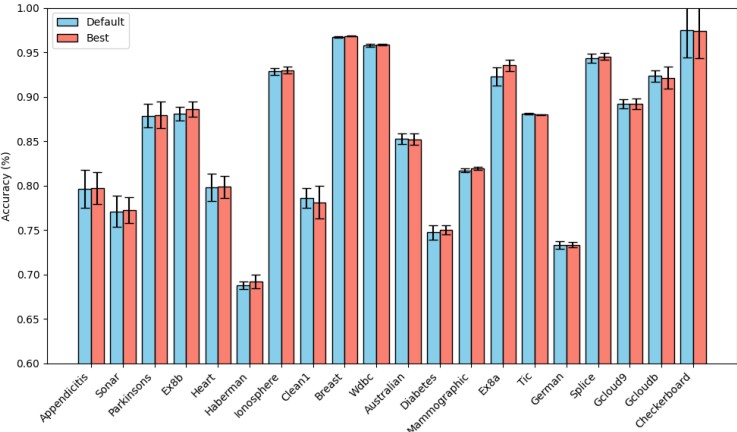

Figure 11: Mean and standard deviation AUBC of US with the default and the best hyper-parameters for XGBoost as the query-oriented model for each dataset. *Note.* We use the best hyper-parameters for XGBoost as the task-oriented model.

### 5.3.3 Outcomes of different query batch sizes

Early studies design the active learning algorithms based on a serial query, i.e., query an example at each round in the early stage. However, a serial query would be inefficient when the size of the query-oriented model increases or the training process becomes slower. Increasing the query batch size $B$ at each round becomes the choice to overcome these issues. Nevertheless, directly selecting the most $B$-highest uncertain

Table 11: Accuracy (↑ is better) of the model with 3000 labeled examples for the batch size $B = 1$, $B = 10$, and $B = 100$ on large-scale datasets.

| Data | 1 (Original) | 10 | 100 |
|---|---|---|---|
| Covertype | **79.34%** | *79.29%* | 79.08% |
| Pol | 98.29% | **98.40%** | *98.39%* |
| Phishing | **96.80%** | *96.75%* | 96.74% |
| Dry Bean | **92.63%** | *92.60%* | *92.60%* |
| Diabetes 130 | **56.48%** | *56.37%* | 56.35% |

examples for Uncertainty Sampling might query redundant examples, resulting in inefficiency. In this section, we study the impact of increasing the batch size on US without designing complicated techniques.

Table 11 shows the degrading of Uncertainty Sampling when increasing batch size from 1 to 10, and 100, indicating the existing ineffective examples queried by batch-mode Uncertainty Sampling. While previous works made an effort to design **hybrid criteria** such as BMDR and SPAL to improve batch-mode Uncertainty Sampling, most methods suffer from the computational cost (in Appendix E) and only bring small benefits compared to Uniform (in Table 7). For future work, we suggest that the researcher consider the trade-off between the improvement over Uniform and the computational cost of their design.

### 5.4 Limitations of uncertainty sampling

Previous benchmarks show that query strategies may not outperform Uniform in specific settings or tasks (Yang & Loog, 2018; Desreumaux & Lemaire, 2020; Karamcheti et al., 2021; Munjal et al., 2022). Our findings demonstrated in Table 7 also indicate that uncertainty sampling does not excel on datasets like *Checkerboard* and *Banana*. Several works study possible reasons for the failure of Uncertainty Sampling (Mussmann & Liang, 2018; Karamcheti et al., 2021; Tifrea et al., 2022) to realize the applicability of active learning algorithms. It underscores the need to explore robust baselines for pool-based active learning, particularly in real-world scenarios (Lu et al., 2023).

In this work, we give the preliminary results of exploration Uncertainty Sampling for tabular datasets, covering binary classification and multi-class classification. Although Uncertainty Sampling provides promising for some of these scenarios, the usefulness and effectiveness of Uncertainty Sampling still have a gap and are unclear in domain-specific classification and imbalance learning (Johnson & Khoshgoftaar, 2019). We believe there are still potential research directions, and our protocol could help researchers explore more scenarios in future works.

## 6 Conclusion

This work presents the most comprehensive survey and open-source benchmark for active learning to date. Our benchmark, with its transparent and unified interface, incorporates existing GitHub repositories, providing a thorough and up-to-date comparison of active learning query strategies. We equip Uncertainty Sampling with compatible models and affirm that it remains superior to other active learning strategies as well as Uniform Sampling on most of the datasets. Furthermore, we discover that Uncertainty Sampling can be affected by the incompatibility between query-oriented and task-oriented models, resulting in discrepancies between previous benchmarks. Our affirmation suggests Uncertainty Sampling with compatible query-oriented and task-oriented models as a first-hand choice for practitioners. These insights not only enhance the community's comprehension of current active learning strategies but also establish a foundation for future research with this practical guide. We anticipate extending our framework to encompass diverse domains like vision and languages and incorporating various models such as deep neural networks, as outlined in Appendix F for future exploration.

**Broader Impact Statement**

Active learning is a long-term research topic in machine learning, yet achieving a consensus on the best strategies within the community is challenging. This work starts from the tabular data to build the most comprehensive open-source active learning benchmark to date. We affirm that Uncertainty Sampling (US) remains superior to other active learning strategies and Uniform on most datasets. We also clarify conflicting conclusions in previous benchmarks by carefully verifying previous settings. Our work will benefit the active learning community by providing a transparent and unified framework for evaluating active learning strategies compared to a strong baseline–US with compatible settings. We hope our work will help practitioners check the reality of existing active learning strategies and settings for different domains. Moreover, re-examine the potential issues in existing benchmarks, such as neglected settings and unpublished analysis steps.

Studying active learning beyond pursuing high accuracy is also important. For example, some AL works studied ML fairness (Anahideh et al., 2022) and privacy issues (Feyisetan et al., 2019) for ethical considerations, which are essential for using new ML/AL techniques for other fields. To study AL for different real-world scenarios, we trust our benchmark and experimental protocol is the solid foundation for future works.

**Acknowledgments**

We thank the anonymous reviewers, the responsible action editor and members of Computational Learning Lab for their constructive feedback and positive interactions. This work is supported by the National Science and Technology Council in Taiwan via NSTC 113-2628-E-002-003, NSTC 113-2634-F-002-008 and NSTC 112-2628-E-002-030. We thank to National Center for High-performance Computing (NCHC) of National Applied Research Laboratories (NARLabs) in Taiwan for providing computational and storage resources.

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

Table 12: Accuracy of the model with 10% labeled examples: We report the accuracy of the model with 10% labeled examples on each dataset. The **bold** indicates the first place. 'TLE' means a query strategy exceeds the time limit.

| | Uniform | US | QBC | BALD | Hier | Graph | Core-Set | HintSVM | QUIRE | DWUS | MCM | BMDR | ALBL | LAL |
|---|---|---|---|---|---|---|---|---|---|---|---|---|---|---|
| Ionosphere | 74.99% | 75.43% | 74.94% | 74.91% | 74.57% | 74.86% | 75.82% | 74.91% | 75.30% | 74.31% | 74.83% | 75.06% | 75.25% | **75.96%** |
| Clean1 | 62.60% | 62.65% | 62.32% | 62.39% | **63.36%** | 61.67% | 62.52% | 61.93% | 62.60% | 62.60% | 61.44% | 61.71% | 62.82% | 62.88% |
| Breast | 93.33% | 94.95% | 93.13% | 94.68% | 94.26% | 90.32% | 93.61% | 93.74% | 93.56% | 90.49% | 94.65% | 93.18% | 94.73% | **95.88%** |
| Wdbc | 90.54% | 90.28% | 90.58% | 91.38% | 90.09% | 87.78% | 91.33% | 90.49% | 90.96% | 90.07% | 89.88% | 89.29% | 91.92% | **92.32%** |
| Australian | 81.00% | **83.13%** | 81.17% | 82.96% | 81.37% | 80.44% | 76.49% | 78.11% | 77.38% | 79.75% | 80.85% | 80.20% | 80.11% | 82.36% |
| Diabetes | 70.22% | **72.07%** | 70.52% | 71.40% | 70.85% | 69.98% | 69.25% | 69.34% | 68.88% | 69.30% | 70.90% | 70.29% | 70.43% | 71.68% |
| Mammographic | 79.92% | 81.45% | 79.38% | **81.76%** | 79.46% | 79.40% | 79.61% | 78.65% | 79.07% | 78.72% | 81.32% | 79.29% | 80.26% | 81.56% |
| Ex8a | 79.53% | 79.62% | 79.70% | 78.66% | 79.49% | 78.32% | **83.99%** | 76.19% | 76.52% | 71.73% | 78.38% | 82.90% | 79.19% | 82.25% |
| Tic | 86.54% | **88.12%** | 86.63% | 87.95% | 86.46% | 84.66% | 87.33% | 85.39% | 85.46% | 81.19% | 87.07% | TLE | 87.58% | 87.09% |
| German | 69.23% | **70.75%** | 69.46% | 70.37% | 69.11% | 69.35% | 68.96% | 69.18% | 69.28% | 68.47% | 69.64% | TLE | 69.29% | 70.45% |
| Splice | 83.16% | **84.98%** | 82.06% | 84.91% | 82.45% | 75.17% | 76.06% | 78.42% | 83.03% | 79.81% | 81.08% | 80.61% | 81.79% | 82.71% |
| Gcloudb | 86.36% | 89.11% | 86.39% | 88.86% | 87.14% | 83.93% | 87.18% | 84.21% | 84.60% | 84.17% | 88.84% | 87.09% | 87.57% | **89.15%** |
| Gcloudub | 88.19% | 88.91% | 88.41% | 87.93% | 88.47% | 86.93% | 86.82% | 82.97% | 84.66% | 79.88% | 86.87% | 87.70% | 86.01% | **90.77%** |
| Checkerboard | 97.32% | 95.81% | 97.08% | 97.67% | 97.93% | 96.57% | 96.78% | 90.18% | 87.09% | 75.27% | 98.66% | 98.23% | 94.78% | **99.80%** |
| Spambase | 90.77% | **93.85%** | 89.93% | 93.62% | 90.63% | 90.97% | 88.96% | 85.38% | TLE | 90.85% | 93.11% | TLE | 91.09% | 93.04% |
| Banana | 85.76% | 83.20% | 86.17% | 82.74% | 86.53% | 83.44% | 87.51% | 69.76% | 62.64% | 70.87% | 84.51% | TLE | 85.29% | **88.43%** |
| Phoneme | 81.66% | **85.13%** | 80.97% | 84.77% | 82.30% | 80.62% | 81.45% | 78.38% | TLE | 76.87% | 83.62% | TLE | 81.47% | 83.64% |
| Ringnorm | 89.55% | 93.89% | 89.92% | **94.02%** | 87.82% | 57.35% | 54.17% | 55.70% | TLE | 57.73% | 92.03% | TLE | 88.21% | 89.32% |
| Twonorm | 93.97% | 95.92% | 93.56% | **95.95%** | 93.31% | 93.73% | 94.45% | 80.79% | TLE | 92.31% | 95.70% | TLE | 94.50% | 94.42% |
| Phishing | 92.17% | 94.62% | 92.06% | **94.84%** | 91.76% | 91.18% | 92.22% | 90.97% | TLE | 87.74% | 93.80% | TLE | 93.20% | 94.28% |
| Covertype | 70.98% | **73.70%** | 70.22% | 73.43% | TLE | 64.46% | TLE | 61.73% | TLE | 58.76% | 67.50% | TLE | 66.92% | 71.90% |
| Bioresponse | 66.64% | 68.38% | 66.74% | **68.99%** | 65.45% | 65.64% | 63.52% | 60.64% | TLE | 66.64% | 67.77% | TLE | 66.55% | 67.88% |
| Pol | 93.24% | 96.67% | 93.27% | **96.84%** | 92.96% | 84.26% | 85.70% | 75.77% | TLE | 93.24% | 95.24% | TLE | 93.67% | 96.15% |

# A    Detailed benchmarking results of XGBoost

We present more settings of the benchmarking results for XGBoost for verifying the superiority of query strategies in Section 4.1. We check the accuracy of the model with different ratios, e.g., 10% and 30% of labeled examples on each dataset. Tables 12 and 13 also confirm that US outperforms other query strategies on most datasets. It is worth mentioning that LAL achieves good performance on *Gcloudb*, *Gcloudub*, and *Checkerboard* when the ratio of labeled examples is 10%. However, these datasets are synthetic, and their features may be more similar to the pre-trained datasets used by LAL, resulting in LAL's exceptional performance on these datasets.

We also extend the Table 5 to the number of queried examples required to reach 90% and 95% of the model's performance trained on the full budget. Table 14 demonstrates that Uncertainty Sampling stably achieves first place or second place on 14 datasets. Similarly, Table 15 demonstrates that Uncertainty Sampling achieves first place or second place on 15 datasets. These results verify that Uncertainty Sampling is competitive in our benchmark.

# B    Revision of Zhan et al. (2021)

In this section, we reveal and revise descriptions in Zhan et al. (2021) to study the conflicting conclusions in previous benchmarks and provide clear information to the active learning community. We appreciate that Zhan et al. (2021) published their source code on GitHub.[9] Thus we could examine the difference from our settings.

## B.1    Experimental Settings

**Inputs and base models.**    At the initial setup, Zhan et al. (2021) employed a random split of 60% of the dataset for the training set and the remaining 40% for the testing set. No pre-processing was applied to the dataset, and fixed random seeds were used to ensure consistency in the training and testing sets across repeated experiments. They used an RBFSVM as the task-oriented model for evaluating the query strategies.

---

[9] https://github.com/SineZHAN/ComparativeSurveyIJCAI2021PoolBasedAL

Table 13: Accuracy of the model with 30% labeled examples: We report the accuracy of the model with 30% labeled examples on each dataset. The **bold** indicates the first place. 'TLE' means a query strategy exceeds the time limit.

| | Uniform | US | QBC | BALD | Hier | Graph | Core-Set | HintSVM | QUIRE | DWUS | MCM | BMDR | ALBL | LAL |
|---|---|---|---|---|---|---|---|---|---|---|---|---|---|---|
| Sonar | 71.18% | 71.71% | 70.73% | 71.90% | 70.89% | 71.02% | 67.89% | 69.06% | 70.20% | 71.43% | 70.10% | **72.37%** | 71.75% | 71.99% |
| Parkinsons | 81.44% | **84.95%** | 81.88% | 84.38% | 81.87% | 81.95% | 81.58% | 80.50% | 81.74% | 81.74% | 83.74% | 82.67% | 83.21% | 84.08% |
| Ex8b | 82.76% | 84.21% | 82.99% | 83.14% | 83.11% | 82.54% | 84.42% | 81.99% | 83.48% | 81.54% | 83.88% | 83.51% | 84.40% | **84.87%** |
| Heart | 77.78% | **79.78%** | 77.90% | 79.50% | 77.85% | 77.84% | 78.07% | 76.76% | 77.99% | 77.84% | 78.29% | 78.49% | 78.37% | 79.27% |
| Haberman | 68.87% | **71.21%** | 68.41% | 70.92% | 68.63% | 68.96% | 68.54% | 67.49% | 67.65% | 69.13% | 70.33% | 69.24% | 68.85% | 70.33% |
| Ionosphere | 85.97% | **90.30%** | 86.70% | 90.27% | 87.21% | 87.41% | 84.42% | 84.07% | 79.92% | 78.34% | 88.85% | 81.04% | 88.76% | 89.99% |
| Clean1 | 72.59% | **74.79%** | 72.52% | 74.26% | 72.36% | 72.40% | 71.87% | 72.59% | 72.87% | 72.59% | 72.54% | 72.03% | 73.59% | 74.59% |
| Breast | 95.68% | 96.69% | 95.77% | 96.76% | 96.26% | 95.52% | 96.08% | 95.74% | 95.99% | 90.15% | 96.64% | 95.62% | 96.36% | **96.81%** |
| Wdbc | 93.73% | 95.95% | 93.77% | 95.95% | 93.86% | 93.86% | 94.07% | 92.98% | 93.94% | 93.58% | 95.94% | 93.71% | 95.75% | **95.99%** |
| Australian | 84.59% | **86.25%** | 84.62% | 86.12% | 84.56% | 85.00% | 84.50% | 82.80% | 84.79% | 83.76% | 86.00% | 84.76% | 84.91% | 85.77% |
| Diabetes | 72.60% | **74.27%** | 72.84% | 73.84% | 72.72% | 73.12% | 73.19% | 70.34% | 72.86% | 71.05% | 73.59% | 72.59% | 72.54% | 72.78% |
| Mammographic | 79.53% | 82.05% | 79.27% | 82.07% | 79.42% | 78.87% | 79.10% | 79.10% | 79.60% | 78.65% | **82.08%** | 79.89% | 79.17% | 81.69% |
| Ex8a | 91.08% | 92.63% | 91.30% | 92.75% | 91.20% | 91.01% | **94.96%** | 77.68% | 80.40% | 82.04% | 93.09% | 92.75% | 88.08% | 93.89% |
| Tic | 90.12% | 91.07% | 90.12% | **91.29%** | 90.11% | 90.15% | 89.58% | 89.37% | 90.21% | 88.67% | 90.78% | TLE | 89.79% | 90.11% |
| German | 72.16% | **73.73%** | 72.50% | 73.56% | 72.35% | 72.71% | 72.11% | 72.40% | 71.84% | 70.83% | 73.47% | TLE | 72.96% | 73.24% |
| Splice | 91.53% | **95.47%** | 91.53% | 95.17% | 91.36% | 91.75% | 90.43% | 89.53% | 91.76% | 87.21% | 94.79% | 91.80% | 91.82% | 92.98% |
| Gcloudb | 88.09% | 89.06% | 88.20% | 89.03% | 88.23% | 88.33% | 88.56% | 84.96% | 84.50% | 85.99% | 89.15% | 88.26% | 88.62% | **89.27%** |
| Gcloudub | 92.67% | **95.38%** | 92.52% | 94.84% | 93.37% | 92.76% | 90.66% | 84.03% | 87.36% | 81.48% | 94.49% | 91.84% | 91.76% | 94.42% |
| Checkerboard | 99.29% | 97.22% | 99.19% | 99.76% | 99.61% | 99.16% | 99.47% | 91.51% | 91.71% | 79.49% | 99.65% | 99.56% | 99.29% | **99.81%** |
| Spambase | 93.10% | **95.14%** | 93.22% | 95.11% | 93.45% | 93.07% | 92.99% | 90.03% | TLE | 93.16% | 94.88% | TLE | 93.49% | 95.04% |
| Banana | 88.02% | 89.08% | 87.83% | 88.98% | 87.86% | 87.75% | 87.96% | 74.16% | 79.78% | 76.83% | **89.16%** | TLE | 87.43% | 88.98% |
| Phoneme | 85.14% | **88.47%** | 85.14% | 88.20% | 86.02% | 84.93% | 85.59% | 83.45% | TLE | 80.68% | 88.15% | TLE | 85.15% | 88.00% |
| Ringnorm | 94.22% | 96.15% | 94.59% | **96.32%** | 93.38% | 94.01% | 51.04% | 64.64% | TLE | 55.09% | 96.12% | TLE | 94.06% | 94.94% |
| Twonorm | 95.67% | 96.76% | 95.67% | **96.77%** | 95.76% | 95.73% | 96.03% | 77.35% | TLE | 95.08% | 96.75% | TLE | 96.02% | 96.57% |
| Phishing | 94.09% | 96.56% | 93.67% | 96.55% | 93.79% | 93.98% | 93.83% | 91.96% | TLE | 91.03% | 96.38% | TLE | 94.81% | **96.60%** |
| Covertype | 73.54% | **76.30%** | 73.16% | 76.25% | TLE | 65.82% | TLE | 63.94% | TLE | 60.21% | 73.13% | TLE | 69.72% | 74.64% |
| Bioresponse | 72.11% | 74.10% | 72.51% | **74.72%** | 71.91% | 73.20% | 69.89% | 69.11% | TLE | 72.11% | 73.33% | TLE | 71.76% | 73.78% |
| Pol | 96.34% | **98.28%** | 96.22% | 98.25% | 96.41% | 96.70% | 94.84% | 91.99% | TLE | 96.34% | 98.14% | TLE | 95.93% | 98.25% |

Table 14: The minimum number of queried examples required to reach 90% accuracy (↓ is better) of the model: The **bold** indicates the first place and *italics* indicates the second place. 'TLE' means a query strategy exceeds the time limit.

| | Uniform | US | QBC | BALD | Hier | Graph | Core-Set | HintSVM | QUIRE | DWUS | MCM | BMDR | ALBL | LAL |
|---|---|---|---|---|---|---|---|---|---|---|---|---|---|---|
| Appendicitis | 20.88 | **20.62** | 21.16 | 20.71 | 20.89 | 21.32 | 20.85 | 20.98 | 21.09 | 21.82 | *20.64* | 21.04 | 20.87 | 20.82 |
| Sonar | 34.95 | 33.94 | 34.65 | 34.23 | 33.62 | 35.54 | 39.05 | 38.70 | 33.47 | 35.75 | 34.23 | **31.54** | 33.84 | *32.67* |
| Parkinsons | 26.22 | *23.29* | 26.81 | 23.68 | 25.02 | 27.32 | 26.84 | 32.36 | 27.29 | 26.10 | 24.09 | 24.74 | 24.61 | **23.28** |
| Ex8b | 25.01 | 23.57 | 24.60 | 23.24 | 24.67 | 23.34 | *23.02* | 27.20 | 24.15 | 26.79 | 23.36 | 23.03 | 23.13 | **22.07** |
| Heart | 23.35 | 23.20 | 23.21 | *22.76* | 23.28 | 22.76 | 24.40 | 22.98 | 24.64 | 23.85 | 23.85 | 23.85 | 23.85 | 23.13 |
| Haberman | 20.45 | *20.32* | 20.51 | 20.51 | 20.60 | 20.62 | 22.09 | 20.95 | 21.21 | 20.54 | **20.26** | 20.53 | 20.40 | 20.47 |
| Ionosphere | 35.05 | 30.38 | 32.74 | 30.02 | 32.79 | 38.32 | 32.26 | 35.42 | 39.80 | 93.23 | 32.94 | 50.42 | **27.56** | *28.05* |
| Clean1 | 97.02 | 85.99 | 99.13 | *85.04* | 102.03 | 101.63 | 102.60 | 101.68 | 96.60 | 97.02 | 97.67 | 104.31 | 87.96 | **84.58** |
| Breast | 21.04 | 20.88 | 20.95 | 22.25 | 20.89 | 24.60 | 20.98 | 21.03 | 20.85 | 44.26 | 21.50 | 21.50 | *20.53* | **20.35** |
| Wdbc | 23.33 | 23.00 | 23.38 | 22.37 | 22.63 | 25.25 | 22.04 | 23.56 | 22.42 | 23.41 | 23.00 | 24.37 | *21.53* | **20.92** |
| Australian | 25.02 | 24.35 | 24.61 | **23.85** | 25.30 | 26.25 | 30.30 | 33.27 | 28.87 | 26.19 | 27.65 | 26.27 | 26.89 | *24.32* |
| Diabetes | 22.42 | *22.02* | 22.99 | 22.30 | 22.02 | 22.87 | 22.53 | 23.22 | 22.80 | 25.03 | 23.36 | 23.37 | **21.86** | 22.16 |
| Mammographic | 21.07 | **20.28** | 20.40 | 20.70 | 20.39 | 20.44 | 20.49 | *20.29* | 20.33 | 20.99 | 20.33 | 20.39 | 20.39 | 20.38 |
| Ex8a | 74.56 | 76.86 | 76.89 | 75.54 | 79.67 | 71.65 | **55.77** | 248.32 | 214.78 | 154.69 | 82.22 | 63.65 | 99.55 | *58.75* |
| Tic | 22.80 | *21.60* | 23.68 | 22.06 | 22.40 | 24.23 | 22.12 | 24.19 | 25.99 | 27.35 | 21.70 | TLE | **21.55** | 22.04 |
| German | 29.63 | **27.72** | 30.15 | *28.29* | 29.54 | 29.94 | 30.92 | 33.47 | 32.72 | 35.41 | 28.32 | TLE | 30.26 | 29.29 |
| Splice | 71.31 | **57.28** | 77.10 | *57.87* | 75.11 | 91.75 | 107.48 | 107.19 | 70.71 | 120.35 | 73.65 | 80.92 | 77.76 | 69.37 |
| Gcloudb | 22.37 | 22.99 | 22.41 | 22.18 | 22.18 | 24.08 | **20.92** | 21.53 | 22.42 | 24.02 | 22.84 | *21.15* | 21.19 | 21.28 |
| Gcloudub | 38.13 | 38.29 | 36.94 | 45.03 | 36.18 | 41.46 | 35.65 | 84.29 | 60.09 | 206.32 | 44.59 | **35.10** | 38.94 | *35.51* |
| Checkerboard | 39.15 | 95.46 | 41.08 | 41.61 | 37.36 | 41.37 | 33.60 | 126.14 | 93.89 | 420.74 | 45.56 | *30.83* | 53.35 | **28.22** |
| Spambase | 63.70 | **37.10** | 53.10 | 40.70 | 59.60 | 91.80 | 83.60 | 184.30 | TLE | 61.60 | 46.90 | TLE | 56.10 | *38.10* |
| BaTLEa | 73.10 | 222.50 | 85.20 | 201.90 | *72.10* | 195.10 | **67.00** | 1040.10 | 617.60 | 777.70 | 173.90 | TLE | 127.90 | 79.20 |
| Phoneme | 120.50 | 102.20 | 193.70 | 101.70 | *99.60* | 227.00 | 165.20 | 325.10 | TLE | 573.90 | 112.50 | TLE | 138.80 | **91.40** |
| Ringnorm | 165.90 | **101.20** | 162.70 | *104.70* | 209.60 | 478.60 | 2255.00 | 1333.50 | TLE | 2484.30 | 172.20 | TLE | 228.20 | 138.50 |
| Twonorm | 52.50 | **38.40** | 65.70 | 39.10 | 78.70 | 68.60 | 55.60 | 1359.60 | TLE | 80.60 | 54.10 | TLE | 50.70 | *39.00* |
| Phishing | 22.30 | 22.40 | 23.20 | *21.90* | 25.50 | 30.80 | 24.00 | 25.60 | TLE | 52.40 | 22.00 | TLE | **21.30** | 22.80 |
| Covertype | 164.00 | *110.80* | 130.10 | 146.60 | TLE | 1689.20 | TLE | TLE | TLE | **20.00** | TLE | TLE | TLE | 129.70 |
| Bioresponse | 291.00 | *201.70* | 225.10 | **174.60** | 267.10 | 249.40 | 450.10 | TLE | TLE | 291.00 | 222.50 | TLE | TLE | 213.20 |
| Pol | 86.00 | *62.80* | 79.60 | 69.20 | 105.80 | 649.20 | 286.70 | 531.60 | TLE | 86.00 | 96.40 | TLE | 78.30 | **58.60** |

**Query strategies.** To compare the performance of 17 query strategies, they implemented random sampling and all query strategies using different libraries. The libact library provided implementations for Uncertainty Sampling (US), Query by Committee (QBC), Hinted Support Vector Machine (HintSVM),

Table 15: The minimum number of queried examples required to reach 99% accuracy (↓ is better) of the model: The **bold** indicates the first place and *italics* indicates the second place. 'TLE' means a query strategy exceeds the time limit.

| | Uniform | US | QBC | BALD | Hier | Graph | Core-Set | HintSVM | QUIRE | DWUS | MCM | BMDR | ALBL | LAL |
|---|---|---|---|---|---|---|---|---|---|---|---|---|---|---|
| Appendicitis | 24.56 | *22.39* | 24.72 | 22.69 | 23.65 | 24.23 | 23.93 | 23.05 | 24.41 | 26.31 | **22.24** | 24.22 | 23.36 | 22.46 |
| Sonar | 51.31 | 48.78 | 51.26 | *47.09* | 51.27 | 53.33 | 55.92 | 55.22 | 51.48 | 53.32 | 50.29 | 49.14 | 49.01 | **46.11** |
| Parkinsons | 41.89 | **29.74** | 41.95 | *31.35* | 36.36 | 44.87 | 42.01 | 52.58 | 40.87 | 41.77 | 32.14 | 36.16 | 34.49 | 31.69 |
| Ex8b | 34.15 | 30.30 | 34.30 | 31.70 | 32.04 | 34.86 | *29.73* | 39.04 | 33.11 | 42.11 | 30.27 | 32.50 | 30.51 | **28.12** |
| Heart | 28.16 | 28.83 | 30.43 | *27.84* | 29.53 | 30.43 | 27.85 | 31.22 | 28.53 | 31.49 | 30.53 | 30.13 | **27.62** | 28.60 |
| Haberman | 22.17 | 21.35 | 21.32 | *21.27* | 21.70 | 21.59 | 22.86 | 23.73 | 22.91 | 21.95 | **21.06** | 21.49 | 21.53 | 21.36 |
| Ionosphere | 57.59 | *38.77* | 51.75 | 38.94 | 54.52 | 56.78 | 53.82 | 58.17 | 63.56 | 145.18 | 45.75 | 84.66 | 39.71 | **37.18** |
| Clean1 | 154.98 | **116.87** | 149.53 | *117.15* | 152.53 | 156.88 | 145.54 | 145.98 | 143.03 | 154.98 | 129.95 | 149.07 | 129.58 | 121.23 |
| Breast | 27.80 | 26.48 | 27.98 | 26.80 | 24.51 | 38.88 | 26.27 | 28.45 | 24.37 | 110.20 | 26.50 | 29.27 | *23.51* | **21.66** |
| Wdbc | 33.43 | 33.02 | 35.83 | 29.50 | 35.15 | 38.93 | 30.16 | 36.22 | 30.91 | 37.32 | 33.26 | 37.70 | *26.67* | **25.62** |
| Australian | 36.67 | *29.97* | 34.00 | **29.72** | 35.67 | 37.48 | 43.39 | 53.25 | 43.54 | 43.85 | 35.10 | 39.84 | 43.85 | 30.29 |
| Diabetes | 30.10 | 28.37 | 32.14 | 31.09 | *28.33* | 31.92 | 31.15 | 45.45 | 33.72 | 41.25 | 32.06 | 33.58 | 29.50 | **26.06** |
| Mammographic | 22.50 | 21.49 | 22.24 | 21.42 | 23.53 | 23.99 | 21.87 | 22.20 | *21.11* | 25.54 | 23.82 | 21.67 | 21.16 | **21.01** |
| Ex8a | 137.20 | 131.24 | 136.96 | 127.49 | 133.07 | 131.67 | **87.53** | 303.87 | 292.35 | 236.73 | 125.00 | 107.80 | 213.37 | *92.54* |
| Tic | 38.18 | **31.05** | 39.40 | *31.64* | 36.90 | 43.42 | 36.85 | 48.37 | 42.04 | 66.66 | 33.24 | TLE | 34.47 | 35.07 |
| German | 77.75 | **53.58** | 61.17 | 58.86 | 73.46 | 72.13 | 75.27 | 66.40 | 70.93 | 99.70 | 57.47 | TLE | 58.21 | *54.55* |
| Splice | 136.05 | **85.07** | 136.84 | *85.34* | 134.67 | 139.17 | 165.62 | 172.66 | 133.49 | 263.52 | 106.11 | 138.55 | 137.54 | 112.67 |
| Gcloudb | 28.93 | 28.11 | 32.15 | 26.43 | 28.58 | 39.38 | *25.40* | 32.25 | 38.50 | 37.28 | 27.56 | 26.09 | 27.03 | **24.57** |
| Gcloudub | 89.46 | 70.17 | 78.74 | 80.73 | *63.78* | 76.47 | 89.61 | 325.75 | 178.15 | 364.91 | 93.69 | 88.56 | 99.21 | **59.37** |
| Checkerboard | 55.70 | 195.19 | 59.63 | 61.28 | 47.50 | 58.96 | 54.80 | 276.80 | 239.73 | 584.71 | 56.07 | *43.10* | 85.12 | **32.18** |
| Spambase | 223.00 | *81.20* | 256.90 | **78.70** | 180.20 | 205.40 | 252.20 | 620.00 | TLE | 220.00 | 114.10 | TLE | 167.80 | 96.80 |
| BaTLEa | 141.90 | 337.30 | 153.70 | 349.30 | 144.40 | 313.00 | **105.80** | 1474.50 | 888.20 | 1761.10 | 286.30 | TLE | 182.30 | *112.00* |
| Phoneme | 509.70 | **226.40** | 527.30 | *233.70* | 427.00 | 442.00 | 445.00 | 880.60 | TLE | 1455.90 | 303.60 | TLE | 556.80 | 296.00 |
| Ringnorm | 390.40 | *175.10* | 354.40 | **169.20** | 481.80 | 546.80 | 2337.70 | 1503.90 | TLE | 2621.40 | 253.00 | TLE | 442.80 | 361.00 |
| Twonorm | 127.50 | *62.10* | 123.50 | **61.30** | 176.20 | 140.80 | 121.30 | 2320.40 | TLE | 218.30 | 93.20 | TLE | 89.70 | 73.30 |
| Phishing | 99.00 | *47.20* | 101.40 | **46.80** | 75.00 | 215.70 | 83.30 | 138.90 | TLE | 695.40 | 70.70 | TLE | 80.50 | 53.50 |
| Covertype | 642.20 | *231.90* | 759.80 | 248.40 | TLE | 561.70 | TLE | TLE | TLE | **20.00** | TLE | TLE | TLE | 421.80 |
| Bioresponse | 602.40 | *374.70* | 540.10 | **354.60** | 560.40 | 529.00 | 713.60 | TLE | TLE | 602.40 | 496.50 | TLE | TLE | 447.30 |
| Pol | 224.10 | 131.40 | 227.20 | **122.00** | 290.30 | 772.60 | 429.40 | 1009.80 | TLE | 224.10 | 175.20 | TLE | 205.30 | *124.50* |

QUerying Informative and REpresentative Examples (QUIRE), Active Learning by Learning (ALBL), Density Weighted Uncertainty Sampling (DWUS), and Variation Reduction (VR). The Google library included Random Sampling (Uniform), k-Center-Greedy (KCenter or Core-Set), Margin-based Uncertainty Sampling (Margin), Graph Density (Graph), Hierarchical Sampling (Hier), Informative Cluster Diverse (InfoDiv), and Representative Sampling (MCM). The ALiPy library contributed Estimation of Error Reduction (EER), BMDR, SPAL, and LAL. Besides, they proposed the Beam-Search Oracle (BSO) as a reference to approximate the optimal sequence of queried samples that maximizes performance on the testing set, aiming to assess the potential improvement space for query strategies on specific datasets. Through reviewing their released source code, we identified differences between the task-oriented and query-oriented models for specific query strategies. Table 16 highlights the discrepancies between the two models for each query strategy.[10] In particular, Margin and US (US-C and US-NC in our notation) are variant settings for Uncertainty Sampling. We further discuss such differences in Section 5.1. In re-benchmarking (Appendix C), we adopt RBFSVM for a query strategy and evaluation by default.

**Experimental design.** The active learning algorithm was stopped when the total budget was equal to the size of the unlabeled pool, $T = |D_\mathrm{u}^{(0)}|$. They collected the testing accuracy at each round to construct a learning curve, and the AUBC was calculated to summarize the performance of a query strategy on a dataset. To ensure reliable results, they conducted $K_\mathrm{S} = 100$ repeated experiments for small datasets ($n < 2000$) and $K_\mathrm{L} = 10$ repeated experiments for large datasets ($n \geq 2000$), where $n$ represents the size of the dataset. Finally, they compute the average AUBCs across repeated experiments for each query strategy on each dataset.

**Analysis methods.** Zhan et al. (2021) benchmarked the pool-based active learning for classifications on 35 datasets, including 26 binary-class and 9 multi-class datasets collected from LIBSVM and UCI (Chang & Lin, 2011; Dua & Graff, 2017). They provided the data properties, such as the number of samples $n$, dimension $d$, and imbalance ratio IR, where the imbalance ratio is the proportion of negative labels to the

---

[10]The settings are different from their source code for Google and ALiPy[9].

Table 16: Settings of query-oriented models $\mathcal{H}$ for specific query strategies $\mathcal{Q}$ in Zhan et al. (2021).

| $\mathcal{Q}$ | $\mathcal{H}$ |
|---|---|
| US (Zhan et al., 2021) , US-NC (Ours) | LR($C = 0.1$) |
| QBC | LR($C = 1$), SVM(Linear, `probability = True`), SVM(RBF, `probability = True`), Linear Discriminant Analysis |
| ALBL | Combination of QSs with same $\mathcal{H}$: US-C, US-NC, HintSVM |
| VR | LR($C = 1$) |
| EER | SVM(RBF, `probability = True`) |

number of positive labels

$$\text{IR} = \frac{|\{(x_i, y_i) : y_i = +1\}|}{|\{(x_j, y_j) : y_j = -1\}|}.$$

They employed these metrics to analyze the results from different aspects to explain the results of the query strategy's performance on a dataset. We agree with their core idea of the analysis methods and believe their benchmark benefits the research community. However, we observe that the conclusion of their work differs from several previous works. For example, Zhan et al. (2021) claimed that LAL performs better on binary datasets than Uncertainty Sampling while not in the other benchmark (Yang & Loog, 2018). The evidence urges us to re-implement the active learning benchmark to clarify the conflicting claims.

### B.2    Benchmarking datasets

Section 3 records the datasets used in the previous benchmark (Zhan et al., 2021). However, we discover that the attributes of datasets are different. We report the revision in Table 17 via 'Zhan et al. (2021) $\rightarrow$ Our new version'.

### B.3    Failure of the Reproducing Uniform

Table 20 (Table 21) shows the significant difference between ours and Zhan et al. (2021). We noticed an implementation error in the previous benchmark. In Google, Uniform assumes that the data has already been shuffled.[11] However, the implementation in Zhan et al. (2021) does not shuffle the unlabeled pool at first.[12]

```Python
[Code=Python]
dict_data,labeled_data,test_data,unlabeled_data = \
    split_data(dataset_filepath, test_size, n_labeled)

print(unlabeled_data)
# results of indices of unlabeled pool
#[3, 4, 5, 10, 11, 13, 15, 16, 20, 23, 24, 26, 27, 29, 30, 31, 33, 36, 37, 41, 43, 44, 45 \
# 49, 50, 51, 53, 54, 55, 57, 63, 64, 65, 70, 73, 75, 77, 78, 79, 83, 84, 86, 87, 88, 89, \
# 91, 92, 95, 97, 102, 105, 110, 112, 114, 115, 121, 122, 127, 128, 131, 132, 136, 137, \
# 139, 140, 144, 148, 150, 151, 155, 157, 159, 160, 161, 162, 164, 165, 167, 168, 172, 175, \
# 176, 177, 178, 181, 182, 183, 184, 185, 186, 187, 188, 190, 191, 193, 194, 197, 198, 199, \
```

---

[11]https://github.com/google/active-learning/blob/master/sampling_methods/uniform_sampling.py#L40
[12]https://github.com/SineZHAN/ComparativeSurveyIJCAI2021PoolBasedAL/blob/master/Algorithm/
baseline-google-binary.py#L331

Table 17: Benchmarking datasets in this work and revision of Table 2 in Zhan et al. (2021).

|  | Domain | $k$ | $r$ | $d$ | $n$ |
|---|---|---|---|---|---|
| Appendicitis | Health and Medicine | 2 | 4.05 | 7 | 106 |
| Iris | Biology | 3 | 1 | 4 | 150 |
| Wine | Physics and Chemistry | 3 | 1.48 | 13 | 178 |
| Sonar | Physics and Chemistry | 2 | 1 | 60 | 108→208 |
| Parkinsons | Health and Medicine | 2 | 3.06 | 22 | 195 |
| Ex8b | Synthetic | 2 | 1 | 2 | 206→210 |
| Heart | Health and Medicine | 2 | 1 | 13 | 270 |
| Haberman | Health and Medicine | 2 | 2 | 3 | 306 |
| Ionosphere | Physics and Chemistry | 2 | 1 | 34 | 351 |
| Clean1 | Physics and Chemistry | 2 | 1 | 168→166 | 475→476 |
| Breast | Health and Medicine | 2 | 1 | 10 | 478 |
| Wdbc | Health and Medicine | 2 | 1 | 30 | 569 |
| Australian | Business | 2 | 1 | 14 | 690 |
| Diabetes | Health and Medicine | 2 | 1 | 8 | 768 |
| Mammographic | Health and Medicine | 2 | 1 | 5 | 830 |
| Ex8a | Synthetic | 2 | 1 | 2 | 863→766 |
| Tic | Games | 2 | 6 | 9 | 958 |
| German | Social Science | 2 | 2 | 20→24 | 1000 |
| Splice | Biology | 2 | 1 | 61→60 | 1000 |
| Gcloudb | Synthetic | 2 | 1 | 2 | 1000 |
| Gcloudub | Synthetic | 2 | 2→2.03 | 2 | 1000 |
| Checkerboard | Synthetic | 2 | 1→1.82 | 2 | 1600 |
| Myocardial | Health and Medicine | 2 | 20.52 | 111 | 1700 |
| Bioresponse | Biology | 2 | 1 | 419 | 3434 |
| Abalone | Biology | 21 | 114.83 | 8 | 4168 |
| Academic Success | Social Science | 3 | 2.78 | 36 | 4424 |
| Spambase | Computer Science | 2 | 1→1.54 | 57 | 4601 |
| Banana | Synthetic | 2 | 1 | 2 | 5300 |
| Phoneme | Speech | 2 | 2 | 5 | 5404 |
| Satellite | Climate and Environment | 6 | 2.45 | 36 | 6435 |
| Ringnorm | Synthetic | 2 | 1 | 21→20 | 7400 |
| Twonorm | Synthetic | 2 | 1 | 50→20 | 7400 |
| Pol | Synthetic | 2 | 1 | 26 | 10082 |
| Phishing | Computer Science | 2 | 1 | 30 | 2456→11055 |
| Dry Bean | Biology | 7 | 6.79 | 16 | 13611 |
| IMDB | Text | 2 | 1 | 768 | 50000 |
| CIFAR-10 | Image | 10 | 1 | 768 | 60000 |
| Diabetes 130-US Hospitals | Health and Medicine | 3 | 4.83 | 2463 | 101766 |
| RT-IoT2022 | Engineering | 12 | 3380.68 | 94 | 123117 |
| Covertype | Biology | 2 | 1 | 10 | 566602 |

```
# 202, 203, 204, 205, 207, 208]
```

We also modify their Uniform implementation by `shuffle` the `unlabeled_data`. Then, we can obtain similar results based on their source code, see Table 18.

```Python
dict_data,labeled_data,test_data,unlabeled_data = \
    split_data(dataset_filepath, test_size, n_labeled)
```

Table 18: Comparing different train/test/labeled splits on *Sonar*: first column is reprot and reproducing results in Zhan et al. (2021), second column in our implementation, and the third column is reproducing results after we revise Zhan et al. (2021)'s code.

| **Uniform** | Report and code in Zhan et al. (2021) | Our code | Modified code in Zhan et al. (2021) |
|---|---|---|---|
| Google | **0.6274**\* | 0.7513 | 0.7577 |
| libact | _ | 0.7520 | 0.7543 |
| ALiPy | _ | 0.7556 | 0.7579 |

```
random.shuffle(unlabeled_data)
```

The unshuffled implementation in Google significantly impacts binary classification datasets, such as *Sonar*, *Clean1*, and *Spambase*. Also, it affects *Ex8a* and *German*, which enlarges the difference AUBCs between Uniform and other query strategies. Due to this experience, we suggest practitioners ensure the correctness of the baseline method by comparing different implementations before conducting the benchmarking experiments.

## B.4 Query Strategy and Implementation

We revise some descriptions of the query strategies in Zhan et al. (2021):

(1) 'Graph Density (Graph) is a typical parallel-form combined strategy that balances the uncertainty and representative based measure simultaneously via a time-varying parameter (Ebert et al., 2012).'

(2) 'Marginal Probability based Batch Mode AL (Margin) Chattopadhyay et al. (2012) selects a batch that makes the marginal probability of the new labeled set similar to the one of the unlabeled set via optimization by Maximum Mean Discrepancy (MMD).'

(3) 'Kremer et al. (2014) proposed an SVM-based AL strategy by minimizing the distances between data points and classification hyperplane (HintSVM).'

Issue (1): Although Ebert et al. (2012) proposed the reinforcement learning method to select uncertainty and diversity sample(s) during the procedure, Google (Yilei "Dolee" Yang, 2017) does not implement the whole procedure but only the diversity sampling method.[13] Thus, we should categorize it as **diversity-based** method.
Issue (2): Google (Yilei "Dolee" Yang, 2017) does not use MMD to measure the distance. The implementation is uncertainty sampling with a margin score is mentioned in the survey paper (Settles, 2012). Therefore, we should categorize it to **uncertainty-based** method.
Issue (3): libact (Yang et al., 2017) implemented HintSVM based on the work of Li et al. (2015) rather than Kremer et al. (2014).

## B.5 Relationship between query strategies

We provide additional evidence to explain the relationship between query strategies, which supports our experimental results.

(1) US-C and InfoDiv should be the same when the query batch size is one.

(2) Different uncertainty measurements should be the same in the binary classification, indicating that different uncertainty measurements do not cause differences between US-C and US-NC.

---

[13]https://github.com/google/active-learning/blob/master/sampling_methods/graph_density.py

(3) SPAL changes the condition of variables used for discriminative and representative objective functions in BMDR.

Issue (1): InfoDiv clusters unlabeled samples into several clusters, then selects uncertain samples and keeps the same cluster distribution simultaneously.[14] Therefore, it is the same when we set the $B = 1$ to query the most uncertain sample. Zhan et al. (2021) provided the different numbers of US-C and InfoDiv in Table 4, which might have resulted from using the different batch sizes of these query strategies.

Issue (2): The least confidence, margin, and entropy are monotonic functions with a peak equal to $\mathbb{P}(y = +1 \mid x) = 0.5$ in binary classification, such that all of these uncertainty measurements would query the same point (Settles, 2012).

Issue (3) The optimization problem in BMDR is

$$
\begin{aligned}
\min_{\alpha^\top 1_{|D_u|}=b,w} & \sum_{i=1}^{|D_l|}(y_i - w^\top \phi(x_i))^2 + \lambda\|w\|^2 \\
& + \sum_{i=1}^{|D_u|} \alpha_i \left[\|w^\top \phi(x_j)\|_2^2 + 2|w^\top \phi(x_j)|\right] \\
& + \beta(\alpha^\top K_1 \alpha + k\alpha),
\end{aligned}
\tag{1}
$$

where $\phi(x)$ is the feature mapping, $\lambda$ is the hyper-parameter for the regularization term, $\beta$ is the hyper-parameter for the diversity term, $1_{|D_u|}$ means ones vector with length of the unlabelled pool $|D_u|$. $K_1$ is defined as $K_1 = \frac{1}{2}K_{UU}$, where $K_{UU}$ means the kernel matrix with sub-index $U$ of unlabelled pool $D_u$. SPAL only changes $\alpha^\top 1_{|D_u|} = b$ to $\alpha^\top e_{|D_u|} = b$.[15]

## B.6 Comparison between Zhan et al. (2021) and Yang & Loog (2018)

Yang & Loog (2018) propose the first benchmark for pool-based active learning for the conventional Logistic Regression model. The work compares 10 query strategies that could be categorized into **model uncertainty** and **hybrid criteria**. In datasets, they adopt 44 binary datasets and follow data pre-processing in Chang & Lin (2011). To compare performance across different query strategies, they also use an Area Under the Learning Curve with accuracy to show the average performance of a query strategy, named AUBC in Zhan et al. (2021). Furthermore, they compare the performance of each query strategy by average rank and improvement (win/tie/loss) from random sampling, which has the same purpose as our work (See Section 4.1 and Section 4.2).

# C Re-benchmarking results of Zhan et al. (2021)

After we accomplish experiments under the settings in Appendix B.1, we obtain the benchmarking results for RBFSVM with the form (query strategy, dataset, seed, $|D_l|$, accuracy) for each round. A (random) seed corresponds to the different training sets, test sets, and initial label pool splits for a dataset. We collect the accuracy at each round ($|D_l|$, accuracy) to plot a learning curve for query strategy on a dataset with a seed and summarize it as the mean AUBC in Table 19. Our re-benchmarking results show that Uncertainty Sampling with compatible models (US-C) outperforms the other query strategies on most datasets.

## C.1 Statistical comparison of re-benchmarking results

We show our re-benchmarking results for RBFSVM side-by-side with Zhan et al. (2021)'s Table 3 in Table 20. To determine if there is a statistical difference between the two benchmarking results, we construct the confidence interval with the $t$-distribution of mean AUBCs. If a result in Zhan et al. (2021) falls outside the interval, their mean significantly differs from ours. We notice significant differences in Uniform on several datasets in Table 20. Therefore, we focus on comparing Uniform in Table 21, demonstrating our mean and

---

[14]https://github.com/google/active-learning/blob/master/sampling_methods/informative_diverse.py
[15]https://github.com/NUAA-AL/ALiPy/blob/master/alipy/query_strategy/query_labels.py#L1469

Table 19: Mean AUBC of Query Strategies: We report query strategies with mean of repeated experiments.

| | Uniform | US-C | US-NC | InfoDiv | QBC | EER | VR | Hier | Graph | Core-Set | HintSVM | QUIRE | DWUS | MCM | BMDR | SPAL | ALBL | LAL |
|---|---|---|---|---|---|---|---|---|---|---|---|---|---|---|---|---|---|---|
| Appendicitis | 83.95% | 84.54% | 84.49% | 84.54% | 84.41% | 84.26% | 83.95% | 84.14% | 84.19% | 83.98% | 83.90% | 83.99% | 84.21% | 84.57% | 84.18% | 84.15% | 84.49% | 84.33% |
| Sonar | 75.00% | 77.88% | 76.54% | 77.88% | 76.75% | 75.73% | 75.00% | 75.54% | 75.58% | 74.06% | 75.11% | 74.35% | 77.51% | 75.98% | TLE | 76.28% | 76.76% |
| Parkinsons | 83.05% | 85.31% | 85.11% | 85.31% | 84.49% | 84.51% | 83.05% | 83.57% | 82.91% | 83.56% | 81.78% | 83.05% | 82.74% | 85.27% | 83.69% | 83.85% | 84.61% | 84.63% |
| Ex8b | 88.53% | 89.81% | 89.39% | 89.81% | 89.38% | 89.36% | 88.53% | 88.81% | 88.50% | 89.15% | 86.95% | 87.86% | 88.42% | 89.77% | 88.84% | TLE | 88.74% | 89.42% |
| Heart | 80.51% | 81.57% | 81.20% | 81.57% | 81.30% | 80.85% | 80.51% | 80.75% | 80.54% | 81.05% | 80.39% | 81.03% | 80.57% | 81.54% | 80.65% | 80.97% | 81.18% | 81.24% |
| Haberman | 73.09% | 72.99% | 73.02% | 72.99% | 73.05% | 73.14% | 73.08% | 73.01% | 73.04% | 72.67% | 72.62% | 72.46% | 73.16% | 72.92% | 73.23% | TLE | 72.97% | 73.11% |
| Ionosphere | 91.80% | 93.00% | 91.97% | 93.00% | 92.78% | 92.49% | 91.80% | 92.04% | 91.62% | 91.34% | 89.64% | 90.15% | 87.93% | 92.96% | 92.32% | TLE | 92.06% | 92.65% |
| Clean1 | 81.79% | 84.32% | 83.41% | 84.32% | 83.42% | 82.15% | TLE | 81.86% | 81.00% | 79.02% | 76.97% | 81.81% | 81.79% | 84.16% | TLE | TLE | 82.64% | 83.34% |
| Breast | 96.14% | 96.34% | 96.26% | 96.34% | 96.33% | 96.31% | 95.82% | 96.17% | 96.15% | 96.28% | 96.24% | 96.24% | 96.06% | 96.34% | 96.18% | TLE | 96.26% | 95.86% |
| Wdbc | 95.39% | 96.52% | 95.97% | 96.52% | 96.26% | 96.26% | 95.73% | 95.65% | 95.39% | 95.65% | 95.40% | 95.86% | 95.58% | 95.83% | 95.04% | 96.50% | 95.12% | 95.72% | 96.12% | 96.13% |
| Australian | 84.83% | 85.04% | 84.59% | 85.04% | 84.94% | 84.72% | 84.83% | 84.87% | 84.69% | 84.78% | 84.44% | 84.76% | 84.73% | 85.04% | 84.73% | 85.04% | 84.86% | 84.83% |
| Diabetes | 74.24% | 74.79% | 74.32% | 74.79% | 74.72% | 74.57% | 74.24% | 74.34% | 74.24% | 74.91% | 74.56% | 74.70% | 72.27% | 74.71% | 74.23% | 74.65% | 74.43% | 74.62% |
| Mammographic | 81.25% | 81.64% | 81.65% | 81.64% | 81.61% | 81.58% | 81.23% | 81.40% | 81.22% | 81.48% | 80.94% | 81.42% | 79.95% | 81.68% | 81.32% | TLE | 81.59% | 81.39% |
| Ex8a | 85.52% | 88.01% | 82.83% | 88.01% | 86.16% | 85.22% | 85.52% | 86.10% | 85.13% | 85.55% | 81.34% | 80.95% | 79.24% | 87.80% | 85.39% | TLE | 84.19% | 83.54% |
| Tic | 87.18% | 87.20% | 87.18% | 87.20% | 87.19% | 87.18% | 87.18% | 87.19% | 87.19% | 87.20% | 87.16% | 87.19% | 86.99% | 87.10% | 87.20% | 87.19% | 87.20% | 87.20% |
| German | 73.40% | 74.17% | 73.87% | 74.17% | 73.96% | 73.80% | 73.40% | 73.48% | 73.54% | 73.62% | 73.06% | 73.55% | 72.69% | 74.09% | TLE | 73.62% | 73.66% | 74.00% |
| Splice | 80.68% | 82.28% | 81.47% | 82.28% | 81.50% | 80.73% | 80.68% | 80.62% | 78.21% | 74.76% | 77.57% | 80.35% | 76.08% | 82.39% | TLE | TLE | 81.00% | 80.45% |
| Gcloudb | 89.50% | 89.85% | 88.58% | 89.85% | 89.73% | 89.47% | 89.50% | 89.49% | 89.40% | 89.25% | 87.55% | 87.85% | 88.62% | 89.82% | 89.54% | TLE | 89.72% | 89.46% |
| Gcloudub | 94.40% | 95.67% | 94.89% | 95.67% | 95.36% | 93.87% | 94.40% | 94.75% | 94.40% | 89.12% | 89.35% | 93.17% | 93.62% | 95.57% | 93.77% | TLE | 93.83% | 94.76% |
| Checkerboard | 97.81% | 98.47% | 91.34% | 98.47% | 97.02% | 96.26% | 97.81% | 97.81% | 97.85% | 97.37% | 98.74% | 92.42% | 94.37% | 90.45% | 98.47% | 98.32% | TLE | 96.79% | 96.41% |
| Spambase | 91.03% | 92.05% | 90.10% | 92.05% | 91.90% | TLE | TLE | 91.22% | 90.73% | 90.52% | 89.85% | TLE | 91.03% | 92.00% | TLE | TLE | 91.62% | 90.62% |
| Banana | 89.26% | 87.87% | 80.50% | 87.87% | 89.08% | TLE | 89.25% | 89.29% | 88.48% | 89.30% | 85.10% | 82.99% | 81.64% | 87.54% | TLE | TLE | 88.51% | 89.23% |
| Phoneme | 82.54% | 83.55% | 82.11% | 83.55% | 83.18% | TLE | TLE | 83.00% | 82.09% | 82.40% | 80.83% | 81.83% | 81.37% | 83.59% | TLE | TLE | 82.47% | 82.42% |
| Ringnorm | 97.76% | 97.86% | 97.67% | 97.86% | 97.71% | TLE | TLE | 97.66% | 97.11% | 94.77% | 97.15% | TLE | 93.46% | 97.82% | TLE | TLE | 97.69% | 97.80% |
| Twonorm | 97.53% | 97.64% | 97.55% | 97.64% | 97.60% | TLE | TLE | 97.52% | 97.54% | 97.55% | 97.36% | TLE | 97.31% | 97.63% | TLE | TLE | 97.52% | 97.61% |
| Phishing | 93.82% | 94.60% | 93.91% | 94.60% | 94.41% | TLE | TLE | 93.80% | 93.27% | 94.06% | 92.96% | TLE | 89.23% | 94.49% | TLE | TLE | 94.20% | 94.29% |

Table 20: We report our AUBCs (%) with Table 3 in Zhan et al. (2021) side-by-side. A score denoted with format: `Zhan et al. (2021)` → `ours`. The symbol '*' indicates a significant difference with the significance level $\alpha = 5\%$.

| | Uniform | BSO | Avg | BEST | BEST_QS | | WORST | WORST_QS | |
|---|---|---|---|---|---|---|---|---|---|
| Appendicitis | 84% → 83.95% | 88% → 88.37% | 84% → 84.25% | 86% → 84.57%* | EER → | MCM | 83% → 83.90%* | DWUS → | HintSVM |
| Sonar | 62% → 74.63%* | 83% → 88.40%* | 76% → 75.60% | 78% → 77.62%* | LAL → | US-C | 73% → 73.57% | HintSVM → | HintSVM |
| Parkinsons | 84% → 83.05%* | 87% → 88.28%* | 85% → 83.97% | 86% → 85.31%* | QBC → | US-C | 83% → 81.78% | HintSVM → | HintSVM |
| Ex8b | 87% → 88.53%* | 92% → 93.76%* | 89% → 88.88% | 91% → 89.81%* | SPAL → | US-C | 86% → 86.99% | HintSVM → | HintSVM |
| Heart | 81% → 80.51% | 85% → 89.30%* | 79% → 80.99% | 83% → 81.57%* | InfoDiv → | US-C | 72% → 80.39%* | DWUS → | HintSVM |
| Haberman | 73% → 73.08% | 75% → 78.96%* | 73% → 72.95% | 74% → 73.19% | BMDR → | BMDR | 72% → 72.44% | QUIRE → | QUIRE |
| Ionosphere | 90% → 91.80%* | 93% → 95.45%* | 91% → 91.59% | 93% → 93.00%* | LAL → | US-C | 88% → 87.93%* | HintSVM → | DWUS |
| Clean1 | 65% → 81.83%* | 87% → 92.19%* | 81% → 81.97% | 84% → 84.25%* | LAL → | US-C | 75% → 76.95% | HintSVM → | HintSVM |
| Breast | 95% → 96.16%* | 96% → 97.60%* | 96% → 96.19% | 96% → 96.32%* | SPAL → | US-C | 95% → 95.82%* | DWUS → | VR |
| Wdbc | 95% → 95.39% | 97% → 98.41%* | 96% → 95.87% | 97% → 96.52%* | LAL → | US-C | 94% → 95.04%* | EER → | DWUS |
| Australian | 85% → 84.83% | 88% → 90.46%* | 85% → 84.82% | 85% → 84.44%* | Core-Set → | US-C | 82% → 84.44%* | DWUS → | HintSVM |
| Diabetes | 74% → 74.24%* | 78% → 82.57%* | 74% → 74.42% | 75% → 74.91% | Core-Set → | Core-Set | 69% → 72.27%* | EER → | DWUS |
| Mammographic | 82% → 81.30%* | 84% → 85.03%* | 82% → 81.44% | 83% → 81.78% | MCM → | MCM | 80% → 79.99%* | EER → | DWUS |
| Ex8a | 84% → 85.39%* | 87% → 88.28%* | 84% → 84.62% | 86% → 87.88%* | Hier → | US-C | 80% → 79.11%* | QUIRE → | DWUS |
| Tic | 87% → 87.18% | 87% → 90.77%* | 87% → 87.17% | 87% → 87.20%* | EER → | US-C | 87% → 86.99% | QUIRE → | QUIRE |
| German | 73% → 73.40%* | 78% → 82.08%* | 74% → 73.65% | 74% → 74.17% | QBC → | US-C | 72% → 72.68% | DWUS → | DWUS |
| Splice | 81% → 80.75% | 87% → 91.02%* | 79% → 80.08% | 82% → 82.34%* | QBC → | MCM | 68% → 75.18%* | EER → | Core-Set |
| Gcloudb | 89% → 89.52% | 90% → 90.91%* | 89% → 89.20% | 90% → 89.81%* | Graph → | US-C | 87% → 87.48% | HintSVM → | HintSVM |
| Gcloudub | 94% → 94.37% | 96% → 96.83%* | 93% → 93.72% | 95% → 95.60%* | QBC → | US-C | 86% → 89.29%* | EER → | Core-Set |
| Checkerboard | 98% → 97.81% | 99% → 99.72%* | 94% → 96.42% | 99% → 98.74% | Core-Set → | Core-Set | 90% → 90.45%* | VR → | DWUS |
| Spambase | 69% → 91.03%* | TLE | 88% → 91.14% | 92% → 92.05%* | QBC → | US-C | 69% → 89.85%* | DWUS → | HintSVM |
| Banana | 90% → 89.26% | TLE | 85% → 86.90% | 89% → 89.30%* | Hier → | Core-Set | 78% → 80.50%* | QUIRE → | US-NC |
| Phoneme | 82% → 82.54% | TLE | 82% → 82.49% | 83% → 83.59%* | QBC → | MCM | 80% → 80.83% | HintSVM → | HintSVM |
| Ringnorm | 98% → 97.76% | TLE | 95% → 97.05% | 98% → 97.86%* | LAL → | US-C | 80% → 93.46% | DWUS → | DWUS |
| Twonorm | 98% → 97.53% | TLE | 98% → 97.54% | 98% → 97.64%* | Core-Set → | US-C | 97% → 97.31% | DWUS → | DWUS |
| Phishing | 93% → 93.82%* | TLE | 94% → 93.65% | 95% → 94.60%* | LAL → | US-C | 92% → 89.23%* | Graph → | DWUS |

standard deviation (SD) AUBC of Uniform and the mean AUBC of Uniform reported by Zhan et al. (2021). There are 13, nearly half of the datasets, significantly different from the existing benchmark with significance level $\alpha = 5\%$. Furthermore, we perform better on most datasets except for *Parkinsons* and *Mammographic*. 1% of mean AUBC is larger than previous work on 8 datasets, especially for *Sonar*, *Clean1*, and *Spambase*. Following the same procedure of statistical testing, Table 22 demonstrates **BSO** of ours and Zhan et al. (2021). This phenomenon is more evident in BSO than in Uniform. We still get significantly different and better performances on most datasets except for *Appendicitis*.

## C.2 Verify usefulness

Zhan et al. (2021) verified the applicability of a query strategy by several aspects:

Table 21: Reporducing Failure of **Uniform**

|  | Mean | SD | Zhan et al. (2021) | $\alpha = 5\%$ | $\alpha = 1\%$ |
|---|---|---|---|---|---|
| Appendicitis | 83.95% | 3.63% | 83.6% | In | In |
| Sonar | 74.63% | 3.79% | 61.7% | Out | Out |
| Parkinsons | 83.05% | 3.68% | 84.0% | Out | In |
| Ex8b | 88.53% | 2.80% | 86.6% | Out | Out |
| Heart | 80.51% | 2.79% | 80.8% | In | In |
| Haberman | 73.08% | 2.70% | 72.7% | In | In |
| Ionosphere | 91.80% | 1.78% | 90.1% | Out | Out |
| Clean1 | 81.83% | 1.94% | 64.9% | Out | Out |
| Breast | 96.16% | 0.90% | 95.4% | Out | Out |
| Wdbc | 95.39% | 1.30% | 95.2% | In | In |
| Australian | 84.83% | 1.58% | 84.6% | In | In |
| Diabetes | 74.24% | 1.52% | 73.6% | Out | Out |
| Mammographic | 81.30% | 1.98% | 81.9% | Out | Out |
| Ex8a | 85.39% | 2.17% | 83.8% | Out | Out |
| Tic | 87.18% | 1.53% | 87.0% | In | In |
| German | 73.40% | 1.73% | 72.6% | Out | Out |
| Splice | 80.75% | 1.61% | 80.6% | In | In |
| Gcloudb | 89.52% | 1.17% | 89.3% | In | In |
| Gcloudub | 94.37% | 0.96% | 94.2% | In | In |
| Checkerboard | 97.81% | 0.59% | 97.8% | In | In |
| Spambase | 91.03% | 0.57% | 68.5% | Out | Out |
| Banana | 89.26% | 0.38% | 89.5% | In | In |
| Phoneme | 82.54% | 1.01% | 82.2% | In | In |
| Ringnorm | 97.76% | 0.21% | 97.6% | Out | In |
| Twonorm | 97.53% | 0.19% | 97.6% | In | In |
| Phishing | 93.82% | 0.48% | 92.6% | Out | Out |

- *Low/high dimension view* (*LD* for $d < 50$, *HD* for $d \geq 50$),

- *Data scale view* (*SS* for $n < 1000$, *LS* for $n \geq 1000$),

- *Data balance/imbalance view* (*BAL* for $\gamma < 1.5$, *IMB* for $\gamma \geq 1.5$).

They compare these aspects with a score

$$\delta_{q,s} = \max\left\{\overline{\text{AUBC}_{\text{BSO},s}}, \overline{\text{AUBC}_{\text{US},s}}, \ldots, \overline{\text{AUBC}_{\text{LAL},s}}\right\} - \overline{\text{AUBC}_{q,s}},$$

Specifically, they grouped $\delta_{q,s}$ by different aspects to generate the metric for the report

$$\bar{\delta}_{q,v} = \frac{\sum_{s \in v} \delta_{q,s}}{|\{s \in v\}|},$$

where $v$ is one of a dataset's dimension, scale, or class-balance views. We re-benchmark results and denote the rank of the query strategy with a superscript in Table 23. Table 23 shows that the US-C (InfoDiv) and MCM occupy the first and second ranks in different aspects, and the QBC keeps the third rank. The results are unlike those of Zhan et al. (2021) except for the QBC performance well on both of us. We explain the reason for the same performance of US-C and InfoDiv in Appendix B.5.

Using score $\bar{\delta}_{q,v}$ to ascertain the applicability of several query strategies is straightforward. However, it could bring an issue: BSO outperforms query strategies significantly on most datasets in our benchmarking results. We cannot exclude those remaining large-scale datasets without BSO, i.e., $n > 1000$, having the same pattern, such that their results could impact different aspects. Therefore, we replace $\bar{\delta}_{q,v}$ with the improvement of query strategy $q$ over Uniform, i.e., $\tau_{q,s,k}$ in Section 4.2, because Uniform is the baseline and most efficient across all experiments, which is essential to complete.

Table 22: Reporducing Failure of **BSO**

|  | Mean | SD | Zhan et al. (2021) | $\alpha = 5\%$ | $\alpha = 1\%$ |
|---|---|---|---|---|---|
| Appendicitis | 88.37% | 2.95% | 88.1% | In | In |
| Sonar | 88.40% | 2.84% | 83.0% | Out | Out |
| Parkinsons | 88.28% | 3.19% | 86.5% | Out | Out |
| Ex8b | 93.76% | 1.82% | 92.4% | Out | Out |
| Heart | 89.30% | 2.47% | 84.8% | Out | Out |
| Haberman | 78.96% | 3.05% | 75.1% | Out | Out |
| Ionosphere | 95.45% | 1.42% | 93.3% | Out | Out |
| Clean1 | 92.19% | 1.69% | 87.1% | Out | Out |
| Breast | 97.60% | 0.67% | 96.1% | Out | Out |
| Wdbc | 98.41% | 0.65% | 97.3% | Out | Out |
| Australian | 90.46% | 1.48% | 87.8% | Out | Out |
| Diabetes | 82.57% | 1.70% | 78.4% | Out | Out |
| Mammographic | 85.03% | 1.97% | 84.4% | Out | Out |
| Ex8a | 88.28% | 2.03% | 87.3% | Out | Out |
| Tic | 90.77% | 2.27% | 87.3% | Out | Out |
| German | 82.08% | 2.01% | 78.3% | Out | Out |
| Splice | 91.02% | 1.18% | 87.1% | Out | Out |
| Gcloudb | 90.91% | 1.09% | 90.1% | Out | Out |
| Gcloudub | 96.83% | 0.78% | 96.3% | Out | Out |
| Checkerboard | 99.72% | 0.36% | 99.2% | Out | Out |

Table 23: Verifying Applicability with $\delta_i$

|  | B | LD | HD | SS | LS | BAL | IMB |
|---|---|---|---|---|---|---|---|
| US-NC | 4.77 | 4.16 | 8.12 | 5.36 | 3.96 | 5.09 | 4.39 |
| QBC | $3.83^3$ | $3.15^3$ | $7.57^3$ | $5.02^3$ | $2.20^3$ | $4.05^3$ | $3.57^3$ |
| HintSVM | 5.91 | 4.92 | 11.37 | 6.77 | 4.73 | 6.25 | 5.51 |
| QUIRE | 5.96 | 5.08 | 11.54 | 6.13 | 5.60 | 6.94 | 4.98 |
| ALBL | 4.20 | 3.49 | 8.06 | 5.37 | 2.59 | 4.45 | 3.90 |
| DWUS | 6.20 | 5.46 | 10.24 | 6.71 | 5.50 | 6.83 | 5.46 |
| VR | 5.04 | 4.26 | 12.02 | 5.43 | 4.13 | 5.36 | 4.72 |
| Core-Set | 4.92 | 3.78 | 11.20 | 5.79 | 3.72 | 5.35 | 4.42 |
| US-C | $3.50^1$ | $2.89^1$ | $6.86^1$ | $4.62^1$ | $1.97^1$ | $3.72^1$ | $3.24^1$ |
| Graph | 4.62 | 3.72 | 9.58 | 5.77 | 3.05 | 4.98 | 4.20 |
| Hier | 4.22 | 3.41 | 8.69 | 5.53 | 2.43 | 4.49 | 3.90 |
| InfoDiv | $3.50^1$ | $2.89^1$ | $6.86^1$ | $4.62^1$ | $1.97^1$ | $3.72^1$ | $3.24^1$ |
| MCM | $3.56^2$ | $2.94^2$ | $6.98^2$ | $4.68^2$ | $2.03^2$ | $3.80^2$ | $3.27^2$ |
| EER | 5.21 | 4.18 | 11.09 | 5.33 | 4.86 | 6.13 | 4.30 |
| BMDR | 5.61 | 4.57 | 11.50 | 5.77 | 5.11 | 6.33 | 4.89 |
| SPAL | 5.90 | 4.69 | 12.32 | 5.67 | 6.77 | 6.56 | 5.17 |
| LAL | 4.14 | 3.41 | 8.14 | 5.27 | 2.59 | 4.37 | 3.86 |

The other issue is heuristically grouping the views into a binary category and averaging the performance with the same views $\bar{\delta}_{q,v}$ without reporting SDs. These analysis methods may be biased when the properties of datasets are not balanced. To address this issue, we plot a matrix of scatter plots that directly demonstrates the improvement of US-C for each property on all datasets with different colors. Figure 12 shows a low correlation ($|r| < 0.4$) and no apparent patterns between properties and the improvement of US-C, indicating that Our analysis results do not support the claims of 'Method aspects' in the existing benchmark (Zhan et al., 2021), either. In conclusion, we want to emphasize that **revealing the analysis methods is**

Table 24: Mean and standard deviation of AUBCs of query strategies under *no-fixed* and *fixed* initial sets.

|  | Uniform | | US | |
|  | Table 3 | FixInit | Table 3 | FixInit |
|---|---|---|---|---|
| Heart | 78.37%±2.59% | 77.88%±2.58% | 79.32%±2.43% | 78.82%±2.43% |
| Mammographic | 79.46%±1.50% | 79.14%±1.59% | 80.75%±1.39% | 80.53%±1.38% |
| Phoneme | 85.78%±0.56% | 85.66%±0.55% | 87.77%±0.38% | 87.67%±0.37% |

**as important as the experimental settings** because the analysis method employed will influence the conclusion.

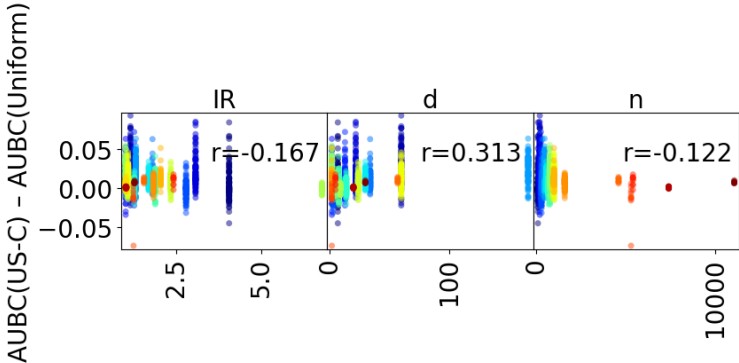

Figure 12: A matrix of scatter plots of the improvement of US-C

## C.3 The impact of initial sets

The current results are based on the train-test split described in Section 3, followed by random sampling to select the initial labeled and unlabeled pools. In this section, we follow Ji et al. (2023)'s recommendation to check the influence of the consistent initial set (labeled pool and unlabeled pool) on Uniform and Uncertainty Sampling. Specifically, we investigate two settings on *Heart*, *Mammographic*, and *Phoneme* datasets:

- No fixed train-test split and no fixed initial sets (Table 3/current results),
- Fixed initial sets and no fixed train-test split (FixInit).

The similar standard deviations between no-fixed (Table 3) and fixed initial sets in Table 24 indicate that fixing the seed of the initial sets across different train-test splits does not lead to significant differences. This is mainly because the primary source of randomness comes from the train-test split itself. Because the datasets in our benchmark do not have predefined train-test splits, we would not eliminate the randomness by fixing the train-test split in our experiments. Although the randomness of the train-test split exists, our benchmarking results show that Uncertainty Sampling performs consistently on most datasets.

## C.4 More on analysis of non-compatible models for uncertainty sampling

Section 5.1 demonstrates results involving different combinations of query-oriented and task-oriented models on *Checkerboard* and *Gcloudb* datasets. We reveal more datasets from Figure 13 to Figure 17. These results still hold for the compatible models for uncertainty sampling outperform non-compatible ones on most datasets, i.e., the diagonal entries of the heatmap are larger than non-diagonal entries. Figure 18 demonstrates that non-compatible models achieve slightly better performance than compatible models. When

query-oriented and task-oriented models are heterogeneous, we conjecture that it could improve uncertainty sampling by exploring more diverse examples like the hybrid criteria approach (Settles, 2012; Sinha et al., 2019).

## D    Benchmarking results of Random Forest

This section follows the analysis procedure in Section 4 to benchmark Random Forest (RF) on the same datasets. The analysis results are listed as follows:

1. Verify the superiority by comparing the mean AUBC of query strategies in Table 25.

2. Verify the superiority by comparing the average accuracy of the model with 20% of the total budget in Table 26.

3. Verify the superiority by comparing the average ranking of query strategies in Table 27.

4. Verify the usefulness by comparing the data utilization rate of query strategies in Table 28.

These results are consistent with the previous benchmarking results in Section 4 and Appendix C. We conclude that the uncertainty sampling with compatible RF models gains superiority and usefulness for tabular datasets.

## E    Computational resource

We test the time of an experiment for query strategy running on a dataset. Our resource is: *DELL PowerEdge R730* with CPU *Intel Xeon E5-2640 v3 @2.6GHz * 2* and memory *192 GB*. We report the computational time for a query strategy for each dataset per round in Table 29 following the setting in Section 3.

Note that this work does not optimize libact, Google, and ALiPy performance. If practitioners discover inefficient implementation, please contact us by mail or leave issues on GitHub.

## F    Limitations, related benchmarks, and future works

While we intentionally constrain our benchmark's scope to maintain fairness and reproducibility, this focus might give the impression of limitations. We encourage practitioners to explore active learning techniques in broader tasks and domains. For example, ample room exists to investigate active learning's applicability in areas like regression problems, object detection, and natural language processing (Cai et al., 2016; Wu et al., 2019; Zhang et al., 2020; Yuan et al., 2021; Brust et al., 2018; Zhang et al., 2022).

Evaluating the performance of query strategy is a challenge in benchmarking. Kottke et al. (2017) and Trittenbach et al. (2021) propose metrics such as *Deficiency score*, *Data Utilization Rate*, *Start Quality*, and *Average End Quality* to summarize the performance of a query strategy from learning curves. Our implementation saves querying results at each round, enabling thorough analysis without costly re-runs, which empowers researchers to develop novel metrics and methods.

The stability of experimental results is another challenge to a fair comparison. Ji et al. (2023); Lüth et al. (2023); Munjal et al. (2022) have revealed variations in performance metrics stemming from different query strategies, causing inconsistent results and claims in previous research. They suggest standardizing experimental settings like data augmentation, neural network structures, and optimizers. These findings emphasize the sensitivity of active learning algorithms to experimental settings, a critical consideration for future work.

Table 25: Benchmarking results of Random Forest. The numbers are mean AUBC (↑, %). We report the baseline method (Uniform), the best query strategy with its mean AUBC (BEST_QS, BEST), and the worst query strategy with its mean AUBC (WORST_QS, WORST) across datasets in Table 25.

|  | Uniform | BEST_QS | BEST | WORST_QS | WORST |
|---|---|---|---|---|---|
| Appendicitis | 83.70% | US | 84.48% | DWUS | 83.12% |
| Sonar | 75.66% | US | 77.31% | HintSVM | 74.75% |
| Parkinsons | 84.31% | US | 86.61% | HintSVM | 82.79% |
| Ex8b | 85.97% | US | 87.07% | HintSVM | 84.67% |
| Heart | 80.19% | DWUS | 80.93% | HintSVM | 79.64% |
| Haberman | 69.56% | US | 70.61% | QUIRE | 68.91% |
| Ionosphere | 90.98% | BALD | 92.08% | HintSVM | 87.22% |
| Clean1 | 79.15% | BALD | 82.09% | HintSVM | 75.18% |
| Breast | 96.42% | US | 96.82% | DWUS | 95.44% |
| Wdbc | 94.29% | LAL | 95.32% | HintSVM | 93.92% |
| Australian | 85.77% | US | 86.20% | DWUS | 85.55% |
| Diabetes | 74.60% | LAL | 74.97% | DWUS | 73.59% |
| Mammographic | 79.36% | LAL | 80.82% | DWUS | 78.82% |
| Ex8a | 93.09% | BALD | 95.50% | HintSVM | 87.65% |
| Tic | 86.36% | Core-Set | 86.43% | DWUS | 85.48% |
| German | 74.02% | US | 74.74% | DWUS | 72.86% |
| Splice | 90.49% | MCM | 91.52% | Core-Set | 84.17% |
| Gcloudb | 88.33% | LAL | 88.96% | QUIRE | 86.50% |
| Gcloudub | 93.83% | BALD | 95.34% | HintSVM | 87.38% |
| Checkerboard | 99.24% | LAL | 99.67% | DWUS | 95.00% |
| Spambase | 93.54% | BALD | 94.74% | HintSVM | 92.11% |
| Banana | 88.25% | LAL | 88.82% | DWUS | 81.54% |
| Phoneme | 86.63% | BALD | 88.81% | HintSVM | 84.75% |
| Ringnorm | 94.15% | US | 95.66% | Core-Set | 70.55% |
| Twonorm | 96.60% | BALD | 96.88% | HintSVM | 94.78% |
| Phishing | 95.61% | US | 96.68% | HintSVM | 94.24% |
| Covertype | 76.47% | US | 79.20% | Uniform | 76.47% |
| Bioresponse | 73.57% | US | 74.83% | Uniform | 73.57% |
| Pol | 96.58% | US | 97.85% | Uniform | 96.58% |

Table 26: Accuracy of the model with 20% labeled examples: We report the accuracy of the model with 20% labeled examples on each dataset. The scores with **bold** mean the best performance on a dataset. 'TLE' means a query strategy exceeds the time limit.

| | Uniform | US | QBC | BALD | Hier | Graph | Core-Set | HintSVM | QUIRE | DWUS | MCM | BMDR | ALBL | LAL |
|---|---|---|---|---|---|---|---|---|---|---|---|---|---|---|
| Sonar | 68.98% | 69.99% | 69.96% | 70.45% | 69.11% | 70.77% | 68.57% | 69.40% | 70.11% | 70.15% | 69.81% | **70.92%** | 70.33% | 70.17% |
| Parkinsons | 81.04% | 81.94% | 80.87% | **82.21%** | 81.56% | 81.04% | 80.47% | 80.54% | 80.53% | 81.05% | 81.97% | 81.41% | 81.24% | 81.94% |
| Ex8b | 82.44% | 83.40% | 82.45% | 83.62% | 82.56% | 82.04% | **83.83%** | 81.85% | 83.04% | 82.61% | 83.20% | 83.43% | 83.18% | 83.33% |
| Heart | 78.37% | **79.42%** | 77.80% | 78.97% | 78.89% | 79.08% | 78.81% | 77.40% | 78.78% | 79.05% | 79.29% | 79.40% | 78.54% | 78.83% |
| Haberman | 70.11% | **71.63%** | 70.24% | 71.58% | 70.67% | 70.79% | 69.34% | 69.57% | 68.76% | 71.47% | 71.41% | 70.49% | 70.39% | 71.28% |
| Ionosphere | 88.65% | 91.34% | 88.42% | **91.91%** | 89.13% | 88.50% | 83.14% | 82.30% | 80.99% | 84.30% | 91.54% | TLE | 88.73% | 90.87% |
| Clean1 | 71.01% | 72.94% | 70.82% | **73.42%** | 71.25% | 71.27% | 66.95% | 66.69% | 71.31% | 70.98% | 73.32% | 68.87% | 72.34% | 72.54% |
| Breast | 96.35% | 97.18% | 96.51% | 97.14% | 96.66% | 96.36% | 95.70% | 96.36% | 95.93% | 95.20% | 97.22% | 96.31% | 96.89% | **97.23%** |
| Wdbc | 93.56% | 96.18% | 93.35% | **96.27%** | 93.55% | 93.62% | 93.12% | 92.69% | 93.28% | 93.46% | 96.07% | 93.85% | 94.98% | 96.18% |
| Australian | 84.97% | **86.17%** | 85.25% | 86.09% | 85.42% | 85.34% | 85.01% | 84.69% | 85.39% | 84.45% | 85.72% | 85.00% | 85.47% | 85.98% |
| Diabetes | 73.84% | 74.13% | 74.07% | **74.39%** | 73.96% | 73.86% | 73.85% | 73.07% | 73.71% | 72.36% | 74.24% | 73.50% | 73.68% | 74.33% |
| Mammographic | 79.74% | **82.46%** | 79.75% | 82.23% | 80.46% | 79.61% | 81.19% | 80.97% | 82.44% | 80.13% | 82.39% | 79.95% | 80.67% | 82.22% |
| Ex8a | 89.91% | **95.05%** | 89.58% | 95.02% | 89.86% | 91.50% | 93.15% | 77.99% | 82.09% | 80.45% | 94.51% | 91.23% | 87.17% | 94.46% |
| Tic | 86.92% | 86.92% | 86.84% | 86.96% | 86.96% | **87.02%** | 86.93% | 86.55% | 86.86% | 84.93% | 86.94% | 86.63% | 86.98% | TLE |
| German | 72.86% | 73.62% | 72.64% | 73.46% | 72.66% | 72.89% | 72.75% | 72.53% | 72.67% | 70.20% | 73.41% | 72.57% | 73.22% | **73.72%** |
| Splice | 87.65% | **88.77%** | 87.41% | 88.57% | 87.76% | 87.17% | 70.04% | 78.95% | 87.15% | 78.31% | 88.76% | 87.25% | 87.53% | 86.36% |
| Gcloudb | 88.27% | 89.25% | 88.52% | 89.14% | 88.45% | 88.69% | 88.52% | 84.99% | 85.68% | 86.68% | 89.29% | 88.48% | 89.25% | **89.58%** |
| Gcloudub | 92.02% | 95.31% | 92.24% | **95.68%** | 93.05% | 93.95% | 90.16% | 84.09% | 86.48% | 83.41% | 95.47% | 91.53% | 89.65% | 93.90% |
| Checkerboard | 99.28% | 99.14% | 99.32% | 99.13% | 99.60% | 99.58% | 99.41% | 93.15% | 93.09% | 93.32% | 99.69% | 99.48% | 99.10% | **99.88%** |
| Spambase | 93.09% | 95.32% | 92.93% | 95.31% | 92.98% | 92.92% | 92.17% | 90.28% | 91.81% | 93.20% | **95.39%** | 92.21% | 92.80% | 95.08% |
| Banana | 87.97% | 89.28% | 88.00% | **89.30%** | 87.92% | 88.04% | 88.43% | 75.08% | 74.97% | 77.28% | 89.25% | 88.23% | 87.27% | 89.16% |
| Phoneme | 84.44% | 88.01% | 84.41% | **88.30%** | 85.67% | 84.77% | 85.51% | 81.39% | 83.14% | 82.61% | 87.89% | 85.12% | 84.46% | 87.55% |
| Ringnorm | 93.73% | **96.91%** | 93.90% | 96.80% | 94.33% | 92.80% | 50.68% | 56.12% | 50.68% | 60.49% | 96.86% | 74.30% | 92.83% | 90.30% |
| Twonorm | 96.47% | 96.83% | 96.52% | **96.89%** | 95.30% | 96.60% | 96.46% | 93.07% | 91.79% | 96.15% | 96.77% | TLE | 96.07% | 96.85% |
| Phishing | 94.83% | **96.85%** | 94.50% | 96.72% | 94.66% | 94.61% | 94.56% | 92.91% | 91.25% | 92.79% | 96.80% | TLE | 95.57% | 96.83% |
| Covertype | 74.78% | 76.91% | TLE | **76.97%** | TLE | TLE | TLE | TLE | TLE | TLE | TLE | TLE | TLE | TLE |
| Bioresponse | 70.63% | **73.03%** | TLE | 72.64% | TLE | TLE | TLE | TLE | TLE | TLE | TLE | TLE | TLE | TLE |
| Pol | 95.94% | **98.23%** | TLE | 98.22% | TLE | TLE | TLE | TLE | TLE | TLE | TLE | TLE | TLE | TLE |

Table 27: Average Ranking of Query Strategies: We report query strategies with the best average ranking. The scores with [1], [2], or [3] mean the 1st, 2nd and 3rd performance on a dataset. 'TLE' means a query strategy exceeds the time limit.

| | US | QBC | BALD | Hier | Graph | Core-Set | HintSVM | QUIRE | DWUS | MCM | BMDR | ALBL | LAL |
|---|---|---|---|---|---|---|---|---|---|---|---|---|---|
| Appendicitis | 4.81[1] | 7.76 | 5.44[2] | 7.30 | 7.74 | 7.31 | 8.43 | 7.89 | 8.85 | 5.63 | 7.81 | 6.47 | 5.56[3] |
| Sonar | 3.69[1] | 6.95 | 3.97[3] | 7.86 | 7.77 | 8.15 | 8.87 | 7.71 | 7.83 | 3.96[2] | TLE | 6.52 | 4.72 |
| Parkinsons | 2.97[1] | 8.55 | 3.08[2] | 6.81 | 10.06 | 9.53 | 11.61 | 9.34 | 7.17 | 3.72[3] | 7.77 | 6.66 | 3.73 |
| Ex8b | 4.50[2] | 8.08 | 4.37[1] | 7.56 | 8.60 | 6.04 | 10.10 | 8.26 | 8.90 | 5.13[3] | 7.28 | 6.82 | 5.36 |
| Heart | 6.06[3] | 7.49 | 7.46 | 7.46 | 8.40 | 7.86 | 9.42 | 8.24 | 4.90[1] | 5.74[2] | 6.63 | 6.20 | |
| Haberman | 5.07[1] | 7.22 | 5.33 | 7.42 | 7.26 | 8.38 | 9.35 | 9.71 | 5.25[2] | 5.29[3] | 7.10 | 7.90 | 5.72 |
| Ionosphere | 2.43[2] | 6.53 | 2.40[1] | 6.29 | 7.27 | 9.96 | 11.50 | 11.55 | 11.25 | 2.90[3] | 8.03 | 7.17 | 3.72 |
| Clean1 | 2.69[2] | 7.76 | 2.44[1] | 7.40 | 9.25 | 9.49 | 11.60 | 7.15 | 7.86 | 2.96[3] | TLE | 5.58 | 3.82 |
| Breast | 3.44[1] | 8.35 | 3.64[3] | 6.75 | 9.60 | 8.36 | 7.66 | 9.05 | 12.24 | 3.60[2] | 8.54 | 5.94 | 3.83 |
| Wdbc | 3.05[3] | 9.30 | 2.80[1] | 8.41 | 9.53 | 9.58 | 10.67 | 9.42 | 9.31 | 3.32 | 8.19 | 4.49 | 2.93[2] |
| Australian | 4.77[2] | 7.48 | 4.58[1] | 6.78 | 8.50 | 8.11 | 8.21 | 7.25 | 8.89 | 5.69 | 8.26 | 6.82 | 5.66[3] |
| Diabetes | 5.53[2] | 6.75 | 5.77[3] | 7.22 | 6.83 | 6.30 | 8.27 | 7.78 | 10.86 | 6.03 | 6.86 | 7.75 | 5.05[1] |
| Mammographic | 3.92[3] | 9.62 | 3.73[2] | 7.66 | 8.23 | 9.06 | 6.96 | 6.74 | 10.10 | 3.92 | 9.11 | 8.54 | 3.41[1] |
| Ex8a | 2.37[1] | 7.93 | 2.39[2] | 7.27 | 7.95 | 4.10 | 12.14 | 11.78 | 11.86 | 3.14[3] | 6.43 | 9.83 | 3.81 |
| Tic | 7.49 | 4.53[2] | 7.66 | 6.58 | 5.02 | 3.89[1] | 6.15 | 7.01 | 9.01 | 7.80 | TLE | 4.82[3] | 8.04 |
| German | 3.80[1] | 7.89 | 4.19[2] | 8.44 | 7.55 | 7.07 | 8.71 | 7.55 | 12.52 | 4.34[3] | 8.21 | 5.67 | 5.06 |
| Splice | 2.72[1] | 6.46 | 2.81[3] | 6.15 | 9.43 | 11.59 | 9.86 | 6.20 | 10.77 | 2.72[2] | TLE | 5.23 | 4.06 |
| Gcloudb | 4.81 | 7.34 | 4.76[3] | 7.65 | 8.25 | 6.73 | 11.28 | 10.51 | 10.25 | 4.90 | 6.76 | 4.53[2] | 3.23[1] |
| Gcloudub | 2.47[2] | 7.34 | 2.46[1] | 5.49 | 6.63 | 8.48 | 12.90 | 11.21 | 11.56 | 2.87[3] | 7.48 | 7.80 | 4.31 |
| Checkerboard | 3.00[2] | 7.67 | 3.25 | 6.47 | 7.49 | 7.20 | 12.18 | 11.67 | 11.81 | 3.15[3] | 5.83 | 8.38 | 2.90[1] |
| Spambase | 2.40[2] | 7.50 | 1.60[1] | 5.50 | 9.10 | 9.30 | 11.00 | TLE | 7.60 | 2.60[3] | TLE | 6.00 | 3.40 |
| BaTLEa | 3.10[3] | 6.90 | 2.90[2] | 7.00 | 9.60 | 5.60 | 11.60 | 12.00 | 12.40 | 3.30 | 6.10 | 8.30 | 2.20[1] |
| Phoneme | 2.40[3] | 8.60 | 1.90[1] | 5.00 | 8.60 | 6.50 | 11.60 | 9.60 | 11.30 | 2.20[2] | TLE | 6.80 | 3.50 |
| Ringnorm | 1.40[1] | 6.00 | 1.80[2] | 4.60 | 8.00 | 11.70 | 9.30 | 11.30 | 9.70 | 3.10[3] | TLE | 5.90 | 5.20 |
| Twonorm | 1.90[2] | 6.20 | 1.80[1] | 9.50 | 5.50 | 4.80 | 10.90 | TLE | 8.00 | 2.30[3] | TLE | 8.80 | 6.30 |
| Phishing | 1.50[1] | 7.00 | 2.30[3] | 5.70 | 7.50 | 5.80 | 9.80 | TLE | 9.20 | 2.20[2] | TLE | 4.00 | TLE |
| Covertype | 1.40[1] | TLE | 1.60[2] | TLE | TLE | TLE | TLE | TLE | TLE | TLE | TLE | TLE | TLE |
| Bioresponse | 1.40[1] | TLE | 1.60[2] | TLE | TLE | TLE | TLE | TLE | TLE | TLE | TLE | TLE | TLE |
| Pol | 1.30[1] | TLE | 1.70[2] | TLE | TLE | TLE | TLE | TLE | TLE | TLE | TLE | TLE | TLE |

Table 28: Data utilization rate of query strategies. The scores with [1], [2], or [3] mean the 1st, 2nd and 3rd performance on a dataset. 'TLE' means a query strategy exceeds the time limit.

| | US | QBC | BALD | Hier | Graph | Core-Set | HintSVM | QUIRE | DWUS | MCM | BMDR | ALBL | LAL |
|---|---|---|---|---|---|---|---|---|---|---|---|---|---|
| Appendicitis | 72.68% | 88.27% | 72.37% | 83.77% | 84.57% | 78.46% | 94.03% | 79.59% | 96.47% | 73.37% | 77.20% | 75.18% | 68.32% |
| Sonar | 83.21% | 96.93% | 79.75% | 103.93% | 105.19% | 109.28% | 113.21% | 100.60% | 98.23% | 82.09% | 93.32% | 94.02% | 84.00% |
| Parkinsons | 66.78% | 104.47% | 65.65% | 89.01% | 115.53% | 113.00% | 125.06% | 109.28% | 90.19% | 71.19% | 83.42% | 90.02% | 71.19% |
| Ex8b | 72.10% | 100.51% | 75.02% | 108.36% | 105.54% | 82.26% | 134.12% | 107.66% | 104.83% | 78.49% | 97.42% | 92.07% | 78.03% |
| Heart | 83.52% | 93.96% | 80.04% | 87.83% | 98.14% | 105.58% | 126.91% | 105.54% | 96.75% | 85.71% | 88.20% | 94.19% | 84.66% |
| Haberman | 108.22% | 166.25% | 86.22% | 127.40% | 117.93% | 155.72% | 194.88% | 160.76% | 110.28% | 84.09% | 108.16% | 131.40% | 94.10% |
| Ionosphere | 71.03% | 109.47% | 70.06% | 112.15% | 118.36% | 184.42% | 204.09% | 190.78% | 266.93% | 75.05% | TLE | 117.39% | 78.14% |
| Clean1 | 66.38% | 101.51% | 68.54% | 98.33% | 113.03% | 105.73% | 125.79% | 98.96% | 99.31% | 67.75% | 104.35% | 86.16% | 75.82% |
| Breast | 56.07% | 91.36% | 58.33% | 83.40% | 161.72% | 92.76% | 94.25% | 90.92% | 342.11% | 58.36% | 124.45% | 78.14% | 59.50% |
| Wdbc | 48.49% | 118.85% | 49.57% | 104.03% | 119.32% | 113.68% | 147.97% | 112.58% | 119.54% | 52.55% | 94.83% | 65.46% | 52.33% |
| Australian | 71.43% | 95.51% | 73.10% | 98.52% | 121.25% | 114.84% | 106.65% | 108.79% | 125.02% | 73.67% | 101.76% | 92.53% | 74.20% |
| Diabetes | 95.33% | 104.14% | 92.27% | 104.10% | 122.35% | 109.48% | 119.51% | 116.73% | 178.03% | 96.93% | 110.47% | 117.40% | 92.93% |
| Mammographic | 72.07% | 153.42% | 71.66% | 89.03% | 91.95% | 81.27% | 103.54% | 87.55% | 128.24% | 64.89% | 101.07% | 93.26% | 59.36% |
| Ex8a | 43.75% | 111.29% | 42.78% | 97.60% | 109.29% | 57.61% | 165.84% | 176.49% | 166.71% | 46.08% | 88.94% | 142.32% | 54.80% |
| Tic | 75.39% | 96.17% | 80.45% | 92.89% | 124.21% | 82.87% | 140.92% | 129.96% | 209.75% | 89.96% | 111.72% | 61.44% | TLE |
| German | 92.08% | 119.64% | 96.81% | 129.11% | 122.34% | 114.60% | 144.97% | 120.44% | 237.03% | 104.75% | 136.86% | 106.77% | 92.01% |
| Splice | 77.98% | 108.12% | 79.25% | 101.68% | 108.61% | 164.32% | 144.63% | 97.44% | 181.65% | 77.97% | 104.14% | 96.84% | 84.31% |
| Gcloudb | 61.60% | 147.10% | 60.36% | 94.06% | 147.55% | 104.00% | 488.17% | 423.89% | 142.77% | 66.29% | 98.02% | 81.17% | 64.59% |
| Gcloudub | 46.41% | 105.08% | 48.27% | 84.01% | 85.21% | 119.68% | 273.89% | 186.53% | 168.58% | 47.57% | 123.12% | 103.64% | 59.03% |
| Checkerboard | 80.42% | 125.73% | 70.49% | 99.35% | 79.21% | 124.08% | 916.17% | 801.92% | 553.50% | 58.66% | 106.13% | 141.07% | 50.63% |
| Spambase | 22.96% | 109.32% | 19.14% | 94.56% | 122.51% | 132.64% | 282.80% | 207.05% | 96.33% | 21.76% | TLE | 104.81% | 25.40% |
| Banana | 65.49% | 116.15% | 47.70% | 131.01% | 132.31% | 83.74% | 455.27% | 396.93% | 691.08% | 56.57% | 122.59% | 194.17% | 52.98% |
| Phoneme | 33.78% | 102.37% | 33.95% | 68.43% | 100.16% | 72.78% | 116.82% | 92.83% | 107.17% | 34.62% | 87.11% | 83.28% | 39.12% |
| Ringnorm | 27.95% | 114.87% | 31.49% | 142.95% | 250.54% | 866.58% | 817.08% | 844.31% | 731.19% | 36.33% | TLE | 208.99% | 124.45% |
| Twonorm | 54.11% | 90.13% | 42.61% | 285.71% | 114.89% | 97.52% | 902.62% | TLE | 114.72% | 58.90% | TLE | 134.42% | 68.25% |
| Phishing | 18.38% | 118.07% | 20.24% | 102.20% | 142.07% | 98.98% | 215.34% | TLE | 151.26% | 20.96% | TLE | 59.51% | 22.77% |
| Covertype | 45.70% | TLE | 46.78% | TLE | 116.98% | TLE | TLE | TLE | 117.86% | TLE | TLE | TLE | TLE |
| Bioresponse | 70.86% | TLE | 72.53% | TLE | TLE | TLE | TLE | TLE | TLE | TLE | TLE | TLE | TLE |
| Pol | 17.78% | TLE | 17.21% | TLE | TLE | TLE | TLE | TLE | TLE | TLE | TLE | TLE | TLE |

Table 29: The computational time of a query strategy (column) on a dataset (row) with format 'minutes:seconds'. 'TLE' denotes a query strategy that exceeds the time limit.

| | Uniform | US | QBC | Hier | Graph | Core-Set | HintSVM | QUIRE | DWUS | MCM | BMDR | ALBL | LAL |
|---|---|---|---|---|---|---|---|---|---|---|---|---|---|
| Appendicitis | 0m3.838s | 0m38.605s | 0m39.556s | 0m6.453s | 0m6.473s | 0m4.556s | 0m5.390s | 0m12.681s | 0m9.665s | 0m4.322s | TLE | 0m42.372s | 9m12.175s |
| Sonar | 0m2.278s | 0m42.757s | 0m59.219s | 0m15.093s | 0m3.278s | 0m5.831s | 0m6.422s | 1m11.645s | 0m9.481s | 0m2.926s | 10m42.063s | 0m51.094s | 6m6.153s |
| Parkinsons | 0m2.102s | 0m41.104s | 0m42.915s | 0m10.972s | 0m2.960s | 0m5.147s | 0m5.414s | 0m47.767s | 0m38.258s | 0m2.556s | 22m19.975s | 0m46.091s | 11m40.118s |
| Ex8b | 0m2.099s | 0m41.264s | 0m40.830s | 0m13.134s | 0m4.657s | 0m4.807s | 0m4.982s | 1m17.010s | 0m9.605s | 0m2.648s | 12m48.409s | 0m44.175s | 3m37.467s |
| Heart | 0m2.496s | 0m44.071s | 0m46.790s | 0m24.228s | 0m5.449s | 0m6.257s | 0m6.740s | 3m27.610s | 0m23.185s | 0m3.434s | 59m9.378s | 0m49.261s | 12m23.746s |
| Haberman | 0m2.612s | 0m44.858s | 0m47.148s | 0m29.947s | 0m5.583s | 0m6.364s | 0m6.531s | 5m14.789s | 0m14.081s | 0m3.412s | 15m49.559s | 0m49.798s | 7m17.263s |
| Ionosphere | 0m3.012s | 0m45.163s | 0m57.073s | 0m55.119s | 0m4.748s | 0m8.249s | 0m9.165s | 7m51.793s | 1m7.518s | 0m4.479s | 30m20.452s | 1m3.266s | 12m52.246s |
| Clean1 | 0m8.529s | 1m39.377s | 3m7.041s | 1m16.212s | 0m10.947s | 0m31.609s | 0m39.084s | 19m34.331s | 0m43.750s | 0m15.876s | TLE | 2m59.617s | 20m54.571s |
| Breast | 0m4.879s | 0m52.967s | 1m15.016s | 2m52.406s | 0m11.266s | 0m15.623s | 0m17.907s | 40m7.140s | 0m56.034s | 0m11.091s | 87m17.190s | 1m30.334s | 60m28.833s |
| Wdbc | 0m4.867s | 0m55.079s | 1m18.105s | 1m42.001s | 0m8.129s | 0m14.407s | 0m22.674s | 31m10.174s | 0m19.239s | 0m9.264s | 160m47.424s | 1m35.925s | 31m4.656s |
| Australian | 0m7.588s | 1m1.552s | 1m38.815s | 2m42.369s | 0m14.618s | 0m22.574s | 0m28.341s | 51m13.865s | 0m24.982s | 0m15.664s | 189m13.127s | 1m41.586s | 55m13.816s |
| Diabetes | 0m9.436s | 1m9.050s | 1m52.802s | 3m10.987s | 0m16.178s | 0m29.073s | 0m33.750s | 66m25.062s | 0m54.033s | 0m20.934s | 112m52.681s | 1m49.799s | 56m45.749s |
| Mammographic | 0m9.026s | 1m6.022s | 1m44.588s | 4m1.753s | 0m17.076s | 0m26.850s | 0m30.192s | 81m58.434s | 0m46.407s | 0m21.149s | 61m52.793s | 1m52.393s | 58m27.473s |
| Ex8a | 0m8.391s | 1m0.039s | 1m30.220s | 2m58.896s | 0m14.736s | 0m22.182s | 0m24.874s | 65m35.275s | 0m37.734s | 0m17.328s | TLE | 1m35.728s | 53m10.547s |
| Tic | 0m12.182s | 1m14.257s | 2m5.724s | 4m20.451s | 0m22.801s | 0m34.233s | 0m47.141s | 106m1.296s | 1m36.144s | 0m31.285s | TLE | 2m16.302s | 81m22.460s |
| German | 0m21.129s | 2m14.906s | 4m56.843s | 5m37.499s | 0m32.877s | 1m4.185s | 1m25.568s | 130m51.383s | 2m40.555s | 0m51.400s | TLE | 4m11.411s | 88m12.394s |
| Splice | 0m30.106s | 3m24.539s | 11m40.330s | 5m35.731s | 0m41.997s | 2m9.672s | 6m32.557s | 114m2.042s | 5m35.220s | 1m21.324s | TLE | 8m19.553s | 88m50.834s |
| Gcloudb | 0m10.540s | 1m13.866s | 1m59.490s | 5m6.187s | 0m21.419s | 0m31.987s | 0m31.938s | 16m13.863s | 1m5.692s | 0m32.361s | 83m49.329s | 2m21.688s | 63m11.067s |
| Gcloudub | 0m8.376s | 1m8.158s | 1m47.249s | 5m23.843s | 0m20.025s | 0m26.050s | 0m31.233s | 16m7.883s | 0m59.193s | 0m26.849s | 85m38.125s | 2m18.006s | 59m32.045s |
| Checkerboard | 0m22.576s | 1m49.740s | 4m28.037s | 14m5.470s | 0m55.399s | 1m7.784s | 1m16.511s | 92m10.770s | 1m51.429s | 1m39.574s | 229m38.358s | 4m22.690s | 191m30.459s |

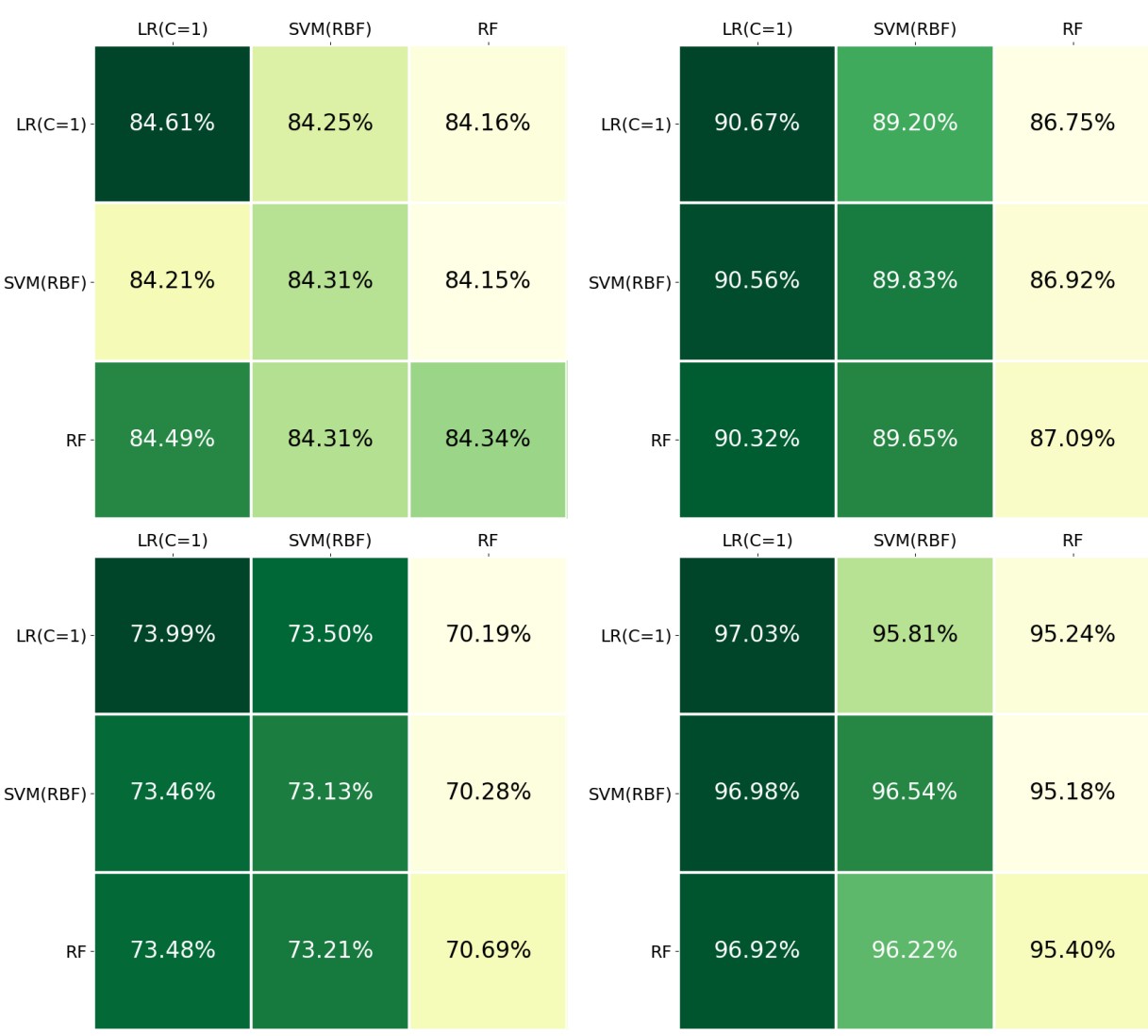

Figure 13: Mean AUBC of query-oriented model and task-oriented model on group 1. (Compatible LRs achieve best results.): *Appendicitis* (top-left), *Ex8b* (top-right), *Haberman* (bottom-left), and *Wdbc* (bottom-right).

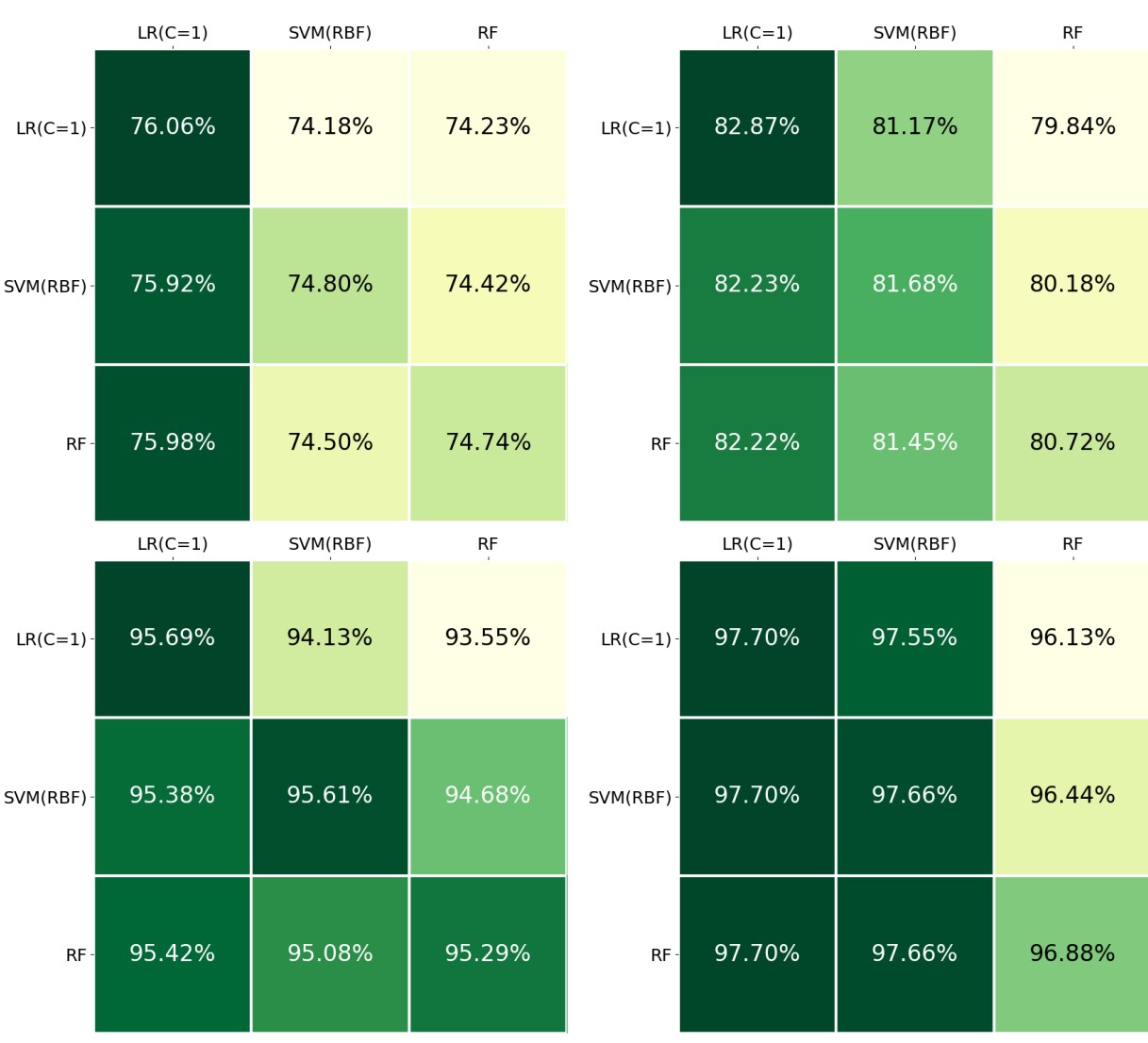

Figure 14: Mean AUBC of query-oriented model and task-oriented model on group 1. (Compatible LRs achieve best results.): *Diabetes* (top-left), *Mammographic* (top-right), *Gcloudub* (bottom-left), and *Twonorm* (bottom-right).

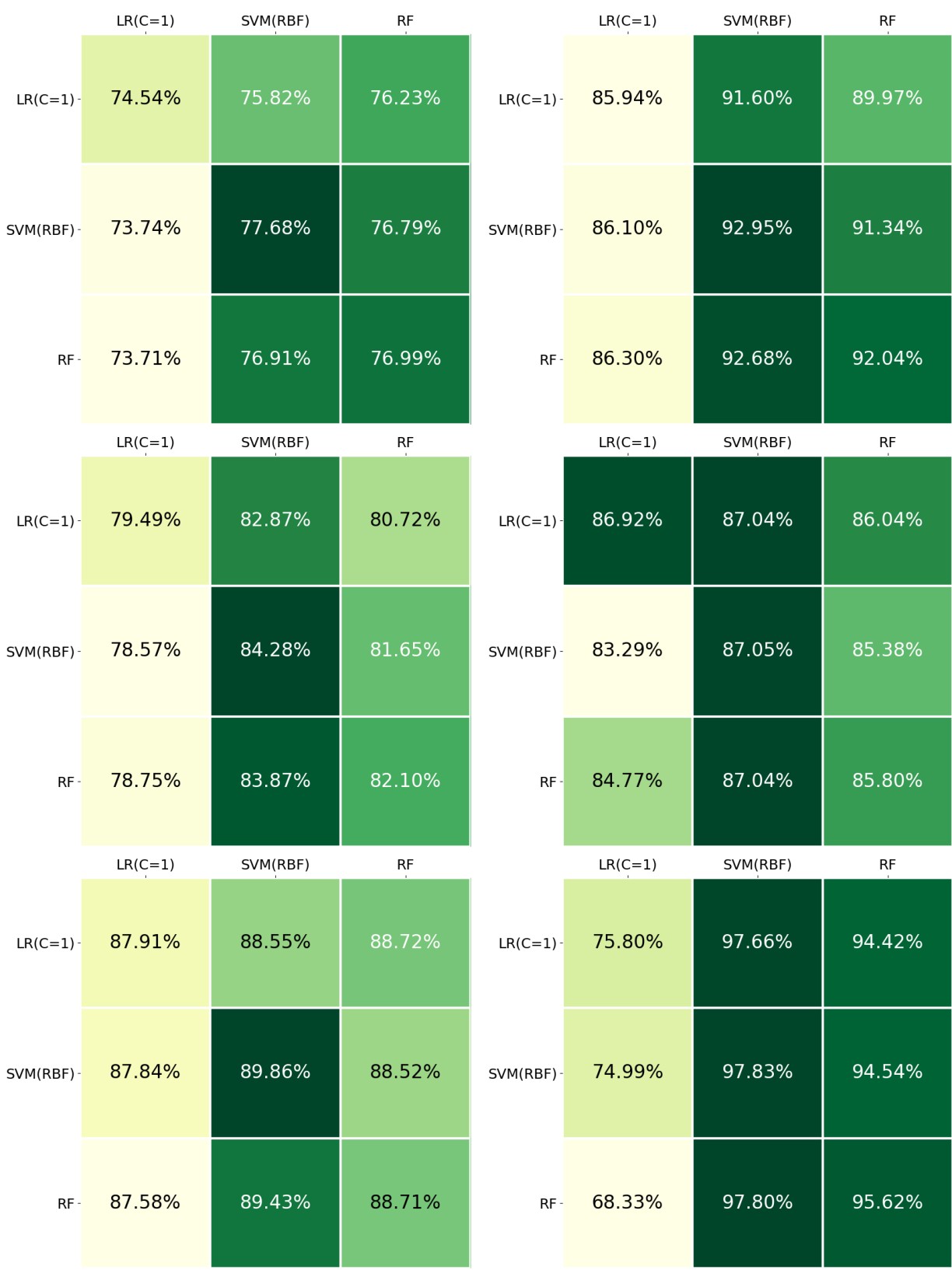

Figure 15: Mean AUBC of query-oriented model and task-oriented model on group 2. (Compatible SVMs achieve best results.): *Sonar* (top-left), *Ionosphere* (top-right), *Clean1* (middle-left), *Tic* (middle-left), *Gcloudb* (bottom-left), and *Ringnorm* (bottom-right).

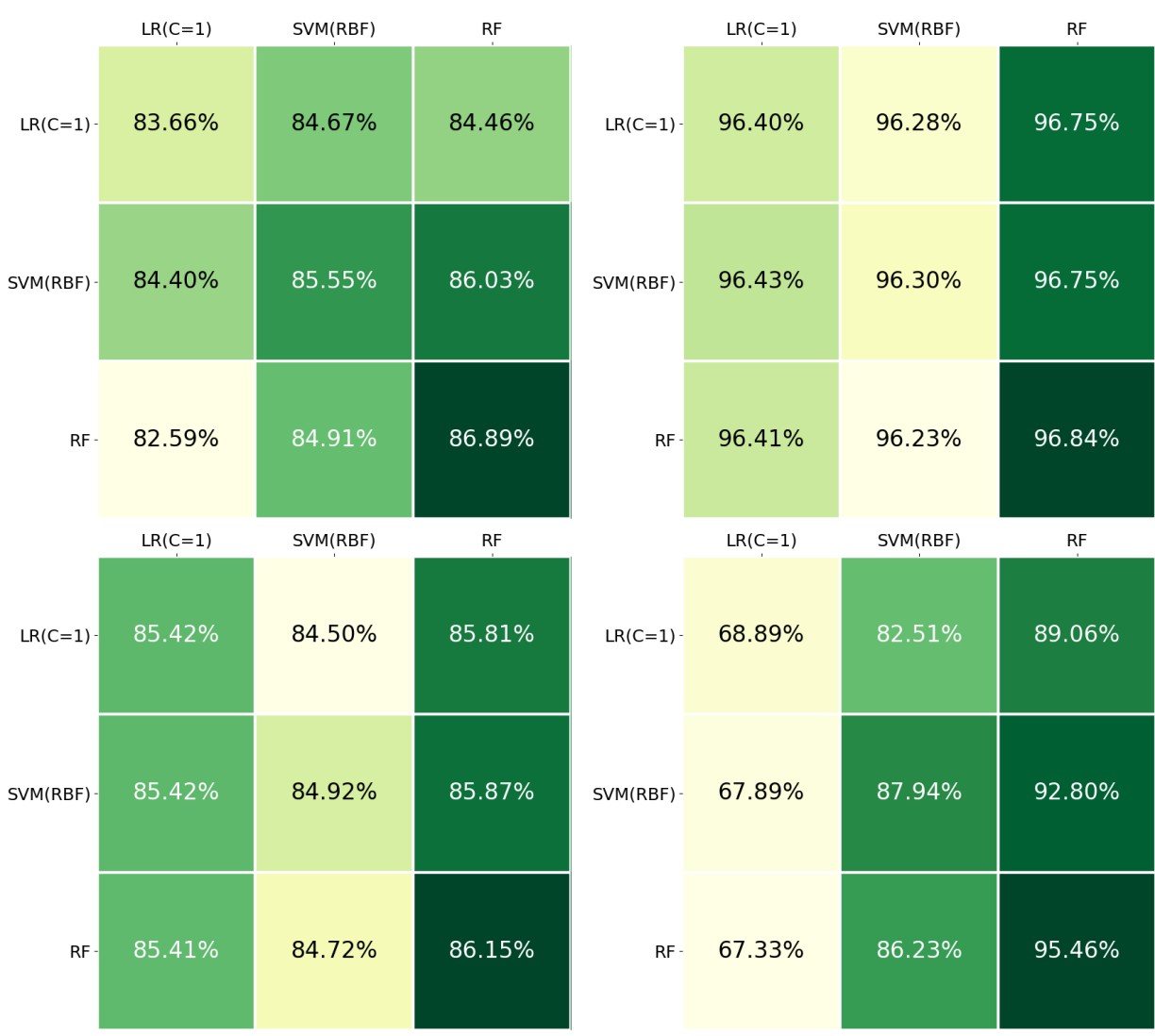

Figure 16: Mean AUBC of query-oriented model and task-oriented model on group 3. (Compatible RFs achieve best results.): *Parkinsons* (top-left), *Breast* (top-right), *Australian* (bottom-left), and *Ex8a* (bottom-right).

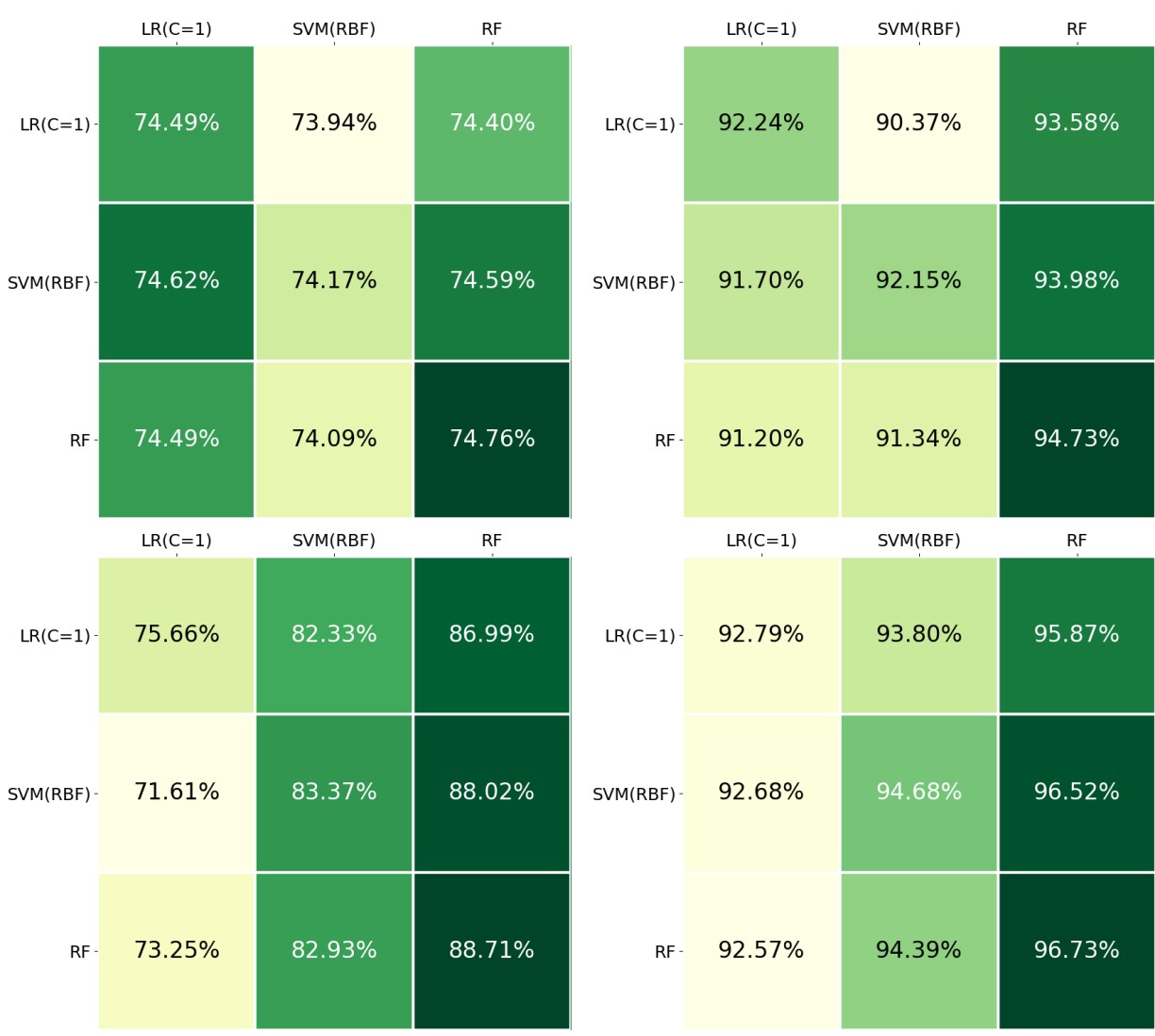

Figure 17: Mean AUBC of query-oriented model and task-oriented model on group 3. (Compatible RFs achieve best results.): *German* (top-left), *Spambase* (top-right), *Phoneme* (bottom-left), and *Phishing* (bottom-right).

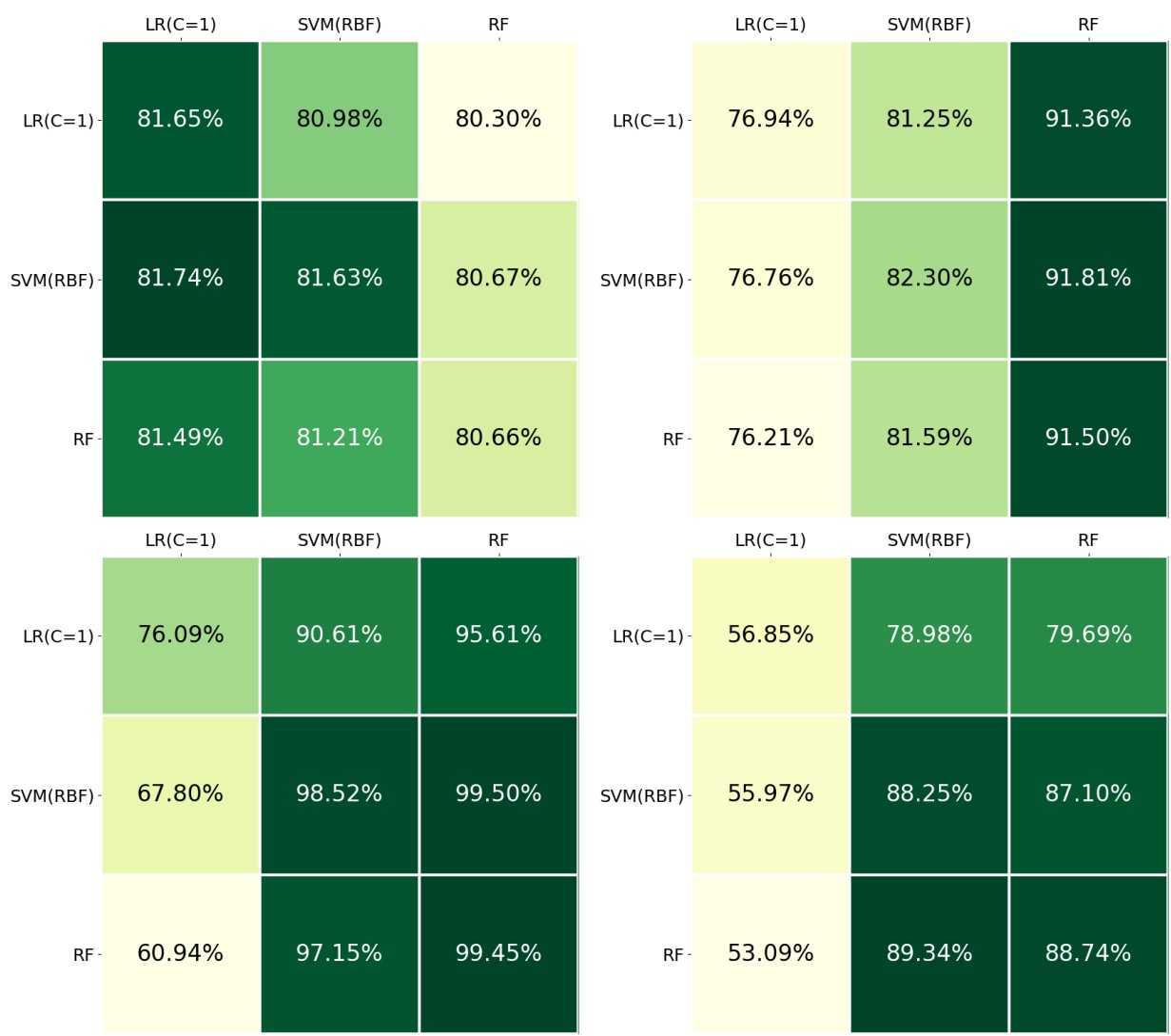

Figure 18: Mean AUBC of query-oriented model and task-oriented model on group 5. (Non-Compatible models achieve best results.): *Heart* (top-left), *Splice* (top-right), *Checkerboard* (bottom-left), and *Banana* (bottom-right).

