# OpenReview forum: "An Expanded Benchmark that Rediscovers and Affirms the Edge of Uncertainty Sampling for Active Learning in Tabular Datasets"
_TMLR — Accepted by TMLR_

### Review · Reviewer_V5HM · 2024-11-14

**Summary Of Contributions:**

This paper provides a comprehensive review and benchmarking of **Active Learning (AL)** strategies, focusing on the widely used **Uncertainty Sampling (US)** method. The key contributions and new knowledge introduced by the paper include:

1. **Expanded Open-Source Benchmark**:
   - The authors present the **most comprehensive open-source AL benchmark** to date, surpassing previous benchmarks in terms of the number of datasets, models, query strategies, and analysis tools.
   - The benchmark unifies previous comparisons by integrating a **transparent and consistent experimental protocol**, allowing for **reproducible evaluation** across various AL strategies.

2. **Uncovering the Role of Model Compatibility**:
   - The study investigates the often-overlooked issue of **model compatibility** between the query-oriented model (used for selecting samples) and the task-oriented model (used for predictions).
   - The authors demonstrate that **Uncertainty Sampling (US)** remains competitive when paired with compatible models, clarifying conflicting conclusions found in previous benchmarks.
   - The paper introduces the terms **US-Compatible (US-C)** and **US-Non-Compatible (US-NC)** to differentiate between these scenarios, showing that **US-C** consistently outperforms **US-NC**.

3. **Comprehensive Analysis of AL Strategies**:
   - The benchmark includes a broader range of AL query strategies (e.g., **US**, **QBC**, **BALD**, **Core-Set**), covering **traditional machine learning models** (Logistic Regression, RBFSVM, Random Forest) and **tree-based models** (XGBoost).
   - The analysis highlights the **robust performance of US**, especially in scenarios involving **binary classification on tabular datasets**, with US achieving the highest **Area Under the Budget Curve (AUBC)** on most datasets.

4. **New Insights and Practical Recommendations**:
   - The study provides fresh insights into the **effectiveness and limitations** of AL strategies, demonstrating that US is not only a strong baseline but also a preferred choice when used with compatible settings.
   - The findings suggest practical guidelines for AL practitioners, emphasizing the importance of model compatibility and offering a **clear recipe for selecting AL strategies** in real-world applications.

5. **Open-Source Framework for Future Research**:
   - The benchmark framework is designed to be extensible, allowing future research to include **diverse domains** (e.g., vision, language) and **advanced models** (e.g., deep neural networks).
   - The open-source nature of the benchmark facilitates collaboration and standardizes the evaluation of AL strategies, addressing the **need for transparent and fair comparisons**.

In summary, the paper reaffirms the competitiveness of Uncertainty Sampling, resolves inconsistencies in previous benchmarks, and establishes a strong foundation for future studies in Active Learning. The new benchmark provides a **practical, reproducible, and comprehensive tool** for evaluating and selecting AL strategies.

**Audience:**

Yes

**Claims And Evidence:**

Yes

**Requested Changes:**

**Critical Adjustments**

1. **Expand the Scope of Datasets**:
   - **Issue**: The current benchmark predominantly focuses on tabular datasets and binary classification tasks.
   - **Proposed Adjustment**: Extend the evaluation to include more diverse datasets, such as **multi-class**, **imbalanced datasets**, and **domain-specific data** (e.g., text, image data). This will demonstrate the **generalizability** of the benchmark across a broader range of applications and enhance the validity of the findings.
   - **Justification**: Expanding the dataset scope is crucial for showcasing the real-world applicability of the proposed benchmark and avoiding limitations in generalizability.

2. **Include Analysis with Deep Learning Models**:
   - **Issue**: The benchmark relies heavily on traditional machine learning models like Logistic Regression and Random Forest, which may not reflect modern usage trends in machine learning.
   - **Proposed Adjustment**: Incorporate **deep learning models**, such as neural networks used in NLP (e.g., BERT, LSTM) or computer vision (e.g., ResNet, CNNs), to evaluate the performance of Active Learning (AL) strategies in **deep learning scenarios**.
   - **Justification**: This addition is critical for aligning the benchmark with current industry and research practices, where deep learning models are widely adopted. It will also address a gap in the analysis and increase the impact of the work.

3. **Provide Detailed Hyperparameter Analysis**:
   - **Issue**: The current submission does not address the impact of **hyperparameters** on the performance of AL strategies.
   - **Proposed Adjustment**: Include an analysis of **hyperparameter sensitivity**, particularly focusing on key parameters like **query batch size**, **model complexity**, and **sampling thresholds**. This could involve experiments where these hyperparameters are systematically varied to observe their effects.
   - **Justification**: Hyperparameter choices can significantly affect the outcomes of AL experiments. Addressing this sensitivity is critical for ensuring the robustness of the benchmark results and guiding practitioners in setting up their experiments.

4. **Clarify the Limitations of Uncertainty Sampling**:
   - **Issue**: The paper heavily emphasizes the advantages of Uncertainty Sampling (US) without adequately discussing its **limitations**.
   - **Proposed Adjustment**: Provide a section that explicitly outlines the **potential drawbacks** of US, such as sensitivity to **outliers**, sampling bias, and limitations in cases of **data drift** or **distribution shifts**.
   - **Justification**: A balanced discussion of the strengths and weaknesses of US is essential for a fair evaluation of AL strategies and for helping readers understand scenarios where alternative methods might be more suitable.

**Enhancements That Would Strengthen the Work**

5. **Improve Documentation and Reproducibility**:
   - **Issue**: The complexity of the experimental setup, including preprocessing and model compatibility settings, may pose challenges for other researchers attempting to reproduce the results.
   - **Proposed Adjustment**: Provide **detailed documentation** and **tutorials**, including clear instructions for setting up experiments, code examples, and a description of key dependencies. Consider releasing **pre-trained models** and **configuration files** for ease of replication.
   - **Justification**: While not critical for acceptance, this improvement would significantly enhance the usability and impact of the benchmark, making it a more valuable resource for the community.

6. **Incorporate State-of-the-Art AL Strategies**:
   - **Issue**: The benchmark includes many well-known AL strategies but lacks a comparison with some **recent advancements**, such as BADGE, LPL, and WAAL, which have shown promising results in specific scenarios.
   - **Proposed Adjustment**: Extend the benchmark to include a comparison with these **state-of-the-art AL methods**, particularly focusing on their performance in scenarios where traditional methods like US might fail (e.g., high-dimensional feature spaces, non-i.i.d data).
   - **Justification**: This adjustment would provide a more comprehensive evaluation and ensure the benchmark remains relevant as the field of AL evolves.

7. **Expand Discussion on Broader Impacts and Future Work**:
   - **Issue**: The current broader impact statement focuses primarily on the benefits to the AL community but does not address potential risks or ethical considerations.
   - **Proposed Adjustment**: Include a more thorough discussion on the **ethical implications** of using AL strategies, particularly in sensitive applications (e.g., medical data annotation, legal text analysis). Additionally, provide more specific directions for **future work**, such as evaluating AL strategies in **low-resource settings** or integrating with **semi-supervised learning** approaches.
   - **Justification**: This enhancement would strengthen the broader impact statement and demonstrate a proactive approach to ethical considerations, which is increasingly valued in machine learning research.

8. **Visualize Model Compatibility Impact More Clearly**:
   - **Issue**: While the paper discusses the importance of model compatibility, the visualizations (e.g., decision boundary illustrations) could be made clearer.
   - **Proposed Adjustment**: Improve the **quality and clarity of visual aids**, using well-labeled plots that highlight the differences in performance between US-Compatible (US-C) and US-Non-Compatible (US-NC) settings across various datasets and models.
   - **Justification**: Enhanced visual representations would make the key insights more accessible and help reinforce the importance of model compatibility in a more digestible format.

**Strengths And Weaknesses:**

1. **Comprehensive Benchmark Design**:
   - The paper offers a **well-structured, comprehensive benchmark**, covering a wide range of datasets, query strategies, and base models. This extensive coverage sets a **new standard for evaluating Active Learning (AL)** techniques.
   - The **transparent and reproducible** open-source framework is a significant contribution, addressing a major gap in previous AL research by enabling standardized comparisons across different settings.

2. **Clarification of Model Compatibility**:
   - The authors provide an insightful analysis on the importance of **model compatibility** between query-oriented and task-oriented models, introducing the concepts of **US-Compatible (US-C)** and **US-Non-Compatible (US-NC)**.
   - This analysis resolves conflicting conclusions in prior benchmarks and offers a **clear and practical recommendation** for practitioners, enhancing the understanding of the conditions under which **Uncertainty Sampling (US)** excels.

3. **Robust Experimental Protocol**:
   - The submission features a **well-defined experimental protocol**, including detailed preprocessing steps, dataset configurations, and evaluation metrics such as **Area Under the Budget Curve (AUBC)**.
   - The use of multiple **base models** (e.g., Logistic Regression, RBFSVM, Random Forest, XGBoost) demonstrates the generalizability of the benchmark results across different machine learning paradigms.

4. **Practical Relevance and Extensibility**:
   - The findings are **highly relevant for real-world applications**, providing actionable insights for AL practitioners on when and how to apply **Uncertainty Sampling (US)** effectively.
   - The open-source nature of the benchmark and its **modular design** make it easy to extend, encouraging further exploration in areas like **deep learning models** and **domain-specific AL tasks** (e.g., vision, language).

5. **High-Quality Analysis and Visualization**:
   - The paper includes a thorough analysis of the benchmarking results, with **clear visualizations** (e.g., learning curves, mean difference plots) that effectively convey the performance trends of different AL strategies.
   - The use of **statistical tests**, such as the Friedman test, adds rigor to the performance comparisons, providing evidence for the significance of the results.

### Weaker Elements and Areas for Improvement

1. **Limited Scope of Datasets**:
   - Although the benchmark includes a variety of tabular datasets, it **lacks diversity** in terms of **domain-specific datasets** (e.g., text, image data). Incorporating **deep learning scenarios** and **more complex data types** could strengthen the generalizability of the findings.
   - The focus on **binary classification** may limit the applicability of the benchmark to more **complex, multi-class** or **imbalanced** dataset scenarios.

2. **Dependency on Traditional Machine Learning Models**:
   - The analysis heavily relies on **traditional machine learning models** (e.g., Logistic Regression, Random Forest). While these models are effective for tabular data, the relevance of the results to **deep learning models** used in modern applications (e.g., neural networks in NLP or CV) remains unexplored.
   - The paper could benefit from evaluating AL strategies on **deep neural networks**, especially given the increasing adoption of these models in industry and academia.

3. **Overemphasis on Uncertainty Sampling**:
   - While the focus on **Uncertainty Sampling (US)** is justified by its widespread use, the submission might have **overemphasized its advantages**. Additional analysis comparing **state-of-the-art AL strategies** (e.g., BADGE, LPL, WAAL) across different model types could provide a more balanced view.
   - The paper could include a discussion on the **limitations** of US, such as its potential biases and sensitivity to **outliers**, and suggest scenarios where alternative strategies might be more effective.

4. **Limited Exploration of Hyperparameter Sensitivity**:
   - The benchmark does not thoroughly address the **impact of hyperparameter choices** on the performance of different query strategies and base models. Given that hyperparameters can significantly influence AL outcomes, this could be an important area for further investigation.
   - Including an analysis of **sensitivity to key hyperparameters** (e.g., query batch size, model complexity) would add depth to the findings and help practitioners better understand the robustness of each strategy.

5. **Potential Issues with Reproducibility**:
   - While the framework aims for reproducibility, the **complexity of the experimental setup** (e.g., model compatibility settings, preprocessing choices) might present challenges for replication by other researchers.
   - Providing **detailed documentation** and **examples** for setting up the experiments, as well as sharing **pre-trained models** and configurations, could help address this concern.

---

> ### Author Response · Authors · 2025-03-19
>
> We thank you for your detailed and thoughtful feedback. Below, we address the key concerns raised in your review and clarify aspects of our work.
>
> ### Critical Adjustments
>
> > Expand the Scope of Datasets:
> > Issue: The current benchmark predominantly focuses on tabular datasets and binary classification tasks.
> > Proposed Adjustment: Extend the evaluation to include more diverse datasets, such as multi-class, imbalanced datasets, and domain-specific data (e.g., text, image data). This will demonstrate the generalizability of the benchmark across a broader range of applications and enhance the validity of the findings.
> > Justification: Expanding the dataset scope is crucial for showcasing the real-world applicability of the proposed benchmark and avoiding limitations in generalizability.
>
> As mentioned in General Response, we scope our benchmark on tabular datasets and extend the evaluation to include multi-class in Section 4.3.1, domain-specific (text, image) datasets in Section 4.3.2, and the one-shot protocol for the imbalanced datasets in Section 5.3.1.
>
> > Include Analysis with Deep Learning Models:
> > Issue: The benchmark relies heavily on traditional machine learning models like Logistic Regression and Random Forest, which may not reflect modern usage trends in machine learning.
> > Proposed Adjustment: Incorporate deep learning models, such as neural networks used in NLP (e.g., BERT, LSTM) or computer vision (e.g., ResNet, CNNs), to evaluate the performance of Active Learning (AL) strategies in deep learning scenarios.
> > Justification: This addition is critical for aligning the benchmark with current industry and research practices, where deep learning models are widely adopted. It will also address a gap in the analysis and increase the impact of the work.
>
> In Section 4.3.2, we initialize the preliminary investigation on active learning for domain-specific scenarios by utilizing the feature extractors to convert the data into tabular datasets. We then verify the feasibility and usefulness of Uncertainty Sampling for this approach to connect our work with modern machine learning usage.
> Besides, Section 2.1 adds more deep active learning benchmarks to reveal the ongoing and future works on active learning for modern machine learning.
>
> > Provide Detailed Hyperparameter Analysis:
> > Issue: The current submission does not address the impact of hyperparameters on the performance of AL strategies.
> > Proposed Adjustment: Include an analysis of hyperparameter sensitivity, particularly focusing on key parameters like query batch size, model complexity, and sampling thresholds. This could involve experiments where these hyperparameters are systematically varied to observe their effects.
> > Justification: Hyperparameter choices can significantly affect the outcomes of AL experiments. Addressing this sensitivity is critical for ensuring the robustness of the benchmark results and guiding practitioners in setting up their experiments.
>
> We investigate the sensitivity of active learning protocols for uncertainty sampling in Section 5.3.
> Here is summarize of our observations:
> 1. We conduct the experiments of imbalanced datasets with the one-shot protocol, i.e., each class has one example at the beginning of the initial labeled pool. Our experimental results show that imbalanced data still challenges existing AL algorithms.
> 2. Use compatible XGBoost models (task-oriented and query-oriented models) that are stable and the effect of hyper-parameters (changing of the model complexity of XGBoost) is negligible.
> 3. There is a trade-off between increasing the query batch size and the performance (AUBC) of US. While several works have made an effort to design batch-mode query strategies such as BMDR, SPAL mentioned in Section 2, most methods suffer from the computational cost and only bring small benefits compared to Uniform. We suggest that researchers balance the trade-off between the improvement over Uniform and the computational cost of their design for future works.

---

> > ### Author Response · Authors · 2025-03-19
> >
> > ### Critical Adjustments (Continue)
> >
> > > Clarify the Limitations of Uncertainty Sampling:
> > > Issue: The paper heavily emphasizes the advantages of Uncertainty Sampling (US) without adequately discussing its limitations.
> > > Proposed Adjustment: Provide a section that explicitly outlines the potential drawbacks of US, such as sensitivity to outliers, sampling bias, and limitations in cases of data drift or distribution shifts.
> > > Justification: A balanced discussion of the strengths and weaknesses of US is essential for a fair evaluation of AL strategies and for helping readers understand scenarios where alternative methods might be more suitable.
> >
> > We summarize the limitations of Uncertainty Sampling in Section 5.4.
> > 1. It is unclear that Uncertainty Sampling does not excel on datasets like Checkerboard and Banana. Besides studying the reason for the ineffectiveness of Uncertainty Sampling, we suggest that researchers explore robust baselines for pool-based active learning.
> > 2. The challenge of designing the batch-mode Uncertainty Sampling and utilizing Uncertainty Sampling for domain-specific and imbalanced data still exists.
> >
> > Although Uncertainty Sampling has limitations, our protocol could help researchers explore its edge for more scenarios in future works.

---

> ### Author Response · Authors · 2025-03-19
>
> ### Enhancements That Would Strengthen the Work
>
> > Improve Documentation and Reproducibility:
> > Issue: The complexity of the experimental setup, including preprocessing and model compatibility settings, may pose challenges for other researchers attempting to reproduce the results.
> > Proposed Adjustment: Provide detailed documentation and tutorials, including clear instructions for setting up experiments, code examples, and a description of key dependencies. Consider releasing pre-trained models and configuration files for ease of replication.
> > Justification: While not critical for acceptance, this improvement would significantly enhance the usability and impact of the benchmark, making it a more valuable resource for the community.
>
> Thanks for your suggestions. We will continually update the README and tutorial for the codebase.
>
> > Incorporate State-of-the-Art AL Strategies:
> > Issue: The benchmark includes many well-known AL strategies but lacks a comparison with some recent advancements, such as BADGE, LPL, and WAAL, which have shown promising results in specific scenarios.
> > Proposed Adjustment: Extend the benchmark to include a comparison with these state-of-the-art AL methods, particularly focusing on their performance in scenarios where traditional methods like US might fail (e.g., high-dimensional feature spaces, non-i.i.d data).
> > Justification: This adjustment would provide a more comprehensive evaluation and ensure the benchmark remains relevant as the field of AL evolves.
>
> Because BADGE, LPL, and WAAL are mainly designed for deep learning-based models and not implemented in existing GitHub codebases (Google AL Playground, libact, ALiPy, ModAL, and scikit-activeml), we position our work on query strategies designed for tabular data and leave these comparisons in future work.
>
> > Expand Discussion on Broader Impacts and Future Work:
> > Issue: The current broader impact statement focuses primarily on the benefits to the AL community but does not address potential risks or ethical considerations.
> > Proposed Adjustment: Include a more thorough discussion on the ethical implications of using AL strategies, particularly in sensitive applications (e.g., medical data annotation, legal text analysis). Additionally, provide more specific directions for future work, such as evaluating AL strategies in low-resource settings or integrating with semi-supervised learning approaches.
> > Justification: This enhancement would strengthen the broader impact statement and demonstrate a proactive approach to ethical considerations, which is increasingly valued in machine learning research.
>
> We add discussions on the AL for present-day machine learning issues such as *fairness* and *privacy* and trust our AL protocol and codebase could help researchers start frontier works.
>
> > Visualize Model Compatibility Impact More Clearly:
> > Issue: While the paper discusses the importance of model compatibility, the visualizations (e.g., decision boundary illustrations) could be made clearer.
> > Proposed Adjustment: Improve the quality and clarity of visual aids, using well-labeled plots that highlight the differences in performance between US-Compatible (US-C) and US-Non-Compatible (US-NC) settings across various datasets and models.
> > Justification: Enhanced visual representations would make the key insights more accessible and help reinforce the importance of model compatibility in a more digestible format.
>
> Thanks for your suggestions. Because most datasets are higher than two dimensions, dimension reduction is required to obtain visual representations of decision boundaries, as we illustrate in Figure 5. We will attempt to improve our analysis methods and visualizations to study the influence of model compatibility in the future.

---

### Review · Reviewer_9t1Y · 2025-02-03

**Summary Of Contributions:**

This paper presents an expanded benchmark for Active Learning (AL) that aims to resolve conflicting conclusions in previous AL benchmarks regarding the effectiveness of Uncertainty Sampling (US). The primary contributions of the paper are as follows:

**Comprehensive AL Benchmark**
- The authors develop the most extensive AL benchmark to date, covering 29 datasets, 12 query strategies, and multiple base models (Logistic Regression, SVMs, Random Forest, and Gradient Boosting Decision Trees).
- The benchmark surpasses Yang & Loog (2018) and Zhan et al. (2021) by offering a broader evaluation framework, including new datasets, models, and query strategies.
- The authors provide a transparent, open-source implementation to ensure reproducibility and enable further research.

**Findings on Uncertainty Sampling (US) Performance**
- The study provides clear guidelines for AL practitioners, recommending US as the default choice when paired with a compatible model.

**Methodological Contributions**
- The paper introduces a new experimental protocol for benchmarking AL strategies fairly, ensuring that query selection and model evaluation are properly aligned.
- The authors analyze AL performance using AUBC (Area Under the Budget Curve), average ranking statistics, and accuracy comparisons at different query budgets.

This work improves the empirical evaluation of AL strategies by addressing benchmarking inconsistencies, introducing a more rigorous experimental protocol, and providing open-source tools. The insights on model compatibility have major implications for how AL strategies should be compared and deployed in practice.

**Audience:**

Yes

**Claims And Evidence:**

No

**Requested Changes:**

**Critical Revisions (Necessary for Acceptance)**

- Clarify the Scope of the Benchmark in the Title and Abstract: The current title and abstract suggest that the benchmark is broadly applicable to active learning, but the study is limited to binary classification, tabular data and classical ML models. This can lead to misinterpretation of its generalizability.

**Recommended Improvements (Not Essential for Acceptance, but Strengthen the Paper)**
- Expand the Literature Review to Include Deep Learning-Based AL Methods
- Include Computational Efficiency Comparisons
- Consider Evaluating AL on More Challenging Scenarios (Imbalanced datasets)

**Strengths And Weaknesses:**

### **Strengths**

**Comprehensive and Rigorous Benchmark**
- The paper presents one of the most extensive Active Learning (AL) benchmarks to date, surpassing prior works in:
  - Dataset variety: 29 datasets covering different domains and scales
  - Model diversity: Including Logistic Regression (LR), RBF Support Vector Machines (RBFSVM), Random Forests (RF), and Gradient Boosting Decision Trees (GBDT)
- Query strategy coverage: Evaluating 12 different AL strategies systematically
- It systematically evaluates 12 AL strategies across 29 datasets, ensuring broad applicability.
- The benchmark includes standardized protocols and open-source tools, which enhance reproducibility and facilitate further research.

**Resolution of Conflicting Findings in AL Research**

- The authors identify and address a key flaw in previous benchmarks: model incompatibility between query selection and task-oriented models. Specifically, they distinguish between:
  - Query-oriented models (H): Used to select which examples to label
  - Task-oriented models (G): Used to make final predictions on test data

By controlling for this compatibility factor and showing that mismatches between H and G can significantly degrade performance, they provide a more accurate assessment of Uncertainty Sampling (US) and clarify why prior studies reached contradictory conclusions.


**Empirical and Statistical Rigor**
- The study employs multiple evaluation metrics and thorough statistical analysis:
  - Comprehensive performance comparisons across datasets (Tables 3-4)
  - Detailed ranking analysis showing strategy performance (Table 5)
  - Data utilization rate analysis (Table 6)
- Statistical significance testing using Friedman tests (p < 0.05) to validate that performance differences between strategies are not due to chance
- The study employs multiple evaluation metrics, including Area Under the Budget Curve (AUBC) and average ranking statistics.
- It includes **statistical significance testing (e.g., Friedman tests)** to validate claims.
- The benchmark results are thoroughly documented, with clear performance trends across datasets.

**Practical Implications**
- The findings provide actionable insights for practitioners, advocating for Uncertainty Sampling (US) as the default choice when paired with a compatible model.
- The benchmark offers a transparent, standardized framework that can serve as a foundation for future AL research and applications.

---

### **Weaknesses**

**Exclusion of Deep Learning-Based Active Learning**
- The benchmark is **limited to traditional machine learning models** (e.g., Logistic Regression, SVMs, Random Forests).
- Modern AL techniques for deep learning models (e.g., CNNs, Transformers) are not included, making the conclusions less applicable.

**Lack of Computational Efficiency Analysis**
- The paper does not analyze the runtime or computational cost of different AL strategies.
- This omission is critical for practitioners deciding between accuracy gains and computational feasibility.

**Limited Exploration of Real-World Challenges**
- The benchmark focuses primarily on tabular datasets and does not address challenges like imbalanced data distributions.
- Expanding to these areas would make the benchmark more representative of real-world AL deployment scenarios.

*Over-Reliance on AUBC as the Primary Metric**
- AUBC provides a high-level summary but does not capture finer details of AL performance, such as:
  - Early-stage performance (some strategies may be better in the first few queries but degrade later).
  - Budget-limited scenarios (e.g., when the labeling budget is very small).

---

> ### Author Response · Authors · 2025-03-19
>
> Thank you for your comments!
>
> ### Critical Revisions (Necessary for Acceptance)
>
> > Clarify the Scope of the Benchmark in the Title and Abstract: The current title and abstract suggest that the benchmark is broadly applicable to active learning, but the study is limited to binary classification, tabular data and classical ML models. This can lead to misinterpretation of its generalizability.
>
> As mentioned in General Response, we clarify our scope by updating the title and extending our benchmark to multi-class classifications in Section 4.3.1 and domain-specific (image and text) datasets in Section 4.3.2.
>
> ---
>
> ### Recommended Improvements (Not Essential for Acceptance, but Strengthen the Paper)
>
> > Expand the Literature Review to Include Deep Learning-Based AL Methods
>
> We include deep learning-based AL methods and related benchmarks in Section 2.1.
>
> > Include Computational Efficiency Comparisons
>
> We include computational time for each query strategy in Appendix E.
>
> > Consider Evaluating AL on More Challenging Scenarios (Imbalanced datasets)
>
> We evaluate Uncertainty Sampling's performance on scenarios such as increasing the batch size for large-scale datasets and the one-shot protocol for the imbalanced datasets in Section 5.3.

---

### Review · Reviewer_6RaL · 2025-03-26

**Summary Of Contributions:**

The paper presents a comprehensive study of various state of the art (SOTA) active learning methods for binary classification on tabular data. It also high-lights the importance of model compatibility to uncertainty-sampling outperforming the other SOTA active learning algorithms.

**Audience:**

Yes

**Claims And Evidence:**

No

**Requested Changes:**

see above

**Strengths And Weaknesses:**

The paper's main strength and contribution is the comprehensiveness of the studied datasets (Table 3) and the various algorithmic & data characteristics (Table 1). It also performs a fairly comprehensive review of the novel active learning approaches proposed over the last decade.

The paper has two main weaknesses: the experimental setup and the presentation.

With respect  to the experimental setup, in order to correctly assess an active learner's impact on a given dataset, one most quantify how much cheaper (i.e., fewer labeled examples) it is to reach (near) SOTA performance compared to the uniform sampling scenario. For example, rather than using a fixed querying budget (e.g., 20% or 2K or 3K of the examples) that leads to a (possibly) mediocre performance, use as metric "the number of examples required to reach 90%, 95%, and 99% of the performance of training on all the available data." The values in Tables 3 & 4 show that US often outperforms other methods on the chosen budgets, but it does compare this performance against the  SOTA results when training on all labeled data.

With respect to the actual presentation, it could easily be improved to make the paper more readable:
- the abstract uses the query/task-oriented  terminology that is only introduced in section 2.2
- make it clear both in the abstract & the intro that your focus is on binary clarification; right now, only in section 3 (2nd pgf) you make it clear that this is the case
- in Section 2.1, the Model uncertainty part is presented fairly well, while the other approaches (Expected model changes, Representation exploiting, and Hybrid) are a lot vaguer and more poorly organized. Please add references to the "Expected model changes" paragraph, and explain what do you mean by model-free. Ideally, you should add a new sub-section with an illustrative example that INTUITIVELY compares & contrasts the 4 main types of approaches and their main variants.
- the second paragraph in 2.2 is poorly written ("might or might not be"), and too complicated; a simpler, intuitive 1-sentence version should be part of your abstract
- please add comprehensive, intuitive discussion on the impact of the initial labeled pool. It seems to be a key issue, for which you are simplify referring the reader to a 2022 paper. Depending on whether you solved or just alleviated this issue, this part may be a major contribution or, alternatively, red flag
- in Table 5, US performs poorly on Chekerboard and Banana. Please analyze and discuss in detail the root causes
- in all tables, make it easy to find the top-2 values (eg, bold & italic, respectively)




OTHER COMMENTS
- explain in detail and add references to the vague statement "However, these strategies often lack a fair and unified comparison across different
contexts."
- avoid weak statements such as the "could be" - as in "Our benchmark could be the most comprehensive ..." Either it is, or it is not.
- replace the weak statement "US remains competitive on most datasets" by "US is SOTA on X of the 26 studied datasets
- you are repeatedly using incorrectly the expression "first-hand"
- Settles' literature review of 2009 did not "initiate"  the study of pool-based AL, which started a decade earlier, in the 1990s
- Fig 2 - add a reference to the Australian dataset
- define the "cold start" issue from section 3
- Section 4,1: "statistically exist" --> - define "statistically significant"
- the caption of Table 3 refers itself

---

> ### Author Response · Authors · 2025-04-09
>
> ## With respect to the experimental setup
>
> > in order to correctly assess an active learner's impact on a given dataset, one most quantify how much cheaper (i.e., fewer labeled examples) it is to reach (near) SOTA performance compared to the uniform sampling scenario. For example, rather than using a fixed querying budget (e.g., 20% or 2K or 3K of the examples) that leads to a (possibly) mediocre performance, use as metric "the number of examples required to reach 90%, 95%, and 99% of the performance of training on all the available data." The values in Tables 3 & 4 show that US often outperforms other methods on the chosen budgets, but it does compare this performance against the SOTA results when training on all labeled data.
>
> We add the number of queried/labeled examples required to reach 99% (Table 5) in Section 4.1 and 90% (Table 14), 95% (Table 15) in Appendix A. Our results demonstrate that Uncertainty Sampling stably achieves first or second place on 21 datasets, which shows a consistent conclusion with Table 3 and Table 4.

---

> > ### Author Response · Authors · 2025-04-09
> >
> > Thank you for your comments!
> > We update the new version shown in violet text.
> >
> > ## With respect to the actual presentation
> >
> > > make it clear both in the abstract & the intro that your focus is on binary clarification; right now, only in section 3 (2nd pgf) you make it clear that this is the case
> >
> > We reposition our work to focus on tabular datasets and extend the existing benchmark to include multi-class classification, domain-specific (CV & NLP), and imbalanced data scenarios as mentioned in the previous comment (https://openreview.net/forum?id=855yo1Ubt2&noteId=4KkbcIz1sG).
> >
> > > the abstract uses the query/task-oriented terminology that is only introduced in section 2.2
> >
> > We make the abstract more clear by rephrasing the abstract. The main revision is to present the term **model compatibility** with a straightforward explanation. We also emphasize the idea of **model compatibility** throughout the article.
> >
> > > in Section 2.1, the Model uncertainty part is presented fairly well, while the other approaches (Expected model changes, Representation exploiting, and Hybrid) are a lot vaguer and more poorly organized.
> > > Please add references to the "Expected model changes" paragraph, and explain what do you mean by model-free.
> > > Ideally, you should add a new sub-section with an illustrative example that INTUITIVELY compares & contrasts the 4 main types of approaches and their main variants.
> >
> > Thank you for your thoughtful suggestions for improvement.
> > 1. We add references to the "Expected model changes".
> > 2. We explain the *model-free* query strategy, which does not rely on query-oriented model predictions and instead focuses on the structure/representation of data (See Section 2.1.1 and Section 2.2).
> > 3. We add an illustrative example of the types of different methods at the beginning of Section 2.1. Please see Section 2.1.1.
> >
> > > the second paragraph in 2.2 is poorly written ("might or might not be"), and too complicated; a simpler, intuitive 1-sentence version should be part of your abstract
> >
> > We re-write this parapgraph. The outline is
> > 1. Define the term of **model compatibility**, which is the setting reflects the relationship between the queiry-oreinted model and the task-oriented model.
> > 2. We observe that the **model compatibility** is the over-looked setting in existing tabular benchmarks and its influence is unclear to Uncertainty Sampling.
> >
> > > please add comprehensive, intuitive discussion on the impact of the initial labeled pool. It seems to be a key issue, for which you are simplify referring the reader to a 2022 paper. Depending on whether you solved or just alleviated this issue, this part may be a major contribution or, alternatively, red flag
> >
> > We add the discussion about the initial sets (including the initial labeled pool and the unlabeled pool). Ji et al. (2023) recommended using consistent "initial sets" across multiple runs to minimize the impact of randomness and ensure fair comparisons.
> > In our study, randomness arises from the train-test split and the construction of initial sets. While the randomness from the train-test split is unavoidable without predefined training and test sets, we adopt Ji et al.'s suggestions by keeping the initial labeled sets fixed to mitigate randomness. The detailed experimental results are shown in Appendix C.3. In summary, our findings indicate that Uncertainty Sampling exhibits consistent performance across most datasets, confirming its stability within our benchmark.
> >
> > > in Table 5, US performs poorly on Chekerboard and Banana. Please analyze and discuss in detail the root causes
> >
> > We add Figure 6 and Figure 7 to study the failure of Uncertainty Sampling by visualizing the scatter plots at the initial and internal rounds within the active learning process. We observe that when datasets have multiple overlapping positive and negative regions, Uncertainty Sampling tends to query examples from these overlapping regions rather than exploring less-covered regions, particularly due to the uneven distribution of the initial labeled pool.
> >
> > > in all tables, make it easy to find the top-2 values (eg, bold & italic, respectively)
> >
> > We update tables with **bold** for the first place and *italic* for the second place performance.

---

> > > ### Author Response · Authors · 2025-04-09
> > >
> > > ## OTHER COMMENTS
> > >
> > > > explain in detail and add references to the vague statement "However, these strategies often lack a fair and unified comparison across different contexts."
> > >
> > > > Settles' literature review of 2009 did not "initiate" the study of pool-based AL, which started a decade earlier, in the 1990s
> > >
> > > We remove vague or imprecise statements without supportive references.
> > >
> > > ---
> > >
> > > > avoid weak statements such as the "could be" - as in "Our benchmark could be the most comprehensive ..." Either it is, or it is not.
> > >
> > > > replace the weak statement "US remains competitive on most datasets" by "US is SOTA on X of the 26 studied datasets
> > >
> > > > you are repeatedly using incorrectly the expression "first-hand"
> > >
> > > > Fig 2 - add a reference to the Australian dataset
> > >
> > > > define the "cold start" issue from section 3
> > >
> > > > Section 4,1: "statistically exist" --> - define "statistically significant"
> > >
> > > > the caption of Table 3 refers itself
> > >
> > > We revise the contents according to your comments.

---

> ### Comment · Reviewer_6RaL · 2025-04-13
> **Thank you very much for the additional experiments & clarifications**
>
> Thank you very much for the performing and adding the results for the additional experiments. IMO, it makes the paper  significantly more compelling. I also appreciate you tackling the other suggested improvements

---

### Comment · Action_Editor_zQ4n · 2024-12-04
**Currently not enough reviewers available**

Dear authors,

My apologies for the delays. We have one review for the paper so far (thank you so much to that reviewer!). A second review will come in by December 31st, and I hope I'll be able to find the required third reviewer eventually. I don't expect that to change within the next weeks, though, to be honest, as there is also NeurIPS coming up, and reviewer availability will further decline over the holiday period. My apologies for this.

Best wishes,\
 Andreas

---

### Decision · Action_Editor_zQ4n · 2025-05-18

**Recommendation:** Accept as is

**Comment:**

I recommend accepting this paper based on the thorough reviews, the authors' comprehensive responses, and the revisions to the paper.

All three reviewers have provided positive recommendations, and the authors have addressed the concerns raised during the review process.

I want to thank the authors and reviewers for their patience as this paper went through the review process. I'm very glad to note the positive interactions that have improved the paper towards acceptance.

**Audience:**

This paper would certainly be of interest to a significant portion of TMLR's audience. Active Learning continues to be an important area of research in machine learning, particularly for applications where labeled data is expensive or difficult to obtain. The paper's comprehensive benchmark and analysis of Uncertainty Sampling provide insights for both researchers and practitioners working with tabular data, which remains prevalent in many real-world applications across science, finance, and healthcare.

**Claims And Evidence:**

The claims made in this submission are well-supported. The authors present a comprehensive benchmark for Active Learning (AL) on tabular datasets, with a particular focus on Uncertainty Sampling (US). They provide extensive empirical evidence through experiments across multiple datasets, models, and query strategies. The paper demonstrates that when paired with compatible models, Uncertainty Sampling remains highly competitive compared to other AL strategies. The authors have addressed reviewer concerns by expanding their analysis to include multi-class classification, domain-specific datasets, and imbalanced data scenarios, strengthening the validity of their claims.